# Revisiting Source-Free Domain Adaptation: A New Perspective via Uncertainty Control

**Gezheng Xu**[1,*]  **Hui Guo**[1,*]  **Li Yi**[4]  **Charles Ling**[1,2]  **Boyu Wang**[1,2,†]  **Grace Y. Yi**[1,2,3,†]
[1]Department of Computer Science, University of Western Ontario  [2]Vector Institute
[3]Department of Statistical and Actuarial Sciences, University of Western Ontario  [4]Tiktok
`{gxu86, hguo288, charles.ling, gyi5}@uwo.ca`
`li.yi@bytedance.com, bwang@csd.uwo.ca`

## Abstract

Source-Free Domain Adaptation (SFDA) seeks to adapt a pre-trained source model to a target domain using only unlabeled target data, without access to the original source data. While current state-of-the-art methods rely on leveraging weak supervision from the source model to extract reliable information for self-supervised adaptation, they often overlook the uncertainty that arises during the transfer process. In this paper, we conduct a systematic and theoretical analysis of the uncertainty inherent in existing SFDA methods and demonstrate its impact on transfer performance through the lens of Distributionally Robust Optimization. Building upon the theoretical results, we propose a novel instance-dependent uncertainty control algorithm for SFDA. Our method quantifies and exploits the uncertainty during adaptation, significantly improving model performance. Extensive experiments on benchmark datasets and empirical analyses confirm our theoretical findings and the effectiveness of the proposed method. This work offers new insights into understanding and advancing SFDA performance. We release our code at `https://github.com/xugezheng/UCon_SFDA`.

## 1 Introduction

Deep neural networks (DNNs) have achieved remarkable performance across a wide range of tasks. However, their performance can degrade significantly when a domain shift occurs between training (source) and test (target) data. Traditional solutions rely on transferable knowledge from labeled source data to classify unlabeled target data, but access to source data is often restricted due to privacy concerns or proprietary constraints. To address this, Source-Free Domain Adaptation (SFDA) has emerged as a solution, aiming to adapt a pre-trained source model to an unlabeled target domain without accessing the original source data (Liang et al., 2020; Yang et al., 2021b;a).

Recent work has integrated self-supervised learning with transfer learning in SFDA, with contrastive learning-based self-supervised methods gaining widespread use and empirical support (Yang et al., 2022; Karim et al., 2023; Chen et al., 2022; Hwang et al., 2024; Mitsuzumi et al., 2024). A key challenge in applying contrastive learning methods to SFDA lies in selecting and utilizing positive and negative samples of target data using a well-trained source model. Different from conventional contrastive learning methods using data augmentations as positive samples, in SFDA, the neighbors in the feature space can provide stronger supervision and usually be treated as positives, and the negative samples are the remaining data in the training mini-batch. However, due to the domain shift, these methods face severe uncertainty, as will be elaborated shortly.

In this paper, we systematically and theoretically examine the uncertainty present in SFDA through the lens of Distributionally Robust Optimization (DRO). Unlike previous studies that primarily focus on empirical strategies (Roy et al., 2022; Litrico et al., 2023; Pei et al., 2023; Lee et al., 2022), our work offers a comprehensive analysis of two types of uncertainty arising from the use of negative and positive samples in existing SFDA methods, aiming to enhance SFDA performance by explicitly controlling the uncertainty. Specifically, on one hand, random sampling of negative samples in

---

*Equal contribution, †Corresponding Authors

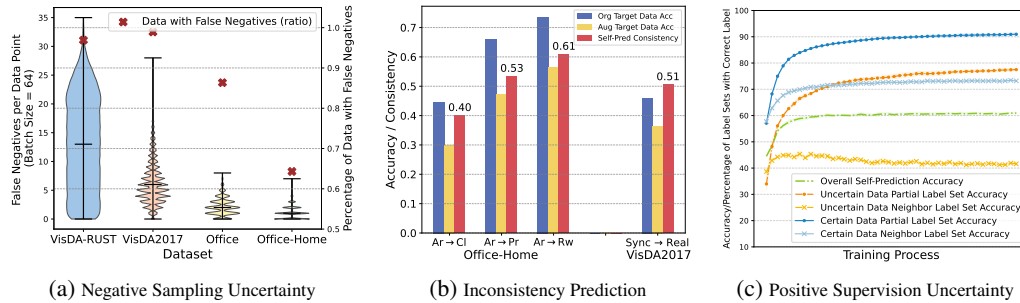

(a) Negative Sampling Uncertainty    (b) Inconsistency Prediction    (c) Positive Supervision Uncertainty

Figure 1: *This figure summarizes key findings from our experiments, to be detailed in Section 4.4: (a) Presence of false negatives across different datasets; (b) Inconsistency between predictions for original images and their augmented views by the source model; (c) Illustration of varying predictive accuracies between certain and uncertain target data during adaptation on Office-Home (Ar → Cl).*

applications often introduces outliers, or "false negatives" – samples that belong to the same class as the considered target data point but are mistakenly selected as negatives (as shown in Figure 1a). This discrepancy leads to a deviation of the empirical negative distribution from the true distribution, thus introducing variability into the loss calculation. To address this sampling bias, we introduce a *negative uncertainty set*, which consists of distributions obtained by slightly perturbing the training negative distribution, and consider an outlier-robust worst-case risk within this set. We theoretically derive an informative upper bound for this risk, which motivates incorporating a dispersion control term into the loss function. Moreover, to address the prediction inconsistency between a target image and its augmented view (as shown in Figure 1b), we propose an augmentation-based dispersion control approach to mitigate uncertainty introduced by noisy negative samples. On the other hand, domain shift causes models trained on source data to produce uncertain probability estimates for target data. In such cases, the supervisory information from positive examples may not fully align with the ground truth, making the use of neighboring predictions for supervision introduce additional uncertainty. Unlike existing methods that focus on mitigating uncertainty (Roy et al., 2022; Litrico et al., 2023; Mitsuzumi et al., 2024), we aim to utilize this information more effectively. To better accommodate the uncertainty in the predicted probabilities of positive samples, we consider a *positive uncertainty set* centered around these probabilities and examine the worst-case risk within this set. We theoretically show that the optimal solution for target points involves a partial label set. To make the most of this uncertain information, we propose novel criteria to identify uncertain data and use partial labels to relax supervision for these samples. As illustrated in Figure 1c, leveraging such uncertainty information improves performance compared to relying only on certain data.

Our contributions are as follows: (1) We theoretically analyze two key sources of uncertainty in contrastive learning-based SFDA methods, identifying two types of worst-case risks under a unified DRO framework. This investigation explains why current contrastive learning methods can significantly improve SFDA performance (Section 4.2) while revealing the overlooked uncertain information in existing methods (Section 4.3). Our theoretical analysis also provides a novel perspective in understanding the SFDA problem. (2) Based on our theoretical result, we design a novel Uncertainty Control algorithm for SFDA (UCon-SFDA) to minimize the negative effects of uncertainty in negative sample selection while leveraging the uncertain information from positive sample predictions to enhance the model's discriminability (Section 4.4). (3) We conduct extensive experiments to demonstrate the effectiveness of the proposed method.

## 2 RELATED WORK

**Source-Free Domain Adaptation (SFDA).** SFDA focuses on adapting a well-trained source model to a target domain with only unlabeled target data. Since source data are not accessible during adaptation, some methods extract source information through prototype generation (Qiu et al., 2021), or minimization of reliance on source data through adversarial training (Li et al., 2020b). To address the lack of labels for target data, several methods aim to enhance supervision. For example, SHOT (Liang et al., 2020) employs deep clustering to create pseudo-labels, while NRC (Yang et al., 2021a) and G-SFDA (Yang et al., 2021b) leverage neighboring predictions to guide the

adaptation process. Yi et al. (2023) formulates SFDA as a problem of learning from noisy labels. Recently, self-supervised learning has gained attention in SFDA, particularly contrastive learning-based self-supervised methods. For instance, AaD (Yang et al., 2022) introduces positive and negative samples into SFDA and uses a simplified contrastive loss to enhance model discriminability while maintaining diversity; C-SFDA (Karim et al., 2023) utilizes a teacher-student framework to enhance the self-training in SFDA; methods like DaC (Zhang et al., 2022), AdaContrast (Chen et al., 2022), and SF(DA)$^2$ (Hwang et al., 2024) explore explicit or implicit data augmentation to further boost SFDA performance. I-SFDA (Mitsuzumi et al., 2024) offers a new perspective by approaching SFDA through self-training. Despite these advancements, a comprehensive theoretical framework explaining their effectiveness remains absent. Moreover, most existing methods do not fully account for the uncertainty inherent in the adaptation process, which can negatively impact SFDA performance.

**Uncertainty in SFDA.** Given the absence of both source data and target labels, handling uncertainty is a key challenge in SFDA, particularly in the presence of domain shifts. Most existing research addresses prediction or representation uncertainty by reweighting loss functions or prioritizing more confident samples during training (Roy et al., 2022; Litrico et al., 2023; Pei et al., 2023; Lee et al., 2022). In contrast to these approaches, we provide a systematic and comprehensive analysis of various sources of uncertainty in contrastive learning-based SFDA from an instance-dependant perspective. Building on this analysis, we propose a novel algorithm that improves SFDA performance by effectively controlling variance during adaptation.

## 3 PRELIMINARIES

We use $[k]$ to denote the set $\{1, \ldots, k\}$ for any positive integer $k$. For $a \in \mathbb{R}$, we define $a_+ = \max\{a, 0\}$, and let $\lfloor a \rfloor$ and $\lceil a \rceil$ denote the floor and the ceiling of $a$, respectively. Let $a \wedge b$ denote $\min(a, b)$ for $a, b \in \mathbb{R}$. For a vector $v$, the $j$th element is denoted as $v_j$, and $v^\top$ indicates its transpose. For vectors $v_1$ and $v_2$, their inner product is denoted $\langle v_1, v_2 \rangle$. Let $\mathbb{1}(\cdot)$ represent the indicator function.

**Problem Setup.** For a $K$-class classification problem, let $\mathcal{X} \subset \mathbb{R}^d$ represent the input space, and let $\mathcal{Y} = [K]$ denote the label space, with $d$ denoting the input dimension. Let X and Y denote the random input and label, respectively, and let x and y represent their realizations. In SFDA, we assume that the source domain distribution $P_{xy}^S$ and the target domain distribution $P_{xy}^T$ are unknown distributions over $\mathcal{X} \times \mathcal{Y}$ that may be distinct. We factorize these distributions as products of the marginal and conditional distributions for the corresponding variables, indicated by the subscripts: $P_{xy}^S = P_x^S P_{y|x}^T$ and $P_{xy}^T = P_x^T P_{y|x}^T$. For the source domain, we have a *source model* $h_S : \mathcal{X} \to \mathcal{Y}$, which, for example, is a DNN-based predictor pre-trained with $N_S$ labeled examples $\mathcal{D}_S \triangleq \{x_i^S, y_i^S\}_{i=1}^{N_S}$ drawn from $P_{xy}^S$. In the target domain, let $\mathcal{D}_T \triangleq \{x_i^T\}_{i=1}^{N_T}$ denote the unlabeled target domain data, consisting of $N_T$ observations of independent and identically distributed (i.i.d.) random variables drawn from $P_x^T$, and is used as the training set. Given the source model $h_S$ and unlabeled target data $\mathcal{D}_T$, our goal is to learn a *target model* $h_T : \mathcal{X} \to \mathcal{Y}$ that predicts labels in the target domain by adapting $h_S$ on $\mathcal{D}_T$.

To facilitate our analysis in the context of deep learning, let $f_T(\cdot; \boldsymbol{\theta}_T) : \mathcal{X} \to \Delta^{K-1}$ denote the network output for the target data, indexed by the parameter vector $\boldsymbol{\theta}_T$ associated with the DNN architecture, taking values in the parameter space $\boldsymbol{\Theta}$, where $\Delta^{K-1}$ denotes the $K$-simplex, and the $j$th component of the vector-valued function $f_T(x; \boldsymbol{\theta}_T)$, denoted $f_T(x; \boldsymbol{\theta}_T)[j]$, models the predicted conditional probability $\mathbb{P}(Y = j | X = x)$ for the target domain data. We then define the target model $h_T$ as $h_T(x; \boldsymbol{\theta}_T) = \arg\max_{j \in [K]} f_T(x; \boldsymbol{\theta}_T)[j]$ for any $x \in \mathcal{X}$. Similarly, the source model $h_S$, along with $f_S(\cdot; \boldsymbol{\theta}_S) : \mathcal{X} \to \Delta^{K-1}$ and $\boldsymbol{\theta}_S$, is defined for the source data. In applications, the source and target models $h_S(\cdot; \boldsymbol{\theta}_S)$ and $h_T(\cdot; \boldsymbol{\theta}_T)$, or more specifically $f_S(\cdot; \boldsymbol{\theta}_S)$ and $f_T(\cdot; \boldsymbol{\theta}_T)$, are often specified with the same network architecture, with the target model's parameters initialized from the source model. Consequently, we use the generic notation $h(\cdot; \boldsymbol{\theta})$ or $f(\cdot; \boldsymbol{\theta})$ to represent these models without distinction in the following development, unless it is needed to distinguish them.

Given an anchor point X = x from the target set (i.e., the data point used as a reference to determine positive and negative examples), a positive example refers to a sample in the target set $\mathcal{D}_T$ that belongs to the same class as x, and a negative example refers to a sample from a different class. The latter is also called a "true negative", in contrast to the "false negative" mentioned in Section 1, which refers to a sample that belongs to the same class as the anchor point but is mistakenly selected as a negative.

## 4 THEORETICAL ANALYSIS AND ALGORITHM

### 4.1 MOTIVATION

Existing SFDA methods typically decompose their training loss into two components: (1) discriminability, which enhances the model's ability to distinguish between unlabeled target samples, and (2) diversity, which encourages predictions to be distributed across different classes (Yang et al., 2022; Mitsuzumi et al., 2024; Cui et al., 2020). Among these approaches, contrastive learning methods are perhaps the most widely used, where the goal is to maximize the similarity between positive pairs to improve discriminability and minimize the similarity between negative pairs to ensure diversity. This can be formulated as the following expected risk with contrastive loss:

$$\mathcal{R}_{\text{basic}}(\boldsymbol{\theta}) = \mathbb{E}_{P_{\text{x}}^{\text{T}}} \left[ -\mathbb{E}_{P^+}\{\mathcal{S}_{\boldsymbol{\theta}}(X^+;X)\} + \mathbb{E}_{P^-}\{\mathcal{S}_{\boldsymbol{\theta}}(X^-;X)\} \right], \tag{1}$$

where the function $\mathcal{S}_{\boldsymbol{\theta}}(\cdot;\cdot)$, mapping from $\mathcal{X} \times \mathcal{X}$ to $[0,1]$, represents the similarity measure between two instances, such as cosine similarity, as detailed in Section 3. The outer expectation $\mathbb{E}_{P_{\text{x}}^{\text{T}}}$ is taken with respect to the input distribution for X in the target domain, and the inner expectations $\mathbb{E}_{P^+}$ and $\mathbb{E}_{P^-}$ are evaluated under the conditional distributions of positive example $X^+$ and negative example $X^-$, respectively, given X.

In contrastive learning-based SFDA, for each target input $x_i^{\text{T}}$ in a mini-batch $\mathcal{B}$, the set of positive examples relative to $x_i^{\text{T}}$, denoted $\mathcal{C}_i$, consists of its $\kappa$-nearest neighbours in the target domain data $\mathcal{D}_{\text{T}}$, where $\kappa$ is typically chosen between 2 and 5. The negative set is taken as $\mathcal{B}\setminus\{x_i^{\text{T}}\}$, which, however, inevitably includes a fraction of false negatives, introducing sampling bias. While a well-trained source model helps to ensure that neighboring positive samples in the feature space provide effective supervision for most unlabeled target data, some highly uncertain samples persist due to domain shift. To address these issues, we propose a robust strategy for managing uncertainty in SFDA using DRO.

### 4.2 NEGATIVE SAMPLING UNCERTAINTY AND DISPERSION CONTROL

To address sampling bias and distribution shift in negative examples, we formulate an expected DRO risk: for each given $x \in \mathcal{X}$ and $\delta > 0$,

$$\mathcal{R}_{\text{x}}^-(\boldsymbol{\theta}; P^-, \delta) = \sup_{Q^- \in \Gamma_\delta(P^-)} \left[ \mathbb{E}_{Q^-}\{\mathcal{S}_{\boldsymbol{\theta}}(X^-;x)\} \right], \tag{2}$$

where the expectation $\mathbb{E}_{Q^-}\{\mathcal{S}_{\boldsymbol{\theta}}(X^-;x)\}$ is evaluated under the conditional distribution $Q^-$ of $X^-$, given $X = x$, taken from the set $\Gamma_\delta(P^-)$. Here, $\Gamma_\delta(P^-)$ represents an *uncertainty set* of probability measures centered around the *reference probability distribution* $P^-$, with a radius $\delta > 0$ that facilitates robustness. Commonly, $\Gamma_\delta(P^-)$ is defined as the distance-based uncertainty set:

$$\Gamma_\delta(P^-) = \{Q^- \in \mathcal{P}_p(\mathcal{X}) : d(Q^-, P^-) \leq \delta\}, \tag{3}$$

where $\mathcal{P}_p(\mathcal{X})$ denotes the class of Borel probability measures on $\mathcal{X}$ with finite $p$th moment for some $p > 1$, and $d$ is a discrepancy metric of probability measures. Popular choices of $d$ are $\varphi$-divergences (including Kullback–Leibler (KL) divergence and $\chi^2$ divergence as special cases (Duchi, 2016)) and Wasserstein distances (Gao, 2023; Gao et al., 2024; Blanchet & Murthy, 2019).

In practice, negative samples are often drawn uniformly from the training data, which frequently leads to the inclusion of false negatives. Let $P_{\text{train}}^-$ represent the distribution of these negative samples, modeled using Huber's $\epsilon$-contamination method: $P_{\text{train}}^- = (1-\epsilon)P^- + \epsilon\widetilde{P}^-$, where $\epsilon \in (0,1)$ is the contamination level, and $\widetilde{P}^-$ represents an arbitrary contamination distribution (Huber, 1992). For instance, consider some $x \in \mathcal{X}$. Suppose we collect $n$ negative samples, where a fraction $\lfloor \varepsilon n \rfloor$ are i.i.d. false negatives drawn from $\widetilde{P}^-$, and the rest are true negatives from $P^-$. The resulting empirical distribution of the observed negative samples follows this model with a contamination level of $\lfloor \varepsilon n \rfloor/n$. To mitigate overfitting to worst-case instances that are likely to be outliers, we minimize a refined outlier-robust expected risk (Nietert et al., 2024a;b; Zhai et al., 2021):

$$\mathcal{R}_{\text{x}}^-(\boldsymbol{\theta}; P_{\text{train}}^-, \delta, \epsilon) = \inf_{P' \in \mathcal{P}_p(\mathcal{X})} \left\{ \mathcal{R}_{\text{x}}^-(\boldsymbol{\theta}; P', \delta) : \exists \widetilde{P}' \in \mathcal{P}_p(\mathcal{X}) \ s.t. \ P_{\text{train}}^- = (1-\epsilon)P' + \epsilon\widetilde{P}' \right\}. \tag{4}$$

By definition, the minimizer of (4) is designed to ignore "hard" data points that contribute the most to the worst-case risk and instead focus on the $(1-\epsilon)$-fraction of "easy" data points in the training set.

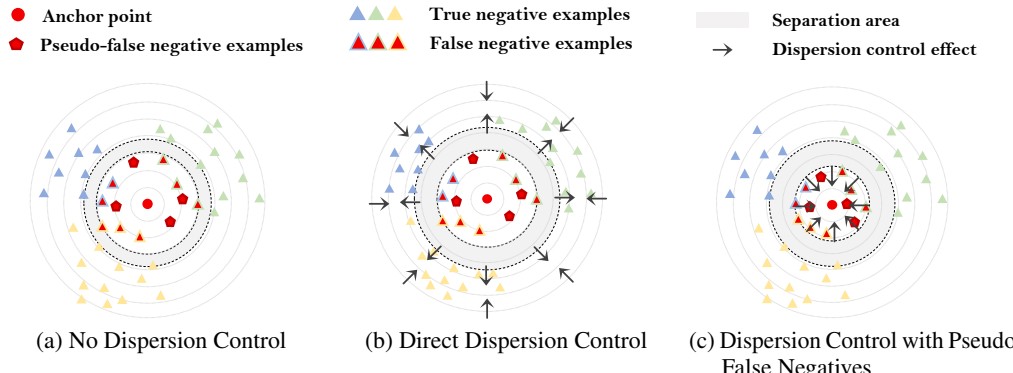

Figure 2: *Illustrative visualization of the effect of dispersion control: (a) No dispersion control, (b) Direct dispersion control between the anchor and false-negative pairs, and (c) Dispersion control with pseudo-false negatives.*

This helps prevent overfitting to outliers, thereby reducing the risk of pushing the target data point away from others within the same class. For different choices of the discrepancy metric $d$ in the uncertainty set (3), we establish a unified upper bound on the outlier-robust risk $\mathcal{R}_x^-(\boldsymbol{\theta}; P_{\text{train}}^-, \delta, \epsilon)$.

Specifically, for an $\epsilon$-contaminated training distribution $P_{\text{train}}^-$ with $0 < \epsilon < 1$, let $p_{\text{train}}^-$ denote the associated density or mass function. For any given $x \in \mathcal{X}$, define the associated truncated distribution $P^*$ with density/mass function $\rho^*$: $p^*(x^-) \triangleq \frac{1}{1-\epsilon} p_{\text{train}}^-(x^-) \mathbb{1}\{\mathcal{S}_{\boldsymbol{\theta}}(x^-; x) \leq s^*\}$ for any negative example $x^-$ of $x$, where $s^*$ is the $1 - \epsilon$ quantile satisfying $P_{\text{train}}^-\{\mathcal{S}_{\boldsymbol{\theta}}(X^-; x) \leq s^*\} = 1 - \epsilon$. In contrastive SFDA, for each anchor point $x$ from the target set, the truncated version of $P_{\text{train}}^-$, denoted as $P^*$, concentrates all its mass on regions where the similarity falls below the $(1 - \epsilon)$-quantile. In the following theorem, let $\Re_1 \triangleq \frac{1}{1-\epsilon} \int_0^{s^*} s \, dP_{\text{train}}^- \{\mathcal{S}_{\boldsymbol{\theta}}(X^-; x) \leq s\}$, $\Re_2 \triangleq \frac{1}{1-\epsilon} \int_0^{s^*} s^2 \, dP_{\text{train}}^- \{\mathcal{S}_{\boldsymbol{\theta}}(X^-; x) \leq s\}$, $\mathbb{E}_{P^*}\{\mathcal{S}_{\boldsymbol{\theta}}(X^-; x)\} = \Re_1$, and $\mathbb{V}_{P^*}\{\mathcal{S}_{\boldsymbol{\theta}}(X^-; x)\} = \Re_2 - \Re_1^2$.

**Theorem 4.1.** *Suppose the similarity measure $\mathcal{S}_{\boldsymbol{\theta}}$ satisfies the smoothness conditions in Lemma 4 in Appendix A.3 for all $\boldsymbol{\theta} \in \Theta$. Then for a small $\delta > 0$ and for different choices of the discrepancy metric $d$ in (3), we have the following upper bounds on the risk $\mathcal{R}_x^-(\boldsymbol{\theta}; P_{train}^-, \delta, \epsilon)$ in (4):*

*(i) If $d$ is the $\chi^2$-divergence, then*
$$\mathcal{R}_x^-(\boldsymbol{\theta}; P_{train}^-, \delta, \epsilon) \leq \mathbb{E}_{P^*}\{\mathcal{S}_{\boldsymbol{\theta}}(X^-; x)\} + \sqrt{\delta \mathbb{V}_{P^*}\{\mathcal{S}_{\boldsymbol{\theta}}(X^-; x)\}}.$$

*(ii) If $d$ is the KL-divergence, then*
$$\mathcal{R}_x^-(\boldsymbol{\theta}; P_{train}^-, \delta, \epsilon) \leq \mathbb{E}_{P^*}\{\mathcal{S}_{\boldsymbol{\theta}}(X^-; x)\} + \sqrt{2\delta \mathbb{V}_{P^*}\{\mathcal{S}_{\boldsymbol{\theta}}(X^-; x)\}} + \mathcal{O}(\delta).$$

*(iii) If $d$ is the $p$-Wasserstein distance with $p \in [1, +\infty)$ and the cost function $c(\cdot, \cdot)$ in Definition A.1 is chosen as a norm $\|\cdot\|$ with the dual norm $\|\cdot\|_*$, then for $q$ satisfying $\frac{1}{p} + \frac{1}{q} = 1$,*
$$\mathcal{R}_x^-(\boldsymbol{\theta}; P_{train}^-, \delta, \epsilon) \leq \mathbb{E}_{P^*}\{\mathcal{S}_{\boldsymbol{\theta}}(X^-; x)\} + \delta\{\mathbb{E}_{P^*}\|\nabla\mathcal{S}_{\boldsymbol{\theta}}(X^-; x)\|_*^q\}^{1/q} + \mathcal{O}(\delta^{2 \wedge p}).$$

**Remark 4.1.** Theorem 4.1 demonstrates that $\mathcal{R}_x^-(\boldsymbol{\theta}; P_{\text{train}}^-, \delta, \epsilon)$ is essentially upper bounded by the sum of two key terms regardless of a specific form of $d$ considered in the theorem. The first term controls the average risk over potential true negatives, behaving similarly to the traditional negative sample loss (Yang et al., 2022). This is implemented as the negative sample loss $\mathcal{L}_{\text{CL}}^-$ in (7), to be presented in Section 4.4. The second term facilitates the dispersion in similarity between these true negatives, helping to distinguish anchor-true-negative pairs from anchor-false-negative ones; it encourages greater separation between prediction similarities for anchor-true-negative pairs and anchor-false-negative pairs, as shown by the wider gray area in Figure 2b than that in Figure 2a.

**Remark 4.2.** In practice, domain shift makes it challenging to distinguish between false negatives and true negatives. To address this, we propose to achieve dispersion control by manually constructing pseudo-false negatives using techniques such as data augmentation. As shown in Figure 1b, for a given anchor point $x$, the source model's prediction on its augmented version, denoted as $\text{AUG}(x)$, may differ from the prediction for $x$. When this happens, $\text{AUG}(x)$ is automatically treated as a false

negative for x. Motivated by the dispersion control term, we treat these augmentations as pseudo-false negatives and minimize the negative similarity between the anchor point and its augmented prediction. As illustrated in Figure 2c, this approach effectively amplifies the contrast between anchor-false-negative similarity and anchor-true-negative similarity, expanding the width of the gray region to improve the separation of the data points in the same class as the anchor point from the true negatives. This dispersion control effect is captured through the loss term $\mathcal{L}_{\mathrm{DC}}^{-}$ in (7), as detailed in Section 4.4.

### 4.3 POSITIVE SUPERVISION UNCERTAINTY AND PARTIAL LABELING

For each anchor point x in the target domain $\mathcal{D}_{\mathrm{T}}$, let $p \triangleq (p_1, \ldots, p_K)^\top \triangleq f(\mathrm{x}; \boldsymbol{\theta}) \in \Delta^{K-1}$ denote the target model's predicted probabilities for x, where $f(\cdot; \boldsymbol{\theta})$ is described at the end of Section 3, and $p_j$ represents the $j$th component of $f(\mathrm{x}; \boldsymbol{\theta})$. Similarly, for the positive example $\mathrm{x}^+$ associated with x, let $p^+ \triangleq (p_1^+, \ldots, p_K^+)^\top \triangleq f(\mathrm{x}^+; \boldsymbol{\theta}) \in \Delta^{K-1}$ represent the predicted probabilities for $\mathrm{x}^+$. When using cosine similarity, the supervision information from the positive example $\mathrm{x}^+$ encourages the model training to minimize the negative similarity, defined as $-\langle p^+, p \rangle = -\sum_{j=1}^K p_j p_j^+$.

In SFDA, leveraging a well-trained source model and the similarity between the source and target domain distributions, neighboring examples in the feature space are often treated as positive samples. While many of these positive samples provide effective supervision for unlabeled target data, there can still be highly uncertain examples due to domain shift. To better handle this uncertainty in model predictions, we explore the optimal prediction for an anchor point x by solving the following worst-case risk minimization problem based on DRO:

$$p^\star \in \inf_{p \in \Delta^{K-1}} \mathcal{R}_{\mathrm{x}}^+(p; \mathrm{x}^+, \delta), \text{ with } \mathcal{R}_{\mathrm{x}}^+(p; \mathrm{x}^+, \delta) \triangleq \sup_{q^+ \in \Gamma_\delta(p^+)} \langle q^+, -p \rangle, \tag{5}$$

where $\Gamma_\delta(p^+)$ is the uncertainty set centered around the reference distribution $p^+$, as defined in (3). Using the proof techniques in Guo et al. (2024), we derive a closed-form expression for $p^\star$ as follows.

**Theorem 4.2.** *Let $\{p_1^+, \ldots, p_K^+\}$ be arranged in decreasing order, denoted as $p_{(1)}^+ \geq \ldots \geq p_{(K)}^+$, with the corresponding indexes denoted as $\chi(1), \ldots, \chi(K)$. If $p$-Wasserstein distance with the 0-1 cost function is used, then the optimal solution $p^\star \triangleq (p_1^\star, \ldots, p_K^\star)^\top$ of (5) is given as follows:*

*(i) If $\frac{1}{K} \geq \frac{1}{k^*} \sum_{j=1}^{k^*} p_{(j)}^+ - \frac{1}{k^*} \delta^p$ for all $k^* \in [K-1]$, then $p_j^\star = \frac{1}{K}$ for all $j \in [K]$.*

*(ii) If there exists some $k_0 \in [K-1]$ such that $\frac{1}{k_0} \sum_{j=1}^{k_0} p_{(j)}^+ - \frac{1}{k_0} \delta^p > \frac{1}{K}$ and $\frac{1}{k_0} \sum_{j=1}^{k_0} p_{(j)}^+ - \frac{1}{k_0} \delta^p \geq \frac{1}{k^*} \sum_{j=1}^{k^*} p_{(j)}^+ - \frac{1}{k^*} \delta^p$ for all $k^* \in [K-1]$, then $p_{(j)}^\star = \frac{1}{k_0}$ for $j \in [k_0]$ and $p_{(j)}^\star = 0$ for $j = k_0 + 1, \ldots, K$, where $p_{(j)}^\star$ denotes the $\chi(j)$th element of $p^\star$ corresponding to $p_{(j)}^+$.*

**Remark 4.3.** Theorem 4.2 suggests that the optimal prediction for an anchor point can be represented by a set of *instance-dependent partial labels*. The advantage of using partial labels, rather than the full predicted probabilities, as the supervision signal is that it retains uncertain yet potentially more accurate label information, while eliminating interference from labels that are more likely to be incorrect. In the special case where $p_{(1)}^+ \geq \max\{\frac{1}{K} + \delta^p, p_{(2)}^+ + \delta^p\}$, the optimal solution simplifies to $p_{(1)}^\star = 1$ and $p_{(j)}^\star = 0$ for $j = 2, \ldots, K$. That is, the optimal solution selects the label with the highest predicted probability for the anchor point, rather than a set of partial labels, when the gap between the top two probabilities exceeds a given threshold. We refer to this scenario as *certain label information*; otherwise, we classify it as *uncertain label information*.

**Remark 4.4.** Motivated by Theorem 4.2 and Remark 4.3, we propose to leverage both certain and uncertain label information in distinct ways to effectively capture and utilize prediction uncertainty. Specifically, when an instance x receives *certain label information*, the optimal prediction for x corresponds to the label with the highest predicted probability. This certain supervision signal is incorporated through the *positive supervision loss* term $\mathcal{L}_{\mathrm{CL}}^+$ in (8). When *uncertain label information* is present, the optimal prediction for x is expressed as a set of partial labels. Instead of relying solely on the estimated pseudo labels, we construct a *partial label set* for x. This approach offers a more robust supervisory signal by accounting for multiple potential labels and reducing reliance on noisy single-label predictions. This information is captured through the *partial label loss* term $\mathcal{L}_{\mathrm{PL}}^+$ in (8). To distinguish between certain and uncertain label information in applications, we use the ratio of the two highest predicted probabilities, as detailed in Section 4.4.

## 4.4 IMPLEMENTATION

Our algorithm builds upon the conventional contrastive loss commonly adopted in previous works (Yang et al., 2022; Mitsuzumi et al., 2024):

$$\mathcal{L}_{\mathrm{CL}} \triangleq \mathcal{L}_{\mathrm{CL}}^{+} + \lambda_{\mathrm{CL}}^{-}\mathcal{L}_{\mathrm{CL}}^{-}, \tag{6}$$

where $\mathcal{L}_{\mathrm{CL}}^{+} = \frac{1}{N_{\mathrm{T}}}\sum_{i=1}^{N_{\mathrm{T}}}\big\{ -\sum_{\mathrm{x}_i^{+}\in\mathcal{C}_i}\mathcal{S}_{\boldsymbol{\theta}}(\mathrm{x}_i^{+};\mathrm{x}_i)\big\}$; $\mathcal{L}_{\mathrm{CL}}^{-} = \frac{1}{N_{\mathrm{T}}}\sum_{i=1}^{N_{\mathrm{T}}}\sum_{\mathrm{x}_i^{-}\in\mathcal{B}\backslash\{\mathrm{x}_i\}}\mathcal{S}_{\boldsymbol{\theta}}(\mathrm{x}_i^{-};\mathrm{x}_i)$ with $\mathcal{B}$ denoting the mini-batch; $\lambda_{\mathrm{CL}}^{-}$ represents a tuning parameter; and similarity is computed as $\mathcal{S}_{\boldsymbol{\theta}}(\mathrm{x}_i^{+};\mathrm{x}_i) = \langle f(\mathrm{x}_i^{+};\boldsymbol{\theta}), f(\mathrm{x}_i;\boldsymbol{\theta})\rangle$ or $\mathcal{S}_{\boldsymbol{\theta}}(\mathrm{x}_i^{-};\mathrm{x}_i) = \langle f(\mathrm{x}_i^{-};\boldsymbol{\theta}), f(\mathrm{x}_i;\boldsymbol{\theta})\rangle$. Here, positive samples are the $\kappa$-nearest neighbours in the feature space from the training set $\mathcal{D}_{\mathrm{T}}$, and negative samples are the remaining data points in the same mini-batch $\mathcal{B}$. Building on this simple yet widely adopted implementation in SFDA, our approach focuses on effectively controlling uncertainty during the adaptation process by refining both the negative and positive sample components.

**Dispersion Control via Data Augmentation Alignment.** To minimize the effect of false negatives, we introduce a dispersion control term $\mathcal{L}_{\mathrm{DC}}^{-}$, which complements the conventional negative sample loss $\mathcal{L}_{\mathrm{CL}}^{-}$. This leads to the following negative uncertainty control loss:

$$\mathcal{L}_{\mathrm{UCon}}^{-} \triangleq \lambda_{\mathrm{CL}}^{-}\mathcal{L}_{\mathrm{CL}}^{-} + \lambda_{\mathrm{DC}}\mathcal{L}_{\mathrm{DC}}^{-} \triangleq \lambda_{\mathrm{CL}}^{-}\mathcal{L}_{\mathrm{CL}}^{-} + \lambda_{\mathrm{DC}}\Big\{ -\frac{1}{N_{\mathrm{T}}}\sum_{i=1}^{N_{\mathrm{T}}}\mathrm{d}_{\boldsymbol{\theta}}\left(\mathrm{AUG}\left(\mathrm{x}_i\right), \mathrm{x}_i\right)\Big\}, \tag{7}$$

where $\mathrm{d}_{\boldsymbol{\theta}}\left(\mathrm{AUG}\left(\mathrm{x}_i\right), \mathrm{x}_i\right) = \langle f(\mathrm{x}_i;\boldsymbol{\theta}), \log f\left(\mathrm{AUG}\left(\mathrm{x}_i\right);\boldsymbol{\theta}\right)\rangle$, which represents the cosine similarity between the network output of $\mathrm{x}_i$ and the log probabilities of its augmented version $\mathrm{AUG}\left(\mathrm{x}_i\right)$. Often, $\mathrm{AUG}\left(\mathrm{x}_i\right)$ can be obtained through a series of stochastic transformations, including grayscale conversion, slight rotation, posterization, and Gaussian blur, as implemented in self-supervised learning (Chen et al., 2020). Similar to previous work (Yang et al., 2022), the decay coefficient $\lambda_{\mathrm{CL}}^{-}$ is defined as $\lambda_{\mathrm{CL}}^{-} = (1+10\cdot\frac{iter}{max\_iter})^{\beta}$, where $\beta$ and $\lambda_{\mathrm{DC}}$ are hyperparameters, and "iter" and "max_iter" represent the current iteration value and the maximum number of adapting iterations, respectively. $\lambda_{\mathrm{DC}}$ regulates the minimization of the negative similarity between the anchor point and its augmented prediction, which is determined either through tuning hyperparameter or the inconsistency rate of data augmentation (shown in Figure 1b). Further details can be found in Appendices C.5 and D.

Different from previous works that exclude false negative (Chen et al., 2022; Litrico et al., 2023) or adjust the coefficient $\lambda_{\mathrm{CL}}^{-}$ (Mitsuzumi et al., 2024), our proposed dispersion control term utilizes data augmentation to mimic false negatives without introducing additional uncertainty. This approach implicitly reduces the variability in prediction similarity between anchor points and noisy negative samples, while enhancing the model's prediction consistency.

**Supervision Relaxation by Partial Label Training.** As highlighted in Theorem 4.2, partial labels help control uncertainty in positive sample predictions in SFDA. Our experimental findings (reported in Figure 1c) demonstrate that neighboring samples in the feature space can provide accurate label information for initially confident target samples. However, highly uncertain samples require additional processing. To handle these uncertain samples, we propose an innovative approach to select uncertain samples during adaptation by tracking the ratio between the largest and second-largest predicted probabilities. Specifically, we maintain an uncertain data bank, defined as $\mathcal{U} = \Big\{\mathrm{x}\in\mathcal{D}_{\mathrm{T}} : \frac{f(\mathrm{x};\boldsymbol{\theta})_{(1)}}{f(\mathrm{x};\boldsymbol{\theta})_{(2)}} \leq \tau\Big\}$, where $f(\mathrm{x};\boldsymbol{\theta})_{(i)}$ is the $i$th largest predicted probabilities for $\mathrm{x}$. The threshold $\tau$ is typically set to a small value, usually between 1 and 1.5, to capture highly uncertain samples. Additionally, we store the historical TOP-$K_{\mathrm{PL}}$ predicted labels for each data $\mathrm{x}_i$ to construct a partial label set, denoted as $\mathcal{Y}_{\mathrm{PL},i}$, which is then used to further supervise the training of uncertain data. The procedures for determining $\tau$ and $K_{\mathrm{PL}}$ are detailed in Appendices B and D. Aftering incorporating the partial label loss $\mathcal{L}_{\mathrm{PL}}^{+}$, the positive uncertainty control loss term $\mathcal{L}_{\mathrm{UCon}}^{+}$ is defined as:

$$\mathcal{L}_{\mathrm{UCon}}^{+} \triangleq \mathcal{L}_{\mathrm{CL}}^{+} + \lambda_{\mathrm{PL}}\mathcal{L}_{\mathrm{PL}}^{+} \triangleq \mathcal{L}_{\mathrm{CL}}^{+} + \lambda_{\mathrm{PL}}\cdot\frac{1}{N_{\mathrm{T}}}\sum_{i=1}^{N_{\mathrm{T}}}\sum_{\mathrm{y}_{k,i}\in\mathcal{Y}_{\mathrm{PL},i}}\mathbb{1}_{\{\mathrm{x}_i\in\mathcal{U}\}}\ell_{\mathrm{CE}}(\mathrm{y}_{k,i}, f(\mathrm{x}_i;\boldsymbol{\theta})), \tag{8}$$

where $\ell_{\mathrm{CE}}$ is the smoothed cross-entropy loss, and $\lambda_{\mathrm{PL}}$ is a hyperparameter.

Unlike most uncertainty-based approaches in SFDA, which focus on excluding or reducing the negative impact of highly uncertain data during adaptation (Roy et al., 2022; Litrico et al., 2023), our method leverages uncertainty to extract additional label information from these samples, relaxing the training process and boosting the performance.

**Overall Uncertainty Control SFDA Loss.** The final Uncertainty Control SFDA loss, $\mathcal{L}_{\text{UCon}-\text{SFDA}}$, is defined as:

$$\mathcal{L}_{\text{UCon}-\text{SFDA}} = \mathcal{L}_{\text{CL}} + \lambda_{\text{PL}}\mathcal{L}_{\text{PL}}^{+} + \lambda_{\text{DC}}\mathcal{L}_{\text{DC}}^{-}. \tag{9}$$

The pseudocode for the algorithm (Algorithm 1) and the training process can be found in Appendix B.

## 5 EXPERIMENTS

### 5.1 EXPERIMENTAL SETUP

**Datasets.** To evaluate the proposed method, we conduct experiments on several SFDA benchmarks under three different domain shift scenarios: general SFDA, SFDA with severe label shift, and source-free partial set domain adaptation. For general SFDA, we test our method on the following datasets: **Office-31** (Saenko et al., 2010), **Office-Home** (Venkateswara et al., 2017), **VisDA2017** (Peng et al., 2017), and **DomainNet-126** (Litrico et al., 2023). **VisDA2017** is a relatively large-scale classification dataset with 12 classes, consisting of 152K synthetic images and 55K real-world object images. We use the synthetic images as the source domain and the real images as the target domain. **Office-31** contains 4,652 images from three domains (Amazon, DSLR, and Webcam) across 31 categories, while **Office-Home** comprises 15,550 images from four domains (Real, Clipart, Art, and Product) with 65 classes. **DomainNet-126** is a subset of the larger DomainNet dataset that includes over 600K images across 345 categories and six domains (Clipart, Infograph, Painting, Quickdraw, Real, and Sketch) (Peng et al., 2019). Following the setup of previous work (Litrico et al., 2023), we use 126 selected classes from four of these sub-domains for our experiments.

We further evaluate our method on more complex SFDA tasks. For SFDA with label shift, we employ the **VisDA-RUST** dataset, which presents a severe label imbalance in the target domain (Li et al., 2021). For source-free partial set domain adaptation, we follow the setup in Liang et al. (2020) for the **Office-Home** dataset, where only the first 24 classes are retained in the target domain. In addition, we use "→" to indicate the adaptation direction from the source to the target domain.

**Implementation Details.** To ensure fair comparisons, we use the same DNN architectures and training schemes as previous state-of-the-art approaches (Liang et al., 2020; Yang et al., 2022; Hwang et al., 2024). Specifically, we adopt ResNet-50 as the backbone model for the Office-31, Office-Home, and DomainNet-126 datasets, and ResNet-101 for VisDA. We replace the original fully connected layer in ResNet with a bottleneck layer followed by batch normalization, and then add a simple linear layer with weight normalization for the classification. For adaptation training on the target domain, we use the SGD optimizer with the same learning rate scheduler as in Liang et al. (2020). For evaluation, we report the average accuracy for Office-31, Office-Home, and DomainNet-126. For VisDA2017 and VisDA-RUST, we report both per-class top-1 accuracy and the overall average. All experiments are run with three random seeds, and the average results are reported. Further implementation details, including hyperparameter selection, can be found in Appendix B.

### 5.2 OVERALL EXPERIMENTAL RESULTS

The experimental results are summarized in Tables 1- 4 and Table C2 in Appendix C.1, with the best results highlighted in bold. Our proposed method consistently outperforms all baseline methods, especially on the large-scale datasets VisDA2017 (+1.2%) and DomainNet-126 (+1.9%). For VisDA2017, a dataset with only 12 classes, conventional negative sample selection methods that treat the entire batch as negative samples often introduce significant noise and uncertainty. By incorporating the negative sample uncertainty loss, we investigate this issue and see a notable performance boost. Furthermore, our method excels in more challenging tasks, characterized by overall lower accuracy, such as Ar → Cl and Pr → Cl on Office-Home (as shown in Table 3 and Table C2), and it consistently performs well across nearly all tasks on DomainNet-126 (as demonstrated in Table 4).

In more complex scenarios like VisDA-RUST (with severe label imbalance), we observe a performance gain of +2.1%, while for the partial set Office-Home setup, our method shows a +0.6% improvement. These results further confirm the robustness and generality of our proposed method, particularly in handling highly imbalanced target domain data and challenging SFDA tasks.

Additional experimental results and analyses, including self-prediction accuracy, data augmentation consistency, variance control effect, hyperparameter sensitivity, performance under various similarity

Table 1: Classwise accuracy (%) on the VisDA2017 dataset (ResNet-101): synthetic (source) → real (target)

| Method | plane | bcycl | bus | car | horse | knife | mcycl | person | plant | sktbrd | train | truck | Per-class |
|---|---|---|---|---|---|---|---|---|---|---|---|---|---|
| 3C-GAN (Li et al., 2020b) | 94.8 | 73.4 | 68.8 | 74.8 | 93.1 | 95.4 | 88.6 | **84.7** | 89.1 | 84.7 | 83.5 | 48.1 | 81.6 |
| SHOT (Liang et al., 2020) | 94.3 | 88.5 | 80.1 | 57.3 | 93.1 | 94.9 | 80.7 | 80.3 | 91.5 | 89.1 | 86.3 | 58.2 | 82.9 |
| A$^2$Net (Xia et al., 2021) | 94.0 | 87.8 | 85.6 | 66.8 | 93.7 | 95.1 | 85.8 | 81.2 | 91.6 | 88.2 | 86.5 | 56.0 | 84.3 |
| G-SFDA (Yang et al., 2021b) | 96.1 | 83.3 | 85.5 | 74.1 | 97.1 | 95.4 | 89.5 | 79.4 | 95.4 | 92.9 | 89.1 | 42.6 | 85.4 |
| NRC (Yang et al., 2021a) | 96.8 | 91.3 | 82.4 | 62.4 | 96.2 | 95.9 | 86.1 | 80.6 | 94.8 | 94.1 | 90.4 | 59.7 | 85.9 |
| CPGA (Qiu et al., 2021) | 95.6 | 89.0 | 75.4 | 64.9 | 91.7 | **97.5** | 89.7 | 83.8 | 93.9 | 93.4 | 87.7 | **69.0** | 86.0 |
| AdaContrast (Chen et al., 2022) | 97.0 | 84.7 | 84.0 | 77.3 | 96.7 | 93.8 | 91.9 | 84.8 | 94.3 | 93.1 | **94.1** | 47.9 | 86.8 |
| CoWA-JMDS (Lee et al., 2022) | 96.2 | 89.7 | 83.9 | 73.8 | 96.4 | 97.4 | 89.3 | 86.8 | 94.6 | 92.1 | 88.7 | 53.8 | 86.9 |
| DaC (Zhang et al., 2022) | 96.6 | 86.8 | 86.4 | 78.4 | 96.4 | 96.2 | 93.6 | 83.8 | 96.8 | 95.1 | 89.6 | 50.0 | 87.3 |
| AaD (Yang et al., 2022) | 97.4 | 90.5 | 80.8 | 76.2 | 97.3 | 96.1 | 89.8 | 82.9 | 95.5 | 93.0 | 92.0 | 64.7 | 88.0 |
| C-SFDA (Karim et al., 2023) | 97.6 | 88.8 | 86.1 | 72.2 | 97.2 | 94.4 | 92.1 | **84.7** | 93.0 | 90.7 | 93.1 | 63.5 | 87.8 |
| SF(DA)$^2$ (Hwang et al., 2024) | 96.8 | 89.3 | 82.9 | **81.4** | 96.8 | 95.7 | 90.4 | 81.3 | 95.5 | 93.7 | 88.5 | 64.7 | 88.1 |
| I-SFDA (Mitsuzumi et al., 2024) | 97.5 | **91.4** | 87.9 | 79.4 | 97.2 | 97.2 | 92.2 | 83.0 | 96.4 | 94.2 | 91.1 | 53.0 | 88.4 |
| **UCon-SFDA (Ours)** | **98.4** | 90.7 | **88.6** | 80.7 | **97.9** | 96.9 | **93.1** | 83.8 | **97.6** | **95.9** | 92.6 | 59.1 | **89.6** |

Table 2: Classwise accuracy (%) on the VisDA-RSUT dataset (ResNet-101)

| Method | plane | bcycle | bus | car | horse | knife | mcycl | person | plant | sktbrd | train | truck | Per-class |
|---|---|---|---|---|---|---|---|---|---|---|---|---|---|
| Source only (He et al., 2016) | 79.9 | 15.7 | 40.6 | **77.2** | 66.8 | 11.1 | 85.1 | 12.9 | 48.3 | 14.3 | 64.6 | 3.3 | 43.3 |
| SHOT (Liang et al., 2020) | **86.2** | **48.1** | 77.0 | 62.8 | 92.0 | 66.2 | 90.7 | 61.3 | 76.9 | 73.5 | 67.2 | 9.1 | 67.6 |
| CoWA-JMDS (Lee et al., 2022) | 63.8 | 32.9 | 69.5 | 59.9 | 93.2 | 95.4 | 92.3 | 69.4 | 85.1 | 68.4 | 64.9 | 32.3 | 68.9 |
| NRC (Yang et al., 2021a) | **86.2** | 47.6 | 66.7 | 68.1 | 94.7 | 76.6 | 93.7 | 63.6 | 87.3 | 89.0 | 83.6 | 20.0 | 73.1 |
| AaD (Yang et al., 2022) | 73.9 | 33.3 | 56.6 | 71.4 | 90.1 | 97.0 | 91.9 | 70.8 | 88.1 | 87.2 | 81.2 | **39.4** | 73.4 |
| SF(DA)$^2$ (Hwang et al., 2024) | 79.0 | 43.3 | 73.6 | 74.7 | 92.8 | **98.3** | 93.4 | 79.1 | 90.1 | 87.5 | 81.1 | 34.2 | 77.3 |
| **UCon-SFDA (Ours)** | 84.1 | 37.1 | **87.4** | 70.6 | **95.4** | 92.9 | **94.4** | **83.0** | **93.7** | **92.0** | **86.7** | 35.3 | **79.4** |

measures utilized in dispersion control term, and complexity analyses, are provided in Appendix C. A further reduction in the number of hyperparameters within our algorithm, along with two enhanced automatic variants of UCon-SFDA, is detailed in the Appendix D.

Table 3: Classification accuracy (%) on the Office-Home dataset (ResNet-50) under source-free partial-set domain adaptation

| Method | Ar→Cl | Ar→Pr | Ar→Rw | Cl→Ar | Cl→Pr | Cl→Rw | Pr→Ar | Pr→Cl | Pr→Rw | Rw→Ar | Rw→Cl | Rw→Pr | Avg. |
|---|---|---|---|---|---|---|---|---|---|---|---|---|---|
| SHOT (Liang et al., 2020) | 64.8 | 85.2 | 92.7 | 76.3 | 77.6 | **88.8** | 79.7 | 64.3 | 89.5 | 80.6 | 66.4 | 85.8 | 79.3 |
| AaD (Yang et al., 2022) | **67.0** | 83.5 | **93.1** | **80.5** | 76.0 | 87.6 | 78.1 | 65.6 | **90.2** | **83.5** | 64.3 | 87.3 | 79.7 |
| **UCon-SFDA (Ours)** | 65.6 | **87.8** | 91.0 | 78.6 | **79.3** | 87.6 | **80.2** | **65.9** | 87.3 | 83.2 | **69.1** | **88.7** | **80.3** |

## 5.3 ANALYSIS

**Ablation Study.** To evaluate the effectiveness and necessity of each component in our algorithm, we conduct an ablation study across four datasets. The results, shown in Table 5, demonstrate that both partial label supervision training and dispersion control can enhance the performance of the baseline approach ($\mathcal{L}_{\text{CL}}$). While $\mathcal{L}_{\text{PL}}^+$ can better handle severe label shift scenarios, as seen in the VisDA-RUST dataset, $\mathcal{L}_{\text{DC}}^-$ performs better on more difficult tasks. Notably, adding the dispersion control term alone improves or matches the performance of most negative sample denoising and uncertainty-based methods, such as those from Roy et al. (2022); Litrico et al. (2023); Chen et al. (2022); Mitsuzumi et al. (2024), without requiring any additional networks. Combining both positive and negative uncertainty control can boost each other and enhance the performance.

**Negative Sampling Dispersion Control.** To further evaluate the effect of the dispersion control by $\mathcal{L}_{\text{DC}}^-$, we calculate the variance in prediction similarity between anchor-true-negative pairs during adaptation. Figure 3c illustrates that introducing $\mathcal{L}_{\text{DC}}^-$ succesfully reduces this variance. Furthermore, the SF(DA)$^2$ method (Hwang et al., 2024) approaches the problem from a graph-based perspective and introduces a quadratic regularized term on the predicted probability similarity of anchor-negative pairs. This is equivalent to directly minimizing the variance. Our experimental results also demonstrate the effectiveness of our data augmentation-based dispersion control.

**Positive Supervision Uncertainty Relaxation.** As shown in Figure 3a, the top-1 self-predicted label is more accurate for certain data (blue dot line) than for uncertain ones (yellow dot line), indicating that uncertain data require additional supervision during adaptation. To further validate the proposed partial label supervision for uncertain target data, we define a neighbor label set that contains top-1

Table 4: Classification accuracy (%) on Office-31 (left) and DomainNet-126 (right) using ResNet-50

| Method | A → D | A → W | D → W | W → D | D → A | W → A | Avg. |
|---|---|---|---|---|---|---|---|
| SHOT (Liang et al., 2020) | 94.0 | 90.1 | 98.4 | 99.9 | 74.7 | 74.3 | 88.6 |
| 3C-GAN (Li et al., 2020b) | 92.7 | 93.7 | 98.5 | 99.8 | 75.3 | 77.8 | 89.6 |
| A²Net (Xia et al., 2021) | 94.5 | 94.0 | 99.2 | 100.0 | 76.7 | 76.1 | 90.1 |
| NRC (Yang et al., 2021a) | 96.0 | 90.8 | 99.0 | 100.0 | 75.3 | 75.0 | 89.4 |
| CPGA (Qiu et al., 2021) | 94.4 | 94.1 | 98.4 | 99.8 | 76.0 | 76.6 | 89.9 |
| CoWA-JMDS (Lee et al., 2022) | 94.4 | 95.2 | 98.5 | 99.8 | 76.2 | 77.6 | 90.3 |
| AaD (Yang et al., 2022) | 96.4 | 92.1 | 99.1 | 100.0 | 75.0 | 76.5 | 89.9 |
| C-SFDA (Karim et al., 2023) | 96.2 | 93.9 | 98.8 | 99.7 | 77.3 | 77.9 | 90.5 |
| I-SFDA (Mitsuzumi et al., 2024) | 95.3 | 94.2 | 98.3 | 99.9 | 76.4 | 77.5 | 90.3 |
| **UCon-SFDA (Ours)** | 94.8 | 95.4 | 98.9 | 100.0 | 77.1 | 77.1 | 90.6 |

| Method | S→P | C→S | P→C | P→R | R→S | R→C | R→P | Avg. |
|---|---|---|---|---|---|---|---|---|
| Source only (He et al., 2016) | 50.1 | 46.9 | 53.0 | 75.0 | 46.3 | 55.5 | 62.7 | 55.6 |
| TENT (Wang et al., 2020) | 52.4 | 48.5 | 57.9 | 67.0 | 54.0 | 58.5 | 65.7 | 57.7 |
| DivideMix (Li et al., 2020a) | 64.3 | 61.3 | 67.7 | 77.3 | 62.4 | 68.1 | 69.5 | 67.2 |
| SHOT (Liang et al., 2020) | 66.1 | 60.1 | 66.9 | 80.8 | 59.9 | 67.7 | 68.4 | 67.1 |
| NRC (Yang et al., 2021a) | 65.7 | 58.6 | 64.5 | 82.3 | 58.4 | 65.2 | 68.2 | 66.1 |
| AaD (Yang et al., 2022) | 65.4 | 54.2 | 59.8 | 81.8 | 54.6 | 60.3 | 68.5 | 63.5 |
| AdaContrast (Chen et al., 2022) | 65.9 | 58.0 | 68.6 | 80.5 | 61.5 | 70.2 | 69.8 | 67.8 |
| GPUE (Litrico et al., 2023) | 67.5 | 64.0 | 68.8 | 76.5 | 65.7 | 74.2 | 70.4 | 69.6 |
| SF(DA)² (Hwang et al., 2024) | 67.7 | 59.6 | 67.8 | 83.5 | 60.2 | 68.8 | 70.5 | 68.3 |
| **UCon-SFDA (Ours)** | 68.1 | 66.5 | 69.3 | 81.0 | 64.3 | 75.2 | 71.1 | 71.5 |

Table 5: Ablation study results across different datasets and tasks

| Method | VisDA2017 | VisDA-RUST | DomainNet-126 | | | OfficeHome | | |
|---|---|---|---|---|---|---|---|---|
| | Sync → Real | Sync → Real | P → R | R → P | Avg. | Ar → Cl | Pr → Cl | Avg. |
| $\mathcal{L}_{\mathrm{CL}}$ | 87.6 | 75.5 | 78.9 | 67.8 | 66.9 | 58.6 | 57.9 | 72.6 |
| $\mathcal{L}_{\mathrm{CL}} + \mathcal{L}_{\mathrm{DC}}^{-}$ | 89.0 | 78.9 | 80.2 | 70.3 | 69.8 | 61.2 | 59.7 | 73.3 |
| $\mathcal{L}_{\mathrm{CL}} + \mathcal{L}_{\mathrm{PL}}^{+}$ | 88.1 | 79.1 | 80.8 | 69.5 | 68.8 | 60.2 | 59.3 | 73.1 |
| $\mathcal{L}_{\mathrm{UCon-SFDA}}$ | 89.6 | 79.4 | 81.0 | 71.1 | 71.5 | 61.5 | 62.2 | 73.6 |

self-predicted labels of the neighbors. We compare the label information provided by this neighbor label set with our proposed partial label set. By comparing the two lines for neighbor label set accuracy marked with 'x' in Figure 3b, we observe that for uncertain data, the neighbor label set becomes increasingly unstable as training progresses, with accuracy sometimes even decreasing. This explains why we choose not to rely on neighbor labels in our algorithm. Instead, we use the sample's own TOP-$K_{\mathrm{PL}}$ predictions to form a partial label set. A closer look at the difference between the two blue lines and the two yellow lines in Figure 3b reveals that the partial label set provides a greater accuracy gain for uncertain data than for certain data. Interestingly, the accuracy of the neighbor's labels is consistently higher than the overall accuracy of the model's self-prediction, which explains why we apply relaxed supervision through partial label loss only for uncertain data.

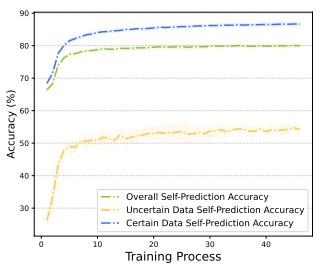

(a) Self-Prediction Accuracies

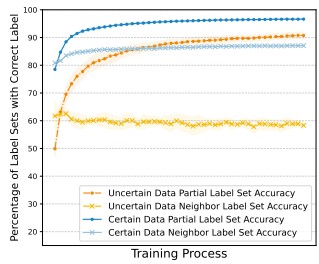

(b) Partial Label Set Quality

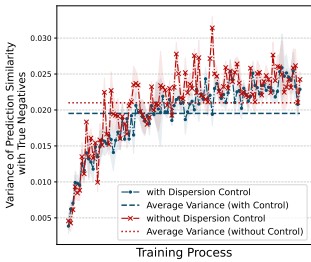

(c) Similarity Dispersion Control

Figure 3: *(a) Self-prediction accuracies across data with varying levels of predictive uncertainty on Office-Home (Ar → Pr). (b) Comparison of the partial label set and neighbor label set quality across different uncertainty levels. (c) Comparison of prediction similarity variances between anchor-true negative pairs, with and without the dispersion control term $\mathcal{L}_{\mathrm{DC}}^{-}$, on Office-Home (Ar → Cl).*

## 6 CONCLUSION

In this paper, we thoroughly analyze two types of uncertainty in SFDA arising from the use of positive and negative samples. By examining the uncertainty in the negative sample distribution during training, we construct an outlier-robust worst-case risk and derive an informative upper bound for it. This analysis not only explains why current contrastive learning methods significantly enhance SFDA performance but also leads to the design of an augmentation-based dispersion control approach to mitigate the uncertainty introduced by noisy negative samples. Furthermore, by investigating the prediction uncertainty of positive examples, we identify a partial label set as the optimal solution for the target data. This insight uncovers previously overlooked uncertain information in existing algorithms and motivates us to propose novel criteria for distinguishing uncertain data, thereby using partial labels to relax the supervision from positive examples.

ACKNOWLEDGEMENTS

Yi is the Canada Research Chair in Data Science (Tier 1). Her research was supported by the Canada Research Chairs Program and the Natural Sciences and Engineering Research Council of Canada (NSERC).

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

## APPENDICES - TECHNICAL DETAILS AND ADDITIONAL EXPERIMENTS

## A    TECHNICAL DETAILS

We introduce key notations for the subsequent subsections. Let $\mathbb{R}_{\geq 0}$ represent the set of all non-negative real values. Given $v = (v_1, \ldots, v_p)^\top$ and $q \in [1, +\infty]$, the $L^q$ norm is defined as $\|v\|_q = \left(\sum_{j=1}^p |v_j|^q\right)^{1/q}$ for $1 \leq q < \infty$, and $\|v\|_\infty = \max_j |v_j|$ when $q = +\infty$. Let $(\Omega, \mathcal{G}, \mu)$ represent a probability measure space, where $\Omega$ is a set, $\mathcal{G}$ is the $\sigma$-algebra of subsets of $\Omega$, and $\mu$ is the associated probability measure. For $q > 0$, let $L^q(\Omega, \mathcal{G}, \mu)$, or simply $L^q(\mu)$, denote the space of Borel-measurable functions $f : \Omega \to \mathbb{R}$ such that $\int |f|^q d\mu < \infty$. For a random variable $Z \sim \mu$, we may interchangeably write the expectation and variance of $f(Z)$ respectively as $\mathbb{E}_\mu\{f(Z)\}$ and $\mathbb{V}_\mu\{f(Z)\}$, $\mathbb{E}_\mu(f)$ and $\mathbb{V}_\mu(f)$, or $\mathbb{E}_{Z \sim \mu}\{f(Z)\}$ and $\mathbb{V}_{Z \sim \mu}\{f(Z)\}$. We use $\mathcal{P}(\Omega)$ to denote the set of Borel probability measures on $\Omega$, and let $\mathcal{P}_p(\Omega)$ represent the subset of $\mathcal{P}(\Omega)$ with finite $q$th moment for $q > 0$. That is, $\mu \in \mathcal{P}_p(\Omega)$ if and only if $\mathbb{E}_{Z \sim \mu}(Z^q) < \infty$. Clearly, $\mathcal{P}_q(\Omega) \subset \mathcal{P}_r(\Omega)$ if $q \geq r$.

### A.1    NOTATION TABLE

The notation table provides a summary for the key notations used throughout the paper, with the symbols, descriptions, and the first appearance place included in the first, second, an the third columns, respectively.

| Notations | Descriptions | First appearance |
|---|---|---|
| $\mathcal{X} \subset \mathbb{R}^d$ | $d$-dimensional input space | Section 3 |
| $\mathcal{Y} = [K]$ | label space for $K$-classification | Section 3 |
| $P_{xy}^{S}$; $\mathcal{D}_{S}$ | underlying distribution over $\mathcal{X} \times \mathcal{Y}$ related to source domain unavailable source domain data $\mathcal{D}_{S} \triangleq \{x_i^{S}, y_i^{S}\}_{i=1}^{N_S}$ | Section 3 |
| $P_{xy}^{T}$; $\mathcal{D}_{T}$ | underlying distribution over $\mathcal{X} \times \mathcal{Y}$ related to target domain unlabeled target domain data $\mathcal{D}_{T} \triangleq \{x_i^{T}\}_{i=1}^{N_T}$ | Section 3 |
| $f_{S}(x; \theta_{S})/f_{T}(x; \theta_{T})/f(x; \theta) :$ $\mathcal{X} \mapsto \Delta^{K-1}$ | predicted probabilities of source/target/general model | Section 3 |
| $h_{S}(x; \theta_{S})/h_{T}(x; \theta_{T})/h(x; \theta) :$ $\mathcal{X} \mapsto \mathcal{Y}$ | source/target/general classifier: $= \arg\max_{j \in [K]} f_{S}(x; \theta_{S})[j]/f_{T}(x; \theta_{T})[j]/f(x; \theta)[j]$ | Section 3 |
| $\mathcal{S}_\theta(x'; x)$ | similarity between $x'$ and $x$ e.g., $\mathcal{S}_\theta(x'; x) = \langle f(x'; \theta), f(x; \theta) \rangle$ | Section 4.1, Eq. (1) |
| $P_x^{T}$ (empirical: $\widehat{P}_x$) | distribution of input X (target) | Section 3 |
| $P^+(\cdot; x)$, or simply $P^+$ (empirical: $\widehat{P}^+$) | conditional distribution for positive sample over $\mathcal{X}$, given x | Section 4.1, Eq. (1) |
| $P^-(\cdot; x)$, or simply $P^-$ (empirical: $\widehat{P}^-$) | conditional distribution for negative sample over $\mathcal{X}$, given x | Section 4.1, Eq. (1) |
| $\mathcal{L}_{CL}^+ / \mathcal{L}_{CL}^-$ | positive/negative contrastive loss | Section 4.3, Remark 4.4 |
| $\mathcal{L}_{PL}^+ / \mathcal{L}_{DC}^-$ | partial label/dispersion control loss | Section 4.3, Remark 4.4 |
| $\mathcal{L}_{UCon}^+ / \mathcal{L}_{UCon}^-$ | overall positive/negative uncertainty control loss | Section 4.4, Eq. (8) |
| $\mathcal{L}_{UCon-SFDA}$ | uncertainty control source-free domain adaptation loss | Section 4.4, Eq. (9) |
| $\lambda_{PL} / \lambda_{DC} / \lambda_{CL}^-$ | partial label/dispersion control/negative contrastive loss coefficient | Section 4.4, Eq. (8) / (7) / (6) |
| $\kappa$ | number of neighbors for each anchor point | Section 4.1 |
| $K_{PL}$ | update number for partial label set | Section 4.4 (Page 7) |
| $\tau$ | uncertain sample selection ratio | Section 4.4 (Page 7) |
| $\beta$ | decay exponent of negative contrastive loss | Section 4.4 (Page 7) |
| $\mathcal{E} / \mathcal{F} / \mathcal{Y}_{PL} / \mathcal{U}$ | feature/predicted probabilities/ partial label set/uncertainty sample bank | Appendix B, Algorithm 1 |
| $\text{AUG}(x)$ | data augmentation of input sample x | Section 4.2, Remark 4.2 |

### A.2    PRELIMINARIES ON DISCREPANCY METRICS AND LINEAR PROGRAMMING

We begin by presenting some definitions and optimization results of the $p$-Wasserstein distance and $\varphi$-divergence, which can be chosen for the discrepancy metric $d$ in (3). These materials will be used in the proof of Theorem 4.1.

**Definition A.1** (*p*-Wasserstein distance (Blanchet & Murthy, 2019)). Let $\Omega$ denote a Polish space (i.e., a complete separable metric space), endowed with a metric $c : \Omega \times \Omega \to \mathbb{R}_{\geq 0}$, also called a cost function. Then, for $p \geq 1$ and any $P, Q \in \mathcal{P}_p(\Omega)$, the Wasserstein distance of order $p$ for $P$ and $Q$ is defined as

$$W_p(P, Q) \triangleq \inf_{\Pi \in \mathrm{Cpl}(P,Q)} \left[ \mathbb{E}_{(S_1, S_2) \sim \Pi} \left\{ c^p(S_1, S_2) \right\} \right]^{1/p},$$

where $\mathrm{Cpl}(P, Q)$, sometimes called the coupling set of $P$ and $Q$, comprises all probability measures on the product space $\Omega \times \Omega$ such that their marginal measures are $P(\cdot)$ and $Q(\cdot)$. Here, $c^p(\cdot, \cdot)$ represents $\{c(\cdot, \cdot)\}^p$.

**Definition A.2** ($\varphi$-divergence (Ali & Silvey, 1966; Duchi, 2019)). Let $P$ and $Q$ be probability distributions on a measurable space $(\Omega, \mathcal{G})$, where $\mathcal{G}$ is the $\sigma$-algebra of subsets of $\Omega$. Let $\varphi : \mathbb{R}_+ \longrightarrow \mathbb{R}$ be a convex function satisfying $\varphi(1) = 0$ and $\varphi(t) = +\infty$ for $t < 0$. Without loss of generality, assume that $P$ and $Q$ are absolutely continuous with respect to the common dominating measure $\mu$. Let $p$ and $q$ denote the density or mass functions of $P$ and $Q$ with respect to the measure $\mu$, respectively; that is, $Q(dx) = q(x)d\mu(x)$ and $P(dx) = p(x)d\mu(x)$. The $\varphi$-divergence between $P$ and $Q$ is then defined as

$$D_\varphi(P\|Q) := \int_\Omega q(x)\varphi\left(\frac{p(x)}{q(x)}\right) d\mu(x) + \varphi'(\infty)P\{q = 0\},$$

where $\varphi'(\infty)$ represents $\lim_{x\to\infty} \varphi(t)/t$.

**Example A.1** (Duchi, 2019, Chapter 2.2). Taking $\varphi$ to be of different forms gives popular examples of $\varphi$-divergences:

- Kullback-Leibler (KL) divergence: taking $\varphi(t) = t \log t$ gives

$$D_\varphi(P\|Q) \triangleq D_{\mathrm{KL}}(P\|Q) = \int p \log(p/q)d\mu.$$

- The total variation distance: taking $\varphi(t) = \frac{1}{2}|t - 1|$ yields

$$D_\varphi(P\|Q) \triangleq \|P - Q\|_{\mathrm{TV}} = \frac{1}{2}\int |\frac{p}{q} - 1|q d\mu = \sup_{A \subset \Omega} |P(A) - Q(A)|.$$

- The Hellinger distance: taking $\varphi(t) = (\sqrt{t} - 1)^2 = t - 2\sqrt{t} + 1$ leads to the squared Hellinger distance

$$D_\varphi(P\|Q) \triangleq H^2(P\|Q) = \int (\sqrt{p} - \sqrt{q})^2 d\mu.$$

- The $\chi^2$-divergence: taking $\varphi(t) = (t - 1)^2$ produces the $\chi^2$-divergence

$$D_\varphi(P\|Q) \triangleq \chi^2(P\|Q) = \int (\frac{p}{q} - 1)^2 d\mu.$$

**Lemma 1** (Strong duality for robust risk based on *p*-Wasserstein distance; Gao et al., 2024, Lemma EC.1). *Consider the p-Wasserstein distance $W_p(\cdot, \cdot)$ with $p \in [1, \infty)$ defined in Definition A.1. Given an upper semi-continuous loss function $\hbar : \Omega \to \mathbb{R}$, a nominal distribution $P \in \mathcal{P}_p(\Omega)$, and a radius $\delta > 0$, the resulting robust risk based on the p-Wasserstein distance $W_p(\cdot, \cdot)$ is defined as*

$$\upsilon_{\mathrm{P}} \triangleq \sup_{Q \in \mathcal{P}(\Omega)} \left[ \mathbb{E}_{Z \sim Q}\{\hbar(Z)\} : W_p(P, Q) \leq \delta \right],$$

*and the dual problem is defined as*

$$\upsilon_{\mathrm{D}} \triangleq \min_{\gamma \geq 0} \left\{ \gamma\delta^p + \mathbb{E}_{Z \sim P}\left[ \sup_{z' \in \Omega} \left\{ \hbar(z') - \gamma c^p(z', Z) \right\} \right] \right\}.$$

*Then, $\upsilon_{\mathrm{P}} = \upsilon_{\mathrm{D}}$.*

**Lemma 2** (Strong duality for robust risk based on $\varphi$-divergence; Duchi & Namkoong, 2021, Proposition 1; Shapiro, 2017, Section 3.2). *Consider the $\varphi$-divergence $D_\varphi(\cdot\|\cdot)$ defined in Definition A.2. Given a loss function $\hbar : \Omega \to \mathbb{R}$, a nominal distribution $P$ on the measure space $(\Omega, \mathcal{G})$, and a radius $\delta > 0$, the resulting robust risk based on the $\varphi$-divergence $D_\varphi(\cdot\|\cdot)$ is defined as*

$$v_{\mathrm{P}} \triangleq \sup_{Q \ll P} \left[ \mathbb{E}_{Z \sim Q}\{\hbar(Z)\} : D_\varphi(Q\|P) \leq \delta \right],$$

*and the dual problem is defined as*

$$v_{\mathrm{D}} \triangleq \inf_{\gamma \geq 0, \eta \in \mathbb{R}} \left\{ \mathbb{E}_P \left[ \gamma \varphi^* \left\{ \frac{\hbar(Z) - \eta}{\gamma} \right\} \right] + \gamma\delta + \eta \right\},$$

*where $\varphi^*(t) = \sup_s\{ts - \varphi(s)\}$ for any $t \in \mathbb{R}$ is the Fenchel conjugate. Then, $v_{\mathrm{P}} = v_{\mathrm{D}}$. Moreover, if the supremum in $v_{\mathrm{P}}$ is finite, then there exist finite $\gamma \geq 0$ and $\eta \in \mathbb{R}$ that attain the infimum in $v_{\mathrm{D}}$.*

**Lemma 3** (Hansen & Sargent, 2008, Proposition 1.4.2). *Let $(\Omega, \mathcal{G}, \mu)$ represent a $\sigma$-finite measure space, where $\Omega$ is a set, $\mathcal{G}$ is the $\sigma$-algebra of subsets of $\Omega$, and $\mu$ is the associated measure. Suppose $\hbar : \Omega \to \mathbb{R}$ is a bounded measurable function. Then the following results hold:*

(i) *The variational formula:*

$$-\log \int_\Omega \exp\{-\hbar(\omega)\}d\mu(\omega) = \inf_{\nu \in \mathcal{P}(\Omega)} \left\{ D_{KL}(\nu\|\mu) + \int_\Omega \hbar(\omega)d\nu(\omega) \right\} \qquad \text{(A1)}$$

(ii) *Suppose $\nu^*$ is the probability measure on $\Omega$ which is absolutely continuous with respect to $\mu$ and satisfies*

$$\frac{d\nu^*}{d\mu}(\omega) \triangleq \frac{\exp\{-\hbar(\omega)\}}{\int_\Omega \exp\{-\hbar(\omega)\}d\mu(\omega)} \quad \text{for } \omega \in \Omega.$$

*Then the infimum in the variational formula (A1) is attained uniquely at $\nu^*$.*

### A.3 PROOF OF THEOREM 4.1

Before presenting and proving the formal version of Theorem 4.1, we first examine the robust risk given in (2) for different choices of the discrepancy metric $d$ in (3), including $\chi^2$-divergence, KL-divergence, and $p$-Wasserstein distance with $p \in [1, +\infty)$. Proof techniques in Duchi & Namkoong (2021); Zhai et al. (2021); Gao (2023); Gao et al. (2024); Lam (2016); Guo et al. (2023) are used.

**Lemma 4.** *For different choices of the discrepancy metric $d$ in (3), we have the following results on the robust risk $\mathcal{R}_{\mathrm{x}}^-(\boldsymbol{\theta}; P^-, \delta)$ given in (2):*

(i) *If $d$ is the $\chi^2$-divergence and $\delta \leq \mathbb{V}_P\text{-}\{\mathcal{S}_{\boldsymbol{\theta}}(\mathrm{X}^-; \mathrm{x})\} / \left[\mathbb{E}_P\text{-}\{\mathcal{S}_{\boldsymbol{\theta}}(\mathrm{X}^-; \mathrm{x})\}\right]^2$, then*

$$\mathcal{R}_{\mathrm{x}}^-(\boldsymbol{\theta}; P^-, \delta) = \mathbb{E}_P\text{-}\{\mathcal{S}_{\boldsymbol{\theta}}(\mathrm{X}^-; \mathrm{x})\} + \sqrt{\delta\mathbb{V}_P\text{-}\{\mathcal{S}_{\boldsymbol{\theta}}(\mathrm{X}^-; \mathrm{x})\}}.$$

(ii) *If $d$ is the KL-divergence, then,*

$$\mathcal{R}_{\mathrm{x}}^-(\boldsymbol{\theta}; P^-, \delta) = \mathbb{E}_P\text{-}\{\mathcal{S}_{\boldsymbol{\theta}}(\mathrm{X}^-; \mathrm{x})\} + \sqrt{2\delta\mathbb{V}_P\text{-}\{\mathcal{S}_{\boldsymbol{\theta}}(\mathrm{X}^-; \mathrm{x})\}} + \mathcal{O}(\delta).$$

*with $\delta > 0$ being small.*

(iii) *Suppose $d$ is the $p$-Wasserstein distance with $p \in [1, +\infty)$ and the cost function $c(\cdot, \cdot)$ in Definition A.1 is chosen as a norm $\|\cdot\|$ with the dual norm $\|\cdot\|_*$. Assume the following smoothness conditions:*

     a. *For any $\widetilde{\mathrm{x}}^-, \mathrm{x}^-, \mathrm{x} \in \mathcal{X}$, $\exists \mathcal{M}_1, \mathcal{M}_2 > 0$ and $\zeta \in [1, p]$, such that*

$$\|\nabla\mathcal{S}_{\boldsymbol{\theta}}(\widetilde{\mathrm{x}}^-; \mathrm{x}) - \nabla\mathcal{S}_{\boldsymbol{\theta}}(\mathrm{x}^-; \mathrm{x})\|_* \leq \mathcal{M}_1 + \mathcal{M}_2\|\widetilde{\mathrm{x}}^- - \mathrm{x}^-\|^{\zeta-1}.$$

     b. *There exists $\eta_0 > 0$ and $\mathcal{M}_3 > 0$, such that for any $\widetilde{\mathrm{x}}^-, \mathrm{x}^-, \mathrm{x} \in \mathcal{X}$, if $\|\widetilde{\mathrm{x}}^- - \mathrm{x}^-\| \leq \eta_0$, then $\|\nabla\mathcal{S}_{\boldsymbol{\theta}}(\widetilde{\mathrm{x}}^-; \mathrm{x}) - \nabla\mathcal{S}_{\boldsymbol{\theta}}(\mathrm{x}^-; \mathrm{x})\|_* \leq \mathcal{M}_3\|\widetilde{\mathrm{x}}^- - \mathrm{x}^-\|$.*

*Let $q$ denote the Hölder number of $p$, that is $\frac{1}{p} + \frac{1}{q} = 1$. Then*

$$\mathcal{R}_{\mathrm{x}}^-(\boldsymbol{\theta}; P^-, \delta) \leq \mathbb{E}_P\text{-}\{\mathcal{S}_{\boldsymbol{\theta}}(\mathrm{X}^-; \mathrm{x})\} + \delta\left\{\mathbb{E}_P\text{-}\|\nabla\mathcal{S}_{\boldsymbol{\theta}}(\mathrm{X}^-; \mathrm{x})\|_*^q\right\}^{1/q} + \mathcal{O}(\delta^{2\wedge p}).$$

**Proof of (i) with $d$ set as the $\chi^2$-divergence:**

For the $\chi^2$-divergence, we have $\varphi(t) = (t-1)^2$ for $t \geq 0$ and $\varphi(t) = +\infty$ for $t < 0$ by Example A.1. Then the Fenchel conjugate of $\varphi$ is given by

$$
\varphi^*(t) = \sup_{s \in \mathbb{R}} \left\{ ts - \varphi(s) \right\} = \sup_{s \geq 0} \left\{ ts - (s-1)^2 \right\} = \sup_{s \geq 0} \left\{ -\left( s - \frac{t+2}{2} \right)^2 + \frac{t^2}{4} + t \right\}
$$

$$
= \begin{cases} \dfrac{t^2}{4} + t, & \text{for } t \geq -2 \\ -1, & \text{for } t < -2 \end{cases} \quad = \quad \frac{1}{4} \{ (t+2)_+ \}^2 - 1. \tag{A2}
$$

**Step (i): Upper bound on the primal problem.**

If the discrepancy metric $d$ in (3) is chosen as the $\chi^2$-divergence, then the robust risk $\mathcal{R}_{\mathsf{x}}^{\text{-}}(\boldsymbol{\theta}; P^{\text{-}}, \delta)$ is expressed as

$$
\mathcal{R}_{\mathsf{x}}^{\text{-}}(\boldsymbol{\theta}; P^{\text{-}}, \delta) = \sup_{Q^{\text{-}} \ll P^{\text{-}}} \left[ \mathbb{E}_{Q^{\text{-}}} \left\{ \mathcal{S}_{\boldsymbol{\theta}}(\mathrm{X}^{\text{-}}; \mathrm{x}) \right\} : \chi^2(Q^{\text{-}} \| P^{\text{-}}) \leq \delta \right]. \tag{A3}
$$

The expectation $\mathbb{E}_{Q^{\text{-}}} \left\{ \mathcal{S}_{\boldsymbol{\theta}}(\mathrm{X}^{\text{-}}; \mathrm{x}) \right\}$ in (A3) can be expressed as:

$$
\mathbb{E}_{Q^{\text{-}}} \left\{ \mathcal{S}_{\boldsymbol{\theta}}(\mathrm{X}^{\text{-}}; \mathrm{x}) \right\} = \mathbb{E}_{P^{\text{-}}} \left\{ \mathcal{S}_{\boldsymbol{\theta}}(\mathrm{X}^{\text{-}}; \mathrm{x}) \frac{dQ^{\text{-}}}{dP^{\text{-}}} \right\}
$$

$$
= \mathbb{E}_{P^{\text{-}}} \left\{ \mathcal{S}_{\boldsymbol{\theta}}(\mathrm{X}^{\text{-}}; \mathrm{x}) \right\} + \mathbb{E}_{P^{\text{-}}} \left\{ \mathcal{S}_{\boldsymbol{\theta}}(\mathrm{X}^{\text{-}}; \mathrm{x}) \left( \frac{dQ^{\text{-}}}{dP^{\text{-}}} - 1 \right) \right\}
$$

$$
= \mathbb{E}_{P^{\text{-}}} \left\{ \mathcal{S}_{\boldsymbol{\theta}}(\mathrm{X}^{\text{-}}; \mathrm{x}) \right\} + \mathbb{E}_{P^{\text{-}}} \left\{ \left[ \mathcal{S}_{\boldsymbol{\theta}}(\mathrm{X}^{\text{-}}; \mathrm{x}) - \mathbb{E}_{P^{\text{-}}} \left\{ \mathcal{S}_{\boldsymbol{\theta}}(\mathrm{X}^{\text{-}}; \mathrm{x}) \right\} \right] \left( \frac{dQ^{\text{-}}}{dP^{\text{-}}} - 1 \right) \right\},
$$

where the first inequality holds via a change of measure and the fact that $Q^{\text{-}} \ll P^{\text{-}}$, $\frac{dQ^{\text{-}}}{dP^{\text{-}}}$ denotes the Radon–Nikodym derivative, and the last equality is true since $\mathbb{E}_{P^{\text{-}}} \left( \frac{dQ^{\text{-}}}{dP^{\text{-}}} - 1 \right) = 0$.

By the Cauchy-Schwarz inequality, we further obtain that

$$
\mathcal{R}_{\mathsf{x}}^{\text{-}}(\boldsymbol{\theta}; P^{\text{-}}, \delta) - \mathbb{E}_{P^{\text{-}}} \left\{ \mathcal{S}_{\boldsymbol{\theta}}(\mathrm{X}^{\text{-}}; \mathrm{x}) \right\}
$$

$$
= \sqrt{ \left\{ \mathbb{E}_{P^{\text{-}}} \left[ \mathcal{S}_{\boldsymbol{\theta}}(\mathrm{X}^{\text{-}}; \mathrm{x}) - \mathbb{E}_{P^{\text{-}}} \left\{ \mathcal{S}_{\boldsymbol{\theta}}(\mathrm{X}^{\text{-}}; \mathrm{x}) \right\} \right]^2 \right\} \cdot \left\{ \mathbb{E}_{P^{\text{-}}} \left( \frac{dQ^{\text{-}}}{dP^{\text{-}}} - 1 \right)^2 \right\} }
$$

$$
= \sqrt{ \left\{ \mathbb{E}_{P^{\text{-}}} \left[ \mathcal{S}_{\boldsymbol{\theta}}(\mathrm{X}^{\text{-}}; \mathrm{x}) - \mathbb{E}_{P^{\text{-}}} \left\{ \mathcal{S}_{\boldsymbol{\theta}}(\mathrm{X}^{\text{-}}; \mathrm{x}) \right\} \right]^2 \right\} \cdot \chi^2(Q^{\text{-}} \| P^{\text{-}}) }
$$

$$
\leq \sqrt{ \left\{ \mathbb{E}_{P^{\text{-}}} \left[ \mathcal{S}_{\boldsymbol{\theta}}(\mathrm{X}^{\text{-}}; \mathrm{x}) - \mathbb{E}_{P^{\text{-}}} \left\{ \mathcal{S}_{\boldsymbol{\theta}}(\mathrm{X}^{\text{-}}; \mathrm{x}) \right\} \right]^2 \right\} \cdot \delta },
$$

where the second equality is due to the definition of $\chi^2$-divergence given in Example A.1, and the last step is due to the constraint in (A3). Therefore, by (A3), we obtain that

$$
\mathcal{R}_{\mathsf{x}}^{\text{-}}(\boldsymbol{\theta}; P^{\text{-}}, \delta) \leq \mathbb{E}_{P^{\text{-}}} \left\{ \mathcal{S}_{\boldsymbol{\theta}}(\mathrm{X}^{\text{-}}; \mathrm{x}) \right\} + \sqrt{ \left\{ \mathbb{E}_{P^{\text{-}}} \left[ \mathcal{S}_{\boldsymbol{\theta}}(\mathrm{X}^{\text{-}}; \mathrm{x}) - \mathbb{E}_{P^{\text{-}}} \left\{ \mathcal{S}_{\boldsymbol{\theta}}(\mathrm{X}^{\text{-}}; \mathrm{x}) \right\} \right]^2 \right\} \cdot \delta }
$$

$$
\triangleq \mu + \sqrt{\delta V}, \tag{A4}
$$

where $\mu \triangleq \mathbb{E}_{P^{\text{-}}} \left\{ \mathcal{S}_{\boldsymbol{\theta}}(\mathrm{X}^{\text{-}}; \mathrm{x}) \right\}$ and $V \triangleq \mathbb{E}_{P^{\text{-}}} \left[ \mathcal{S}_{\boldsymbol{\theta}}(\mathrm{X}^{\text{-}}; \mathrm{x}) - \mathbb{E}_{P^{\text{-}}} \left\{ \mathcal{S}_{\boldsymbol{\theta}}(\mathrm{X}^{\text{-}}; \mathrm{x}) \right\} \right]^2$.

**Step (ii): Attaining the equality in the upper bound using duality.**

Next, we prove that the equality in the upper bound in (A4) can be achieved by leveraging the strong duality result of the $\varphi$-divergence based robust risk. Specifically, according to Lemma 2 and (A2),

$$
\begin{aligned}
\mathcal{R}_{\mathrm{x}}^{-}(\boldsymbol{\theta}; P^{-}, \delta) &= \inf_{\gamma \geq 0, \eta \in \mathbb{R}} \left\{ \mathbb{E}_P \left[ \gamma \varphi^* \left\{ \frac{\mathcal{S}_{\boldsymbol{\theta}}(\mathrm{X}^{-}; \mathrm{x}) - \eta}{\gamma} \right\} \right] + \gamma \delta + \eta \right\} \\
&= \inf_{\gamma \geq 0, \eta \in \mathbb{R}} \left\{ \mathbb{E}_P \left[ \gamma \cdot \frac{1}{4} \left\{ \frac{\mathcal{S}_{\boldsymbol{\theta}}(\mathrm{X}^{-}; \mathrm{x}) - \eta}{\gamma} + 2 \right\}_+^2 - \gamma \right] + \gamma \delta + \eta \right\} \\
&= \inf_{\gamma \geq 0, \eta \in \mathbb{R}} \left[ \frac{1}{4\gamma} \mathbb{E}_P \left\{ \mathcal{S}_{\boldsymbol{\theta}}(\mathrm{X}^{-}; \mathrm{x}) - \eta + 2\gamma \right\}_+^2 - \gamma + \gamma \delta + \eta \right] \\
&= \inf_{\gamma \geq 0, \widetilde{\eta} \in \mathbb{R}} \left[ \frac{1}{4\gamma} \mathbb{E}_P \left\{ \mathcal{S}_{\boldsymbol{\theta}}(\mathrm{X}^{-}; \mathrm{x}) - \widetilde{\eta} \right\}_+^2 + (1 + \delta)\gamma + \widetilde{\eta} \right] \\
&\triangleq \inf_{\gamma \geq 0, \widetilde{\eta} \in \mathbb{R}} \psi(\gamma; \widetilde{\eta}),
\end{aligned}
$$

where the second last equality holds by taking $\widetilde{\eta} \triangleq \eta - 2\gamma$.

We now examine the minimum of $\psi(\gamma; \widetilde{\eta})$ by fixing one argument. First, given $\widetilde{\eta}$, taking derivatives of $\psi(\gamma; \widetilde{\eta})$ with respect to $\gamma$ gives that the optimal $\gamma$ to infimize the preceding expression is given by:

$$
\gamma^* = \sqrt{\frac{\mathbb{E}_P \left\{ \mathcal{S}_{\boldsymbol{\theta}}(\mathrm{X}^{-}; \mathrm{x}) - \widetilde{\eta} \right\}_+^2}{4(1 + \delta)}}.
$$

Then substituting $\gamma^*$ into $\psi(\gamma; \widetilde{\eta})$ gives

$$
\mathcal{R}_{\mathrm{x}}^{-}(\boldsymbol{\theta}; P^{-}, \delta) = \inf_{\widetilde{\eta} \in \mathbb{R}} \left[ \sqrt{(1 + \delta)\mathbb{E}_P \left\{ \mathcal{S}_{\boldsymbol{\theta}}(\mathrm{X}^{-}; \mathrm{x}) - \widetilde{\eta} \right\}_+^2} + \widetilde{\eta} \right]. \tag{A5}
$$

Next, $g(\widetilde{\eta}) \triangleq \sqrt{(1 + \delta)\mathbb{E}_P \left\{ \mathcal{S}_{\boldsymbol{\theta}}(\mathrm{X}^{-}; \mathrm{x}) - \widetilde{\eta} \right\}_+^2} + \widetilde{\eta}$. By taking

$$
\widetilde{\eta}^* = \mu - \sqrt{\frac{V}{\delta}}, \tag{A6}
$$

where $\mu$ and $V$ are defined after (A4), we obtain that

$$
\begin{aligned}
g(\widetilde{\eta}^*) &= \sqrt{(1 + \delta)\mathbb{E}_P \left\{ \mathcal{S}_{\boldsymbol{\theta}}(\mathrm{X}^{-}; \mathrm{x}) - \widetilde{\eta}^* \right\}_+^2} + \widetilde{\eta}^* \\
&= \sqrt{(1 + \delta)\mathbb{E}_P \left\{ \mathcal{S}_{\boldsymbol{\theta}}(\mathrm{X}^{-}; \mathrm{x}) - \widetilde{\eta}^* \right\}^2} + \widetilde{\eta}^* \\
&= \sqrt{(1 + \delta)\mathbb{E}_P \left\{ \mathcal{S}_{\boldsymbol{\theta}}(\mathrm{X}^{-}; \mathrm{x}) - \mu + \sqrt{\frac{V}{\delta}} \right\}^2} + \mu - \sqrt{\frac{V}{\delta}} \\
&= \sqrt{(1 + \delta)\left[ \mathbb{E}_P \left\{ \mathcal{S}_{\boldsymbol{\theta}}(\mathrm{X}^{-}; \mathrm{x}) - \mu \right\}^2 + \frac{V}{\delta} + 2\sqrt{\frac{V}{\delta}}\mathbb{E}_P \left\{ \mathcal{S}_{\boldsymbol{\theta}}(\mathrm{X}^{-}; \mathrm{x}) - \mu \right\} \right]} + \mu - \sqrt{\frac{V}{\delta}} \\
&= \sqrt{(1 + \delta)\left( V + \frac{V}{\delta} \right)} + \mu - \sqrt{\frac{V}{\delta}} \\
&= \mu + \sqrt{\delta V},
\end{aligned}
$$

where the first step holds since $\widetilde{\eta}^* = \mu - \sqrt{\frac{V}{\delta}} < 0$, and the fifth step is due to the definitions of $\mu$ and $V$.

**Step (iii): Mean-dispersion form of the robust risk.**

With $\widetilde{\eta}^* = \mu - \sqrt{\frac{V}{\delta}}$ in (A6), the dual objective (A5) in its infimum form achieves the equality in (A4), which is the upper bound of the primal problem (A3) in its supremum form. Consequently, we obtain that

$$\mathcal{R}_{\mathrm{x}}^{\text{-}}(\boldsymbol{\theta}; P^{\text{-}}, \delta) = \mathbb{E}_{P^{\text{-}}}\{\mathcal{S}_{\boldsymbol{\theta}}(\mathrm{X}^{\text{-}}; \mathrm{x})\} + \sqrt{\left\{\mathbb{E}_{P^{\text{-}}}\left[\mathcal{S}_{\boldsymbol{\theta}}(\mathrm{X}^{\text{-}}; \mathrm{x}) - \mathbb{E}_{P^{\text{-}}}\{\mathcal{S}_{\boldsymbol{\theta}}(\mathrm{X}^{\text{-}}; \mathrm{x})\}\right]^2\right\} \cdot \delta}.$$

The proof is completed.

**Proof of (ii) with $d$ set as the KL-divergence:**

If the discrepancy metric $d$ in (3) is chosen as the KL-divergence, then the robust risk $\mathcal{R}_{\mathrm{x}}^{\text{-}}(\boldsymbol{\theta}; P^{\text{-}}, \delta)$ is expressed as

$$\mathcal{R}_{\mathrm{x}}^{\text{-}}(\boldsymbol{\theta}; P^{\text{-}}, \delta) = \sup_{Q^{\text{-}} \ll P^{\text{-}}} \left[\mathbb{E}_{Q^{\text{-}}}\{\mathcal{S}_{\boldsymbol{\theta}}(\mathrm{X}^{\text{-}}; \mathrm{x})\} : D_{\mathrm{KL}}(Q^{\text{-}}\|P^{\text{-}}) \leq \delta\right]$$

$$= \sup_{Q^{\text{-}} \ll P^{\text{-}}} \left[\mathbb{E}_{Q^{\text{-}}}\{\mathcal{S}_{\boldsymbol{\theta}}(\mathrm{X}^{\text{-}}; \mathrm{x})\} : \mathbb{E}_{Q^{\text{-}}}\left\{\log\left(\frac{dQ^{\text{-}}}{dP^{\text{-}}}\right)\right\} \leq \delta\right]. \qquad (A7)$$

By a change of measure and denoting the likelihood ratio $\mathsf{L}(\omega) \triangleq \frac{dQ^{\text{-}}(\omega)}{dP^{\text{-}}(\omega)}$ for $\omega \in \mathcal{X}$, the objective and the constraint in (A7) can be expressed as

$$\mathbb{E}_{Q^{\text{-}}}\left\{\mathcal{S}_{\boldsymbol{\theta}}(\mathrm{X}^{\text{-}}; \mathrm{x})\right\} = \mathbb{E}_{P^{\text{-}}}\left\{\mathcal{S}_{\boldsymbol{\theta}}(\mathrm{X}^{\text{-}}; \mathrm{x})\frac{dQ^{\text{-}}}{dP^{\text{-}}}\right\} \triangleq \mathbb{E}_{P^{\text{-}}}\left\{\mathcal{S}_{\boldsymbol{\theta}}(\mathrm{X}^{\text{-}}; \mathrm{x})\mathsf{L}(\mathrm{X}^{\text{-}})\right\};$$

$$\mathbb{E}_{Q^{\text{-}}}\left\{\log\left(\frac{dQ^{\text{-}}}{dP^{\text{-}}}\right)\right\} = \mathbb{E}_{P^{\text{-}}}\left[\left\{\log\left(\frac{dQ^{\text{-}}}{dP^{\text{-}}}\right)\right\}\frac{dQ^{\text{-}}}{dP^{\text{-}}}\right] = \mathbb{E}_{P^{\text{-}}}\left[\mathsf{L}(\mathrm{X}^{\text{-}})\log\{\mathsf{L}(\mathrm{X}^{\text{-}})\}\right].$$

Therefore, the expression of the robust risk $\mathcal{R}_{\mathrm{x}}^{\text{-}}(\boldsymbol{\theta}; P^{\text{-}}, \delta)$ can be rewritten as:

$$\mathcal{R}_{\mathrm{x}}^{\text{-}}(\boldsymbol{\theta}; P^{\text{-}}, \delta) = \begin{cases} \max_{\mathsf{L} \in \mathcal{L}} \mathbb{E}_{P^{\text{-}}}\left\{\mathcal{S}_{\boldsymbol{\theta}}(\mathrm{X}^{\text{-}}; \mathrm{x})\mathsf{L}(\mathrm{X}^{\text{-}})\right\} \\ s.t. \ \mathbb{E}_{P^{\text{-}}}\left[\mathsf{L}(\mathrm{X}^{\text{-}})\log\{\mathsf{L}(\mathrm{X}^{\text{-}})\}\right] \leq \delta, \end{cases} \qquad (A8)$$

where $\mathcal{L} = \{\mathsf{L} \in L^1(P^{\text{-}}) : \mathbb{E}_{P^{\text{-}}}\{\mathsf{L}(\mathrm{X}^{\text{-}})\} = 1; \mathsf{L} \geq 0 \ a.s.\}$.

Since (A8) is a convex optimization problem with respect to $\mathsf{L}$, by introducing the Lagrange multiplier $\gamma > 0$, it can be further expressed as:

$$\mathcal{R}_{\mathrm{x}}^{\text{-}}(\boldsymbol{\theta}; P^{\text{-}}, \delta) = \max_{\mathsf{L} \in \mathcal{L}, \gamma \geq 0} \mathbb{E}_{P^{\text{-}}}\left\{\mathcal{S}_{\boldsymbol{\theta}}(\mathrm{X}^{\text{-}}; \mathrm{x})\mathsf{L}(\mathrm{X}^{\text{-}})\right\} - \gamma\left\{\mathbb{E}_{P^{\text{-}}}\left[\mathsf{L}(\mathrm{X}^{\text{-}})\log\{\mathsf{L}(\mathrm{X}^{\text{-}})\}\right] - \delta\right\}. \quad (A9)$$

**Step (i): Optimal form of the likelihood ratio $\mathsf{L}^*$.**

Suppose we can find $\gamma^* \geq 0$ and $\mathsf{L}^* \in \mathcal{L}$ such that $\mathsf{L}^*$ maximizes (A9) for a fixed $\gamma = \gamma^*$ and $\mathbb{E}_{P^{\text{-}}}\left[\mathsf{L}(\mathrm{X}^{\text{-}})\log\{\mathsf{L}(\mathrm{X}^{\text{-}})\}\right] = \delta$. Then, for any $\mathsf{L} \in \mathcal{L}$ satisfying the constraint $\mathbb{E}_{P^{\text{-}}}\left[\mathsf{L}(\mathrm{X}^{\text{-}})\log\{\mathsf{L}(\mathrm{X}^{\text{-}})\}\right] \leq \delta$ in (A8), we have that

$$\mathbb{E}_{P^{\text{-}}}\left\{\mathcal{S}_{\boldsymbol{\theta}}(\mathrm{X}^{\text{-}}; \mathrm{x})\mathsf{L}^*(\mathrm{X}^{\text{-}})\right\}$$
$$= \mathbb{E}_{P^{\text{-}}}\left\{\mathcal{S}_{\boldsymbol{\theta}}(\mathrm{X}^{\text{-}}; \mathrm{x})\mathsf{L}^*(\mathrm{X}^{\text{-}})\right\} - \gamma^*\left\{\mathbb{E}_{P^{\text{-}}}\left[\mathsf{L}^*(\mathrm{X}^{\text{-}})\log\{\mathsf{L}^*(\mathrm{X}^{\text{-}})\}\right] - \delta\right\}$$
$$\geq \mathbb{E}_{P^{\text{-}}}\left\{\mathcal{S}_{\boldsymbol{\theta}}(\mathrm{X}^{\text{-}}; \mathrm{x})\mathsf{L}(\mathrm{X}^{\text{-}})\right\} - \gamma^*\left\{\mathbb{E}_{P^{\text{-}}}\left[\mathsf{L}(\mathrm{X}^{\text{-}})\log\{\mathsf{L}(\mathrm{X}^{\text{-}})\}\right] - \delta\right\}$$
$$\geq \mathbb{E}_{P^{\text{-}}}\left\{\mathcal{S}_{\boldsymbol{\theta}}(\mathrm{X}^{\text{-}}; \mathrm{x})\mathsf{L}(\mathrm{X}^{\text{-}})\right\},$$

and hence, $\mathsf{L}^*$ is the optimal solution of (A8).

We first assume the existence of such $\gamma^* \geq 0$ and consider the form of the corresponding $\mathsf{L}^*$. Let $\mathsf{f}(\mathsf{L}; \gamma) \triangleq \mathbb{E}_{P^{\text{-}}}\left\{\mathcal{S}_{\boldsymbol{\theta}}(\mathrm{X}^{\text{-}}; \mathrm{x})\mathsf{L}(\mathrm{X}^{\text{-}})\right\} - \gamma\left\{\mathbb{E}_{P^{\text{-}}}\left[\mathsf{L}(\mathrm{X}^{\text{-}})\log\{\mathsf{L}(\mathrm{X}^{\text{-}})\}\right] - \delta\right\}$ denote the objective

function in (A9). For a fixed $\gamma^* \in \mathbb{R}$, we consider the form of $\mathsf{L}^* \in \mathrm{argmax}_{\mathsf{L} \in \mathcal{L}}\, \mathsf{f}(\mathsf{L}; \gamma^*)$, which can be expressed as

$$
\begin{aligned}
&\mathsf{L}^* \in \underset{\mathsf{L} \in \mathcal{L}}{\mathrm{argmax}}\; \mathbb{E}_{P^-}\Big\{ \mathcal{S}_{\boldsymbol{\theta}}(\mathrm{X}^-; \mathrm{x}) \mathsf{L}(\mathrm{X}^-) \Big\} - \gamma^* \Big\{ \mathbb{E}_{P^-}\big[ \mathsf{L}(\mathrm{X}^-) \log\{\mathsf{L}(\mathrm{X}^-)\} \big] - \delta \Big\}\\
&\Leftrightarrow \mathsf{L}^* \in \underset{\mathsf{L} \in \mathcal{L}}{\mathrm{argmax}}\; -\gamma^* \Big( \mathbb{E}_{P^-}\Big\{ -\mathcal{S}_{\boldsymbol{\theta}}(\mathrm{X}^-; \mathrm{x}) \mathsf{L}(\mathrm{X}^-)/\gamma^* \Big\} + \mathbb{E}_{P^-}\big[ \mathsf{L}(\mathrm{X}^-) \log\{\mathsf{L}(\mathrm{X}^-)\} \big] \Big)\\
&\Leftrightarrow \mathsf{L}^* dP^- \in \underset{Q^- \in \mathcal{P}_p(\mathcal{X})}{\mathrm{argmin}}\; \mathbb{E}_{Q^-}\Big\{ -\mathcal{S}_{\boldsymbol{\theta}}(\mathrm{X}^-; \mathrm{x})/\gamma^* \Big\} + D_{\mathrm{KL}}(Q^- \| P^-) \Big].
\end{aligned}
$$

By Lemma 3, we obtain that

$$
\mathsf{L}^*(\mathrm{X}^-) = \exp\left\{ \frac{\mathcal{S}_{\boldsymbol{\theta}}(\mathrm{X}^-; \mathrm{x})}{\gamma^*} \right\} \Big/ \mathbb{E}_{P^-}\left[ \exp\left\{ \frac{\mathcal{S}_{\boldsymbol{\theta}}(\mathrm{X}^-; \mathrm{x})}{\gamma^*} \right\} \right]. \tag{A10}
$$

is the unique optimal solution of $\mathsf{L}^* \in \mathrm{argmax}_{\mathsf{L} \in \mathcal{L}}\, \mathsf{f}(\mathsf{L}; \gamma^*)$ for a fixed $\gamma^*$ since the similarity measure $\mathcal{S}_{\boldsymbol{\theta}}$ is a bounded function.

### Step (ii): Existence of $\gamma^*$.

If the $\gamma^*$ in Step (i) exists, then the optimal $\mathsf{L}^*$ is given in (A10), and the constraint and objective in (A8) can be expressed as below:

$$
\begin{aligned}
\delta &= \mathbb{E}_{P^-}\Big[ \mathsf{L}^*(\mathrm{X}^-) \log\{\mathsf{L}^*(\mathrm{X}^-)\} \Big]\\
&= \mathbb{E}_{P^-}\left( \frac{\exp\{\mathcal{S}_{\boldsymbol{\theta}}(\mathrm{X}^-; \mathrm{x})/\gamma^*\}}{\mathbb{E}_{P^-}[\exp\{\mathcal{S}_{\boldsymbol{\theta}}(\mathrm{X}^-; \mathrm{x})/\gamma^*\}]} \cdot \left\{ \frac{\mathcal{S}_{\boldsymbol{\theta}}(\mathrm{X}^-; \mathrm{x})}{\gamma^*} - \log \mathbb{E}_{P^-}\left[ \exp\left\{ \frac{\mathcal{S}_{\boldsymbol{\theta}}(\mathrm{X}^-; \mathrm{x})}{\gamma^*} \right\} \right] \right\} \right)\\
&= \frac{1}{\gamma^*} \cdot \frac{\mathbb{E}_{P^-}[\mathcal{S}_{\boldsymbol{\theta}}(\mathrm{X}^-; \mathrm{x}) \cdot \exp\{\mathcal{S}_{\boldsymbol{\theta}}(\mathrm{X}^-; \mathrm{x})/\gamma^*\}]}{\mathbb{E}_{P^-}[\exp\{\mathcal{S}_{\boldsymbol{\theta}}(\mathrm{X}^-; \mathrm{x})/\gamma^*\}]} - \log \mathbb{E}_{P^-}\left[ \exp\left\{ \frac{\mathcal{S}_{\boldsymbol{\theta}}(\mathrm{X}^-; \mathrm{x})}{\gamma^*} \right\} \right]\\
&= \bar{\varrho} \cdot \frac{\mathbb{E}_{P^-}[\mathcal{S}_{\boldsymbol{\theta}}(\mathrm{X}^-; \mathrm{x}) \cdot \exp\{\bar{\varrho} \cdot \mathcal{S}_{\boldsymbol{\theta}}(\mathrm{X}^-; \mathrm{x})\}]}{\mathbb{E}_{P^-}[\exp\{\bar{\varrho} \cdot \mathcal{S}_{\boldsymbol{\theta}}(\mathrm{X}^-; \mathrm{x})\}]} - \log \mathbb{E}_{P^-}\Big[ \exp\{\bar{\varrho} \cdot \mathcal{S}_{\boldsymbol{\theta}}(\mathrm{X}^-; \mathrm{x})\} \Big]\\
&\triangleq \bar{\varrho} \hbar'(\bar{\varrho}) - \hbar(\bar{\varrho}). \tag{A11}
\end{aligned}
$$

In addition,

$$
\begin{aligned}
\mathbb{E}_{P^-}\Big\{ \mathcal{S}_{\boldsymbol{\theta}}(\mathrm{X}^-; \mathrm{x}) \mathsf{L}^*(\mathrm{X}^-) \Big\} &= \frac{\mathbb{E}_{P^-}[\mathcal{S}_{\boldsymbol{\theta}}(\mathrm{X}^-; \mathrm{x}) \cdot \exp\{\mathcal{S}_{\boldsymbol{\theta}}(\mathrm{X}^-; \mathrm{x})/\gamma^*\}]}{\mathbb{E}_{P^-}[\exp\{\mathcal{S}_{\boldsymbol{\theta}}(\mathrm{X}^-; \mathrm{x})/\gamma^*\}]}\\
&= \hbar'(\bar{\varrho}), \tag{A12}
\end{aligned}
$$

where we let $\varrho \triangleq 1/\gamma$, $\bar{\varrho} \triangleq 1/\gamma^*$, and $\hbar(\varrho) = \log \mathbb{E}_{P^-}[\exp\{\varrho \cdot \mathcal{S}_{\boldsymbol{\theta}}(\mathrm{X}^-; \mathrm{x})\}]$. Here $\hbar$ is the cumulant generating function of $\mathcal{S}_{\boldsymbol{\theta}}(\mathrm{X}^-; \mathrm{x})$, which is infinitely differentiable and strictly convex for non-constant $\mathcal{S}_{\boldsymbol{\theta}}(\mathrm{X}^-; \mathrm{x})$, and passes through the origin (Shalizi & Kontorovich, 2006). Moreover, using a power series expansion, we obtain that:

$$
\hbar(\varrho) = \sum_{j=1}^{\infty} \hbar^{(j)}(0)\, \varrho^j,
$$

where $\hbar^{(j)}$ denotes the $j$th derivative of $\hbar$, and $\hbar^{(j)}(0)$ is referred to as the $j$th cumulant. It can be verified that

$$
\begin{aligned}
\hbar^{(1)}(0) &= \mathbb{E}_{P^-}\{\mathcal{S}_{\boldsymbol{\theta}}(\mathrm{X}^-; \mathrm{x})\};\\
\hbar^{(2)}(0) &= \mathbb{E}_{P^-}\Big\{ \big[\mathcal{S}_{\boldsymbol{\theta}}(\mathrm{X}^-; \mathrm{x}) - \mathbb{E}_{P^-}\{\mathcal{S}_{\boldsymbol{\theta}}(\mathrm{X}^-; \mathrm{x})\}\big]^2 \Big\} > 0;\\
\hbar^{(3)}(0) &= \mathbb{E}_{P^-}\Big\{ \big[\mathcal{S}_{\boldsymbol{\theta}}(\mathrm{X}^-; \mathrm{x}) - \mathbb{E}_{P^-}\{\mathcal{S}_{\boldsymbol{\theta}}(\mathrm{X}^-; \mathrm{x})\}\big]^3 \Big\}.
\end{aligned}
$$

By the strict convexity of $\hbar$, we have that $d\left\{\varrho\hbar'(\varrho) - \hbar(\varrho)\right\}/d\varrho = \hbar''(\varrho) > 0$, and hence $\varrho\hbar'(\varrho) - \hbar(\varrho)$ is strictly increasing in $\varrho$. Moreover, by (A11), using the Taylor series expansion, we obtain that

$$
\begin{aligned}
\delta &= \bar{\varrho}\,\hbar'(\bar{\varrho}) - \hbar(\bar{\varrho}) \\
&= \bar{\varrho}\sum_{j=0}^{+\infty}\frac{1}{j!}\hbar^{(j+1)}(0)\,\bar{\varrho}^j - \sum_{j=0}^{+\infty}\frac{1}{j!}\hbar^{(j)}(0)\,\bar{\varrho}^j \\
&= \sum_{j=1}^{+\infty}\frac{1}{(j-1)!}\hbar^{(j)}(0)\,\bar{\varrho}^j - \sum_{j=1}^{+\infty}\frac{1}{j!}\hbar^{(j)}(0)\,\bar{\varrho}^j \\
&= \sum_{j=1}^{+\infty}\left\{\frac{1}{(j-1)!} - \frac{1}{j!}\right\}\hbar^{(j)}(0)\,\bar{\varrho}^j \\
&= \frac{1}{2}\hbar^{(2)}(0)\,\bar{\varrho}^2 + \frac{1}{3}\hbar^{(3)}(0)\,\bar{\varrho}^3 + \mathcal{O}(\bar{\varrho}^4). \quad\quad\quad (A13)
\end{aligned}
$$

Since $\hbar^{(2)}(0) > 0$ and the remainder is continuous in $\varrho$, we conclude that there exists a small $\bar{\varrho}$ satisfying the equation (A13) for a small enough $\delta$, and that $\bar{\varrho}$ is the unique solution of (A11). Correspondingly, for $\gamma^* = 1/\bar{\varrho}$, the associated $\mathsf{L}^*$ satisfies the constraint $\mathbb{E}_{P^-}\left[\mathsf{L}^*(\mathsf{X}^-)\log\left\{\mathsf{L}^*(\mathsf{X}^-)\right\}\right] = \delta$. Hence, $\mathcal{R}_{\mathsf{x}}^-(\boldsymbol{\theta}; P^-, \delta) = \mathbb{E}_{P^-}\left\{\mathcal{S}_{\boldsymbol{\theta}}(\mathsf{X}^-; \mathsf{x})\mathsf{L}^*(\mathsf{X}^-)\right\}$.

**Step (iii): Mean-dispersion form of the robust risk.**

Now, we examine the form of the robust risk. By (A13), we have

$$
\frac{2\delta}{\hbar^{(2)}(0)} = \bar{\varrho}^2 + \frac{2\hbar^{(3)}(0)}{3\hbar^{(2)}(0)}\bar{\varrho}^3 + \mathcal{O}(\bar{\varrho}^4) = \bar{\varrho}^2\left\{1 + \frac{2\hbar^{(3)}(0)}{3\hbar^{(2)}(0)}\bar{\varrho} + \mathcal{O}(\bar{\varrho}^2)\right\},
$$

and further obtain that

$$
\begin{aligned}
\bar{\varrho} &= \sqrt{\frac{2\delta}{\hbar^{(2)}(0)}} \cdot \sqrt{1\Big/\left\{1 + \frac{2\hbar^{(3)}(0)}{3\hbar^{(2)}(0)}\bar{\varrho} + \mathcal{O}(\bar{\varrho}^2)\right\}} \\
&= \sqrt{\frac{2\delta}{\hbar^{(2)}(0)}} \cdot \sqrt{1 - \frac{2\hbar^{(3)}(0)}{3\hbar^{(2)}(0)}\bar{\varrho} + \mathcal{O}(\bar{\varrho}^2)} \\
&= \sqrt{\frac{2\delta}{\hbar^{(2)}(0)}} \cdot \left\{1 - \frac{\hbar^{(3)}(0)}{3\hbar^{(2)}(0)}\bar{\varrho} + \mathcal{O}(\bar{\varrho}^2)\right\} \\
&= \sqrt{\frac{2\delta}{\hbar^{(2)}(0)}} - \frac{2\hbar^{(3)}(0)}{3\{\hbar^{(2)}(0)\}^2}\delta + \mathcal{O}(\delta).
\end{aligned}
$$

Hence, by (A12), we have that

$$
\begin{aligned}
\mathcal{R}_{\mathsf{x}}^-(\boldsymbol{\theta}; P^-, \delta) &= \mathbb{E}_{P^-}\left\{\mathcal{S}_{\boldsymbol{\theta}}(\mathsf{X}^-; \mathsf{x})\mathsf{L}^*(\mathsf{X}^-)\right\} \\
&= \hbar'(\bar{\varrho}) = \hbar^{(1)}(0) + \hbar^{(2)}(0)\bar{\varrho} + \frac{\hbar^{(3)}(0)}{2}\bar{\varrho}^2 + \mathcal{O}(\bar{\varrho}^2) \\
&= \hbar^{(1)}(0) + \sqrt{2\hbar^{(2)}(0)\delta} + \mathcal{O}(\delta) \\
&= \mathbb{E}_{P^-}\left\{\mathcal{S}_{\boldsymbol{\theta}}(\mathsf{X}^-; \mathsf{x})\right\} + \sqrt{2\mathbb{E}_{P^-}\left\{\left[\mathcal{S}_{\boldsymbol{\theta}}(\mathsf{X}^-; \mathsf{x}) - \mathbb{E}_{P^-}\left\{\mathcal{S}_{\boldsymbol{\theta}}(\mathsf{X}^-; \mathsf{x})\right\}\right]^2\right\}\delta} + \mathcal{O}(\delta).
\end{aligned}
$$

Therefore, the proof is established.

**Proof of (iii) with $d$ set as the $p$-Wasserstein distance:**

If the discrepancy metric $d$ in (3) is chosen as the $p$-Wasserstein distance, then the robust risk $\mathcal{R}_{\mathrm{x}}^{\text{-}}(\boldsymbol{\theta}; P^{\text{-}}, \delta)$ is expressed as

$$\mathcal{R}_{\mathrm{x}}^{\text{-}}(\boldsymbol{\theta}; P^{\text{-}}, \delta) = \sup_{Q^{\text{-}} \in \mathcal{P}(\Omega)} \left[ \mathbb{E}_Q\text{-}\left\{ \mathcal{S}_{\boldsymbol{\theta}}(X^{\text{-}}; \mathrm{x}) \right\} : W_p(Q^{\text{-}}, P^{\text{-}}) \le \delta \right]. \tag{A14}$$

Let $\Delta \mathcal{R}_{\mathrm{x}}^{\text{-}} \triangleq \mathcal{R}_{\mathrm{x}}^{\text{-}}(\boldsymbol{\theta}; P^{\text{-}}, \delta) - \mathbb{E}_P\text{-}\left\{ \mathcal{S}_{\boldsymbol{\theta}}(X^{\text{-}}; \mathrm{x}) \right\}$ denote the difference of the robust risk and the nominal risk. By Lemma 1, we have that

$$\Delta \mathcal{R}_{\mathrm{x}}^{\text{-}} = \min_{\gamma \ge 0} \left\{ \gamma \delta^p + \mathbb{E}_P\text{-} \left[ \sup_{\widetilde{\mathrm{x}}^{\text{-}} \in \Omega} \left\{ \mathcal{S}_{\boldsymbol{\theta}}(\widetilde{\mathrm{x}}^{\text{-}}; \mathrm{x}) - \gamma \| \widetilde{\mathrm{x}}^{\text{-}} - X^{\text{-}} \|^p \right\} \right] \right\} - \mathbb{E}_P\text{-}\left\{ \mathcal{S}_{\boldsymbol{\theta}}(X^{\text{-}}; \mathrm{x}) \right\}$$

$$= \min_{\gamma \ge 0} \left( \gamma \delta^p + \mathbb{E}_P\text{-} \left\{ \sup_{\widetilde{\mathrm{x}}^{\text{-}} \in \Omega} \left[ \left\{ \mathcal{S}_{\boldsymbol{\theta}}(\widetilde{\mathrm{x}}^{\text{-}}; \mathrm{x}) - \mathcal{S}_{\boldsymbol{\theta}}(X^{\text{-}}; \mathrm{x}) \right\} - \gamma \| \widetilde{\mathrm{x}}^{\text{-}} - X^{\text{-}} \|^p \right] \right\} \right). \tag{A15}$$

**Step (i): Upper bound on $\mathcal{S}_{\boldsymbol{\theta}}(\widetilde{\mathrm{x}}^{\text{-}}; \mathrm{x}) - \mathcal{S}_{\boldsymbol{\theta}}(\mathrm{x}^{\text{-}}; \mathrm{x})$.**

For any $\widetilde{\mathrm{x}}^{\text{-}}, \mathrm{x}^{\text{-}} \in \mathcal{X}$, by the mean value theorem, there exists $\check{\mathrm{x}}^{\text{-}} \in \mathcal{X}$ between $\widetilde{\mathrm{x}}^{\text{-}}$ and $\mathrm{x}^{\text{-}}$ such that

$$\mathcal{S}_{\boldsymbol{\theta}}(\widetilde{\mathrm{x}}^{\text{-}}; \mathrm{x}) - \mathcal{S}_{\boldsymbol{\theta}}(\mathrm{x}^{\text{-}}; \mathrm{x}) = \langle \nabla \mathcal{S}_{\boldsymbol{\theta}}(\check{\mathrm{x}}^{\text{-}}; \mathrm{x}), \widetilde{\mathrm{x}}^{\text{-}} - \mathrm{x}^{\text{-}} \rangle,$$

which implies that

$$|\mathcal{S}_{\boldsymbol{\theta}}(\widetilde{\mathrm{x}}^{\text{-}}; \mathrm{x}) - \mathcal{S}_{\boldsymbol{\theta}}(\mathrm{x}^{\text{-}}; \mathrm{x}) - \langle \nabla \mathcal{S}_{\boldsymbol{\theta}}(\mathrm{x}^{\text{-}}; \mathrm{x}), \widetilde{\mathrm{x}}^{\text{-}} - \mathrm{x}^{\text{-}} \rangle|$$
$$= |\langle \nabla \mathcal{S}_{\boldsymbol{\theta}}(\check{\mathrm{x}}^{\text{-}}; \mathrm{x}) - \nabla \mathcal{S}_{\boldsymbol{\theta}}(\mathrm{x}^{\text{-}}; \mathrm{x}), \widetilde{\mathrm{x}}^{\text{-}} - \mathrm{x}^{\text{-}} \rangle|$$
$$\le \| \nabla \mathcal{S}_{\boldsymbol{\theta}}(\widetilde{\mathrm{x}}^{\text{-}}; \mathrm{x}) - \nabla \mathcal{S}_{\boldsymbol{\theta}}(\mathrm{x}^{\text{-}}; \mathrm{x}) \|_* \| \widetilde{\mathrm{x}}^{\text{-}} - \mathrm{x}^{\text{-}} \|$$
$$\le \| \nabla \mathcal{S}_{\boldsymbol{\theta}}(\widetilde{\mathrm{x}}^{\text{-}}; \mathrm{x}) - \nabla \mathcal{S}_{\boldsymbol{\theta}}(\mathrm{x}^{\text{-}}; \mathrm{x}) \|_* \| \widetilde{\mathrm{x}}^{\text{-}} - \mathrm{x}^{\text{-}} \|, \tag{A16}$$

where the inequality in the penultimate step is due to the Cauchy–Schwarz inequality.

If $\| \widetilde{\mathrm{x}}^{\text{-}} - \mathrm{x}^{\text{-}} \| \le \eta_0$, by the smoothness condition (b), we have that

$$\| \nabla \mathcal{S}_{\boldsymbol{\theta}}(\widetilde{\mathrm{x}}^{\text{-}}; \mathrm{x}) - \nabla \mathcal{S}_{\boldsymbol{\theta}}(\mathrm{x}^{\text{-}}; \mathrm{x}) \|_* \le \mathcal{M}_3 \| \widetilde{\mathrm{x}}^{\text{-}} - \mathrm{x}^{\text{-}} \|. \tag{A17}$$

If $\| \widetilde{\mathrm{x}}^{\text{-}} - \mathrm{x}^{\text{-}} \| \ge \eta_0$, by the smoothness condition (a), we have that

$$\| \nabla \mathcal{S}_{\boldsymbol{\theta}}(\widetilde{\mathrm{x}}^{\text{-}}; \mathrm{x}) - \nabla \mathcal{S}_{\boldsymbol{\theta}}(\mathrm{x}^{\text{-}}; \mathrm{x}) \|_* \le \mathcal{M}_1 + \mathcal{M}_2 \| \widetilde{\mathrm{x}}^{\text{-}} - \mathrm{x}^{\text{-}} \|^{\zeta - 1}. \tag{A18}$$

Combining (A16), (A17) and (A18), we obtain that

$$|\mathcal{S}_{\boldsymbol{\theta}}(\widetilde{\mathrm{x}}^{\text{-}}; \mathrm{x}) - \mathcal{S}_{\boldsymbol{\theta}}(\mathrm{x}^{\text{-}}; \mathrm{x}) - < \nabla \mathcal{S}_{\boldsymbol{\theta}}(\mathrm{x}^{\text{-}}; \mathrm{x}), \widetilde{\mathrm{x}}^{\text{-}} - \mathrm{x}^{\text{-}} > |$$
$$= \mathbb{1}(\| \widetilde{\mathrm{x}}^{\text{-}} - \mathrm{x}^{\text{-}} \| \le \eta_0) \cdot \mathcal{M}_3 \| \widetilde{\mathrm{x}}^{\text{-}} - \mathrm{x}^{\text{-}} \|^2 + \mathbb{1}(\| \widetilde{\mathrm{x}}^{\text{-}} - \mathrm{x}^{\text{-}} \| \ge \eta_0) \cdot \left( \mathcal{M}_1 \| \widetilde{\mathrm{x}}^{\text{-}} - \mathrm{x}^{\text{-}} \| + \mathcal{M}_2 \| \widetilde{\mathrm{x}}^{\text{-}} - \mathrm{x}^{\text{-}} \|^{\zeta} \right)$$
$$\triangleq \mathcal{I}_1 + \mathcal{I}_2,$$

where

$$\mathcal{I}_1 \triangleq \mathbb{1}(\| \widetilde{\mathrm{x}}^{\text{-}} - \mathrm{x}^{\text{-}} \| \le \eta_0) \cdot \mathcal{M}_3 \| \widetilde{\mathrm{x}}^{\text{-}} - \mathrm{x}^{\text{-}} \|^2;$$
$$\mathcal{I}_2 \triangleq \mathbb{1}(\| \widetilde{\mathrm{x}}^{\text{-}} - \mathrm{x}^{\text{-}} \| \ge \eta_0) \cdot \left( \mathcal{M}_1 \| \widetilde{\mathrm{x}}^{\text{-}} - \mathrm{x}^{\text{-}} \| + \mathcal{M}_2 \| \widetilde{\mathrm{x}}^{\text{-}} - \mathrm{x}^{\text{-}} \|^{\zeta} \right).$$

For $\mathcal{I}_1$: if $1 \le p \le 2$, we have

$$\mathcal{I}_1 \le \mathbb{1}(\| \widetilde{\mathrm{x}}^{\text{-}} - \mathrm{x}^{\text{-}} \| \le \eta_0) \cdot \mathcal{M}_3 \left( \frac{\eta_0}{\| \widetilde{\mathrm{x}}^{\text{-}} - \mathrm{x}^{\text{-}} \|} \right)^{2-p} \| \widetilde{\mathrm{x}}^{\text{-}} - \mathrm{x}^{\text{-}} \|^2$$
$$\le \mathcal{M}_3 \eta_0^{2-p} \| \widetilde{\mathrm{x}}^{\text{-}} - \mathrm{x}^{\text{-}} \|^p.$$

If $p > 2$, we have $\mathcal{I}_1 \le \mathcal{M}_3 \| \widetilde{\mathrm{x}}^{\text{-}} - \mathrm{x}^{\text{-}} \|^2$.

For $\mathcal{I}_2$, we have the following upper bound:

$$\mathcal{I}_2 \le \mathbb{1}(\| \widetilde{\mathrm{x}}^{\text{-}} - \mathrm{x}^{\text{-}} \| \ge \eta_0) \cdot \left\{ \mathcal{M}_1 \left( \frac{\| \widetilde{\mathrm{x}}^{\text{-}} - \mathrm{x}^{\text{-}} \|}{\eta_0} \right)^{p-1} \| \widetilde{\mathrm{x}}^{\text{-}} - \mathrm{x}^{\text{-}} \| + \mathcal{M}_2 \left( \frac{\| \widetilde{\mathrm{x}}^{\text{-}} - \mathrm{x}^{\text{-}} \|}{\eta_0} \right)^{p-\zeta} \| \widetilde{\mathrm{x}}^{\text{-}} - \mathrm{x}^{\text{-}} \|^{\zeta} \right\}$$
$$\le \left( \mathcal{M}_1 \eta_0^{-(p-1)} + \mathcal{M}_2 \eta_0^{-(p-\zeta)} \right) \| \widetilde{\mathrm{x}}^{\text{-}} - \mathrm{x}^{\text{-}} \|^p.$$

Combining the discussions above, we have that

$$|\mathcal{S}_{\boldsymbol{\theta}}(\widetilde{x}^-;x) - \mathcal{S}_{\boldsymbol{\theta}}(x^-;x) - <\nabla\mathcal{S}_{\boldsymbol{\theta}}(x^-;x),\widetilde{x}^- - x^- > |$$
$$\leq \begin{cases} \bar{\mathcal{M}}\,\|\widetilde{x}^- - x^-\|^p, \text{ if } 1 \leq p \leq 2; \\ \bar{\mathcal{M}}\left(\|\widetilde{x}^- - x^-\|^p + \|\widetilde{x}^- - x^-\|^2\right), \text{ if } p > 2, \end{cases} \tag{A19}$$

where $\bar{\mathcal{M}} \triangleq \max\{\mathcal{M}_3\eta_0^{2-p}, \mathcal{M}_3, \left(\mathcal{M}_1\eta_0^{-(p-1)} + \mathcal{M}_2\eta_0^{-(p-\zeta)}\right)\}$.

**Step (ii): Mean-dispersion form of the robust risk when $p \in [1, 2]$.**

When $p \in [1, 2]$, by (A15) and (A19), we have that

$$\Delta\mathcal{R}_x^- \leq \min_{\gamma \geq 0}\left(\gamma\delta^p + \mathbb{E}_{P^-}\left\{\sup_{\widetilde{x}^- \in \Omega}\left[\left\{\langle\nabla\mathcal{S}_{\boldsymbol{\theta}}(X^-;x),\widetilde{x}^- - X^-\rangle + \bar{\mathcal{M}}\,\|\widetilde{x}^- - X^-\|^p\right\} - \gamma\|\widetilde{x}^- - X^-\|^p\right]\right\}\right)$$

$$= \min_{\gamma \geq 0}\left\{\gamma\delta^p + \mathbb{E}_{P^-}\left[\sup_{\widetilde{x}^- \in \Omega}\left\{\langle\nabla\mathcal{S}_{\boldsymbol{\theta}}(X^-;x),\widetilde{x}^- - X^-\rangle - (\gamma - \bar{\mathcal{M}})\|\widetilde{x}^- - X^-\|^p\right\}\right]\right\}$$

$$\leq \min_{\gamma \geq 0}\left\{\gamma\delta^p + \mathbb{E}_{P^-}\left[\sup_{\widetilde{x}^- \in \Omega}\left\{\|\nabla\mathcal{S}_{\boldsymbol{\theta}}(X^-;x)\|_*\|\widetilde{x}^- - X^-\| - (\gamma - \bar{\mathcal{M}})\|\widetilde{x}^- - X^-\|^p\right\}\right]\right\}$$

$$= \min_{\gamma \geq -\bar{\mathcal{M}}}\left\{\gamma\delta^p + \mathbb{E}_{P^-}\left[\sup_{t \geq 0}\left\{\|\nabla\mathcal{S}_{\boldsymbol{\theta}}(X^-;x)\|_*t - \gamma t^p\right\}\right]\right\} + \bar{\mathcal{M}}\delta^p$$

$$\leq \min_{\gamma \geq 0}\left\{\gamma\delta^p + \mathbb{E}_{P^-}\left[\sup_{t \geq 0}\left\{\|\nabla\mathcal{S}_{\boldsymbol{\theta}}(X^-;x)\|_*t - \gamma t^p\right\}\right]\right\} + \bar{\mathcal{M}}\delta^p$$

$$\triangleq \mathcal{I}_4 + \bar{\mathcal{M}}\delta^p, \tag{A20}$$

where $\mathcal{I}_4 \triangleq \min_{\gamma \geq 0}\left\{\gamma\delta^p + \mathbb{E}_{P^-}\left[\sup_{t \geq 0}\left\{\|\nabla\mathcal{S}_{\boldsymbol{\theta}}(X^-;x)\|_*t - \gamma t^p\right\}\right]\right\}$, and the third step is due to the Cauchy–Schwarz inequality.

By taking the derivative of the function in $\mathcal{I}_4$ with respect to $t$ and setting it to zero, we obtain the optimal value of $t$, given by

$$t^* = \{\|\nabla\mathcal{S}_{\boldsymbol{\theta}}(X^-;x)\|_*/(\gamma p)\}^{1/(p-1)}.$$

Let $q$ denote the Hölder number of $p$, that is $\frac{1}{p} + \frac{1}{q} = 1$. Then, $q = \frac{p}{p-1}$ and $\frac{q}{p} = \frac{1}{p-1}$. We have that

$$\sup_{t \geq 0}\left\{\|\nabla\mathcal{S}_{\boldsymbol{\theta}}(X^-;x)\|_*t - \gamma t^p\right\}$$
$$= \|\nabla\mathcal{S}_{\boldsymbol{\theta}}(X^-;x)\|_*t^* - \gamma(t^*)^p$$
$$= \|\nabla\mathcal{S}_{\boldsymbol{\theta}}(X^-;x)\|_* \cdot \left\{\frac{\|\nabla\mathcal{S}_{\boldsymbol{\theta}}(X^-;x)\|_*}{\gamma p}\right\}^{\frac{1}{p-1}} - \gamma \cdot \left\{\frac{\|\nabla\mathcal{S}_{\boldsymbol{\theta}}(X^-;x)\|_*}{\gamma p}\right\}^{\frac{p}{p-1}}$$
$$= \|\nabla\mathcal{S}_{\boldsymbol{\theta}}(X^-;x)\|_*^{\frac{p}{p-1}}(\gamma p)^{-\frac{1}{p-1}} - \|\nabla\mathcal{S}_{\boldsymbol{\theta}}(X^-;x)\|_*^{\frac{p}{p-1}}\gamma^{-\frac{1}{p-1}}p^{-\frac{p}{p-1}}$$
$$= \|\nabla\mathcal{S}_{\boldsymbol{\theta}}(X^-;x)\|_*^q(\gamma p)^{-\frac{1}{p-1}}\left(1 - \frac{1}{p}\right).$$

Thus, we further obtain that

$$\mathcal{I}_4 = \min_{\gamma \geq 0}\left[\gamma\delta^p + \left(1 - \frac{1}{p}\right)(\gamma p)^{-\frac{1}{p-1}}\mathbb{E}_{P^-}\left\{\|\nabla\mathcal{S}_{\boldsymbol{\theta}}(X^-;x)\|_*^q\right\}\right]. \tag{A21}$$

Similarly, by taking the derivative of the function in (A21) with respect to $\gamma$ and setting it to zero, we obtain the optimal value of $\gamma$, given by

$$\gamma^* = \frac{1}{p}\delta^{-(p-1)}\{\mathbb{E}_{P^-}\|\nabla\mathcal{S}_{\boldsymbol{\theta}}(X^-;x)\|_*^q\}^{1/q}.$$

Hence, by substituting $\gamma^*$ into the corresponding expression and simplifying, we further obtain that

$$
\begin{aligned}
\mathcal{I}_4 =& \frac{1}{p}\delta^{-(p-1)}\left\{\mathbb{E}_{P^{\text{-}}}\|\nabla\mathcal{S}_{\boldsymbol{\theta}}(\mathrm{X}^{\text{-}};\mathrm{x})\|_*^q\right\}^{1/q}\delta^p \\
&+ \left\{\frac{1}{p}\delta^{-(p-1)}\left\{\mathbb{E}_{P^{\text{-}}}\|\nabla\mathcal{S}_{\boldsymbol{\theta}}(\mathrm{X}^{\text{-}};\mathrm{x})\|_*^q\right\}^{1/q}\right\}^{-\frac{1}{p-1}}\left(\frac{p-1}{p}\right)p^{-\frac{1}{p-1}} \\
=& \frac{1}{p}\delta\left\{\mathbb{E}_{P^{\text{-}}}\|\nabla\mathcal{S}_{\boldsymbol{\theta}}(\mathrm{X}^{\text{-}};\mathrm{x})\|_*^q\right\}^{1/q} + \left(\frac{p-1}{p}\right)\delta\left\{\mathbb{E}_{P^{\text{-}}}\|\nabla\mathcal{S}_{\boldsymbol{\theta}}(\mathrm{X}^{\text{-}};\mathrm{x})\|_*^q\right\}^{1/q} \\
=& \delta\left\{\mathbb{E}_{P^{\text{-}}}\|\nabla\mathcal{S}_{\boldsymbol{\theta}}(\mathrm{X}^{\text{-}};\mathrm{x})\|_*^q\right\}^{1/q}.
\end{aligned}
\tag{A22}
$$

Combining (A20) and (A22), we obtain that

$$
\Delta\mathcal{R}_{\mathrm{x}}^{\text{-}} \le \delta\left\{\mathbb{E}_{P^{\text{-}}}\|\nabla\mathcal{S}_{\boldsymbol{\theta}}(\mathrm{X}^{\text{-}};\mathrm{x})\|_*^q\right\}^{1/q} + \bar{\mathcal{M}}\delta^p.
$$

**Step (iii): Mean-dispersion form of the robust risk when $p \in (2, \infty)$.**

When $p \in (2, \infty)$, by (A15) and (A19), similar to (A20) in Step (ii), we have that

$$
\begin{aligned}
\Delta\mathcal{R}_{\mathrm{x}}^{\text{-}} \le& \min_{\gamma \ge 0}\left(\gamma\delta^p + \mathbb{E}_{P^{\text{-}}}\left\{\sup_{\widetilde{\mathrm{x}}^{\text{-}}\in\Omega}\left[\left\{\langle\nabla\mathcal{S}_{\boldsymbol{\theta}}(\mathrm{X}^{\text{-}};\mathrm{x}),\widetilde{\mathrm{x}}^{\text{-}}-\mathrm{X}^{\text{-}}\rangle\right.\right.\right.\right. \\
&\left.\left.\left.\left. + \bar{\mathcal{M}}(\|\widetilde{\mathrm{x}}^{\text{-}}-\mathrm{X}^{\text{-}}\|^p + \|\widetilde{\mathrm{x}}^{\text{-}}-\mathrm{X}^{\text{-}}\|^2)\right\} - \gamma\|\widetilde{\mathrm{x}}^{\text{-}}-\mathrm{X}^{\text{-}}\|^p\right]\right\}\right) \\
\le& \min_{\gamma\ge 0}\left\{\gamma\delta^p + \mathbb{E}_{P^{\text{-}}}\left[\sup_{\widetilde{\mathrm{x}}^{\text{-}}\in\Omega}\left\{\|\nabla\mathcal{S}_{\boldsymbol{\theta}}(\mathrm{X}^{\text{-}};\mathrm{x})\|_*\|\widetilde{\mathrm{x}}^{\text{-}}-\mathrm{X}^{\text{-}}\|\right.\right.\right. \\
&\left.\left.\left. + \bar{\mathcal{M}}\|\widetilde{\mathrm{x}}^{\text{-}}-\mathrm{X}^{\text{-}}\|^p + \bar{\mathcal{M}}\|\widetilde{\mathrm{x}}^{\text{-}}-\mathrm{X}^{\text{-}}\|^2 - \gamma\|\widetilde{\mathrm{x}}^{\text{-}}-\mathrm{X}^{\text{-}}\|^p\right\}\right]\right\} \\
=& \min_{\gamma\ge 0}\left\{\gamma\delta^p + \mathbb{E}_{P^{\text{-}}}\left[\sup_{t\ge 0}\left\{\|\nabla\mathcal{S}_{\boldsymbol{\theta}}(\mathrm{X}^{\text{-}};\mathrm{x})\|_*t + \bar{\mathcal{M}}t^p + \bar{\mathcal{M}}t^2 - \gamma t^p\right\}\right]\right\} \\
\le& \min_{\gamma\ge 0}\left\{\gamma\delta^p + \mathbb{E}_{P^{\text{-}}}\left[\sup_{t\ge 0}\left\{\|\nabla\mathcal{S}_{\boldsymbol{\theta}}(\mathrm{X}^{\text{-}};\mathrm{x})\|_*t + \bar{\mathcal{M}}t^2 - \gamma t^p\right\}\right]\right\} + \bar{\mathcal{M}}\delta^p \\
=& \min_{\gamma_1,\gamma_2\ge 0}\left\{(\gamma_1+\gamma_2)\delta^p + \mathbb{E}_{P^{\text{-}}}\left[\sup_{t\ge 0}\left\{\|\nabla\mathcal{S}_{\boldsymbol{\theta}}(\mathrm{X}^{\text{-}};\mathrm{x})\|_*t + \bar{\mathcal{M}}t^2 - (\gamma_1+\gamma_2)t^p\right\}\right]\right\} + \bar{\mathcal{M}}\delta^p \\
\le& \min_{\gamma_1\ge 0}\left\{\gamma_1\delta^p + \mathbb{E}_{P^{\text{-}}}\left[\sup_{t\ge 0}\left\{\|\nabla\mathcal{S}_{\boldsymbol{\theta}}(\mathrm{X}^{\text{-}};\mathrm{x})\|_*t - \gamma_1 t^p\right\}\right]\right\} \\
&+ \min_{\gamma_2\ge 0}\left\{\gamma_2\delta^p + \sup_{t\ge 0}\left(\bar{\mathcal{M}}t^2 - \gamma_2 t^p\right)\right\} + \bar{\mathcal{M}}\delta^p \\
\triangleq& \mathcal{I}_5 + \mathcal{I}_6 + \bar{\mathcal{M}}\delta^p
\end{aligned}
\tag{A23}
$$

where

$$
\begin{aligned}
\mathcal{I}_5 &\triangleq \min_{\gamma_1\ge 0}\left\{\gamma_1\delta^p + \mathbb{E}_{P^{\text{-}}}\left[\sup_{t\ge 0}\left\{\|\nabla\mathcal{S}_{\boldsymbol{\theta}}(\mathrm{X}^{\text{-}};\mathrm{x})\|_*t - \gamma_1 t^p\right\}\right]\right\}; \\
\mathcal{I}_6 &\triangleq \min_{\gamma_2\ge 0}\left\{\gamma_2\delta^p + \sup_{t\ge 0}\left(\bar{\mathcal{M}}t^2 - \gamma_2 t^p\right)\right\}.
\end{aligned}
$$

For $\mathcal{I}_5$, similar to the discussion on $\mathcal{I}_4$ with $p \in [1, 2]$ as in (A22), we obtain that, for $p \in (2, \infty)$,

$$
\mathcal{I}_5 = \delta\left\{\mathbb{E}_{P^{\text{-}}}\|\nabla\mathcal{S}_{\boldsymbol{\theta}}(\mathrm{X}^{\text{-}};\mathrm{x})\|_*^q\right\}^{1/q}.
\tag{A24}
$$

For $\mathcal{I}_6$, by taking the derivative of the function in $\mathcal{I}_6$ with respect to $t$ and setting it to zero, we obtain the optimal value of $t$, given by $t^* = \left\{2\bar{\mathcal{M}}/(\gamma_2 p)\right\}^{1/(p-2)}$, leading to

$$
\begin{aligned}
\mathcal{I}_6 &= \min_{\gamma_2 \geq 0} \left\{ \gamma_2 \delta^p + \bar{\mathcal{M}}(t^*)^2 - \gamma_2(t^*)^p \right\} \\
&= \min_{\gamma_2 \geq 0} \left\{ \gamma_2 \delta^p + \bar{\mathcal{M}} \cdot \left(\frac{2\bar{\mathcal{M}}}{\gamma_2 p}\right)^{\frac{2}{p-2}} - \gamma_2 \left(\frac{2\bar{\mathcal{M}}}{\gamma_2 p}\right)^{\frac{p}{p-2}} \right\} \\
&= \min_{\gamma_2 \geq 0} \left\{ \gamma_2 \delta^p + \left(\frac{\gamma_2 p}{2}\right)^{-\frac{2}{p-2}} \bar{\mathcal{M}}^{\frac{p}{p-2}} - \gamma_2^{-\frac{2}{p-2}} \cdot \left(\frac{p}{2}\right)^{-\frac{2}{p-2}} \cdot \left(\frac{p}{2}\right)^{-1} \cdot \bar{\mathcal{M}}^{\frac{p}{p-2}} \right\} \\
&= \min_{\gamma_2 \geq 0} \left\{ \gamma_2 \delta^p + \frac{p-2}{p} \left(\frac{\gamma_2 p}{2}\right)^{-\frac{2}{p-2}} \bar{\mathcal{M}}^{\frac{p}{p-2}} \right\}.
\end{aligned}
\tag{A25}
$$

By taking the derivative of the function in (A25) with respect to $\gamma_2$, we obtain the optimal value of $\gamma_2$, given by

$$
\gamma_2^* = \bar{\mathcal{M}} \delta^{-(p-2)} \left(\frac{p}{2}\right)^{-1},
$$

yielding

$$
\mathcal{I}_6 = \gamma_2^* \delta^p + \frac{p-2}{p} \left(\frac{\gamma_2^* p}{2}\right)^{-\frac{2}{p-2}} \bar{\mathcal{M}}^{\frac{p}{p-2}} = \bar{\mathcal{M}} \delta^2.
\tag{A26}
$$

Combining (A24), (A26), and (A26), we obtain

$$
\Delta\mathcal{R}_{\mathrm{x}}^{\text{-}} \leq \delta \left\{ \mathbb{E}_{P^{\text{-}}} \|\nabla \mathcal{S}_{\boldsymbol{\theta}}(\mathrm{X}^{\text{-}}; \mathrm{x})\|_*^q \right\}^{1/q} + \bar{\mathcal{M}} \delta^2 + \bar{\mathcal{M}} \delta^p.
\tag{A27}
$$

Hence, the proof is completed.

**Proof of Theorem 4.1.**

To show the results, we examine the outlier robust risk (4) for different choices of the discrepancy metric $d$ in (3). Proof techniques in Zhai et al. (2021) are used.

**Proof of (i) with $d$ set as the $\chi^2$-divergence:**

If the discrepancy metric $d$ in (3) is chosen as the $\chi^2$-divergence, by (4) and Lemma 4, we have that

$$
\begin{aligned}
\mathcal{R}_{\mathrm{x}}^{\text{-}}(\boldsymbol{\theta}; P_{\mathrm{train}}^{\text{-}}, \delta, \epsilon) &= \inf_{P' \in \mathcal{P}_p(\mathcal{X})} \left\{ \mathcal{R}_{\mathrm{x}}^{\text{-}}(\boldsymbol{\theta}; P', \delta) : \exists \widetilde{P}' \in \mathcal{P}_p(\mathcal{X}) \text{ s.t. } P_{\mathrm{train}}^{\text{-}} = (1-\epsilon)P' + \epsilon\widetilde{P}' \right\} \\
&= \inf_{P' \in \mathcal{P}_p(\mathcal{X})} \left\{ \mathbb{E}_{P'}\left\{\mathcal{S}_{\boldsymbol{\theta}}(\mathrm{X}^{\text{-}}; \mathrm{x})\right\} + \sqrt{\delta \mathbb{V}_{P'}\left\{\mathcal{S}_{\boldsymbol{\theta}}(\mathrm{X}^{\text{-}}; \mathrm{x})\right\}} : \right. \\
&\qquad\qquad \left. \exists \widetilde{P}' \in \mathcal{P}_p(\mathcal{X}) \text{ s.t. } P_{\mathrm{train}}^{\text{-}} = (1-\epsilon)P' + \epsilon\widetilde{P}' \right\}.
\end{aligned}
\tag{A28}
$$

We consider the following quantity:

$$
\begin{aligned}
\Re_1 &\triangleq \inf_{P' \in \mathcal{P}_p(\mathcal{X})} \left\{ \mathbb{E}_{P'}\left\{\mathcal{S}_{\boldsymbol{\theta}}(\mathrm{X}^{\text{-}}; \mathrm{x})\right\} : \exists \widetilde{P}' \in \mathcal{P}_p(\mathcal{X}) \text{ s.t. } P_{\mathrm{train}}^{\text{-}} = (1-\epsilon)P' + \epsilon\widetilde{P}' \right\} \\
&= \inf_{P' \in \mathcal{P}_p(\mathcal{X})} \left\{ \int_0^{+\infty} \left[1 - P'\left\{\mathcal{S}_{\boldsymbol{\theta}}(\mathrm{X}^{\text{-}}; \mathrm{x}) \leq s\right\}\right] ds : \exists \widetilde{P}' \in \mathcal{P}_p(\mathcal{X}) \text{ s.t. } P_{\mathrm{train}}^{\text{-}} = (1-\epsilon)P' + \epsilon\widetilde{P}' \right\},
\end{aligned}
\tag{A29}
$$

where in the second step, we use the fact that for a nonnegative random variable $Z$ with cumulative distribution function $F$, $\mathbb{E}_F(Z^k) = k \int_0^{+\infty} u^{k-1}\{1 - F(u)\}du$ if the $k$th moment $\mathbb{E}_F(Z^k)$ exists.

Since $P_{\mathrm{train}}^{\text{-}} = (1-\epsilon)P' + \epsilon\widetilde{P}'$, we have that for any $s \geq 0$,

$$
P'\left\{\mathcal{S}_{\boldsymbol{\theta}}(\mathrm{X}^{\text{-}}; \mathrm{x}) \leq s\right\} \leq \min\left\{ \frac{1}{1-\epsilon} P_{\mathrm{train}}^{\text{-}}\left\{\mathcal{S}_{\boldsymbol{\theta}}(\mathrm{X}^{\text{-}}; \mathrm{x}) \leq s\right\}, 1 \right\}.
\tag{A30}
$$

As in Zhai et al. (2021), we show that the equality in (A30) can be achieved by some $P^* \in \mathcal{P}_p(\mathcal{X})$. Specifically, since $P^-_{\text{train}}$ and $\mathcal{S}_{\boldsymbol{\theta}}$ are continuous, there exists an $s^*$ such that

$$P^-_{\text{train}} \{\mathcal{S}_{\boldsymbol{\theta}}(X^-; x) > s^*\} = \epsilon.$$

Define

$$p^*(x^-) \triangleq \begin{cases} \dfrac{1}{1-\epsilon} p^-_{\text{train}}(x^-), & \text{if } \mathcal{S}_{\boldsymbol{\theta}}(x^-; x) \le s^*; \\ 0, & \text{if } \mathcal{S}_{\boldsymbol{\theta}}(x^-; x) > s^*, \end{cases} \tag{A31}$$

where $p^-_{\text{train}}$ represents the density or mass function of $P^-_{\text{train}}$. Let $P^*$ denote the associated measure of $p^*$. Then, we have

$$\begin{aligned}
\int_{\mathcal{X}} dP^*(x^-) &= \frac{1}{1-\epsilon} \int_{\mathcal{S}_{\boldsymbol{\theta}}(x^-; x) \le s^*} dP^-_{\text{train}}(x^-) \\
&= \frac{1}{1-\epsilon} P^-_{\text{train}} \{\mathcal{S}_{\boldsymbol{\theta}}(X^-; x) \le s^*\} \\
&= 1.
\end{aligned}$$

Therefore, $P^*$ defined in (A31) is the probability distribution achieving the equality in (A30). Thus, by substituting $P^*$ into (A29) and utilizing (A30), $\Re_1$ can be written as:

$$\begin{aligned}
\Re_1 &= \mathbb{E}_{P^*} \{\mathcal{S}_{\boldsymbol{\theta}}(X^-; x)\} \\
&= \int_0^{+\infty} [1 - P^* \{\mathcal{S}_{\boldsymbol{\theta}}(X^-; x) \le s\}] \, ds \\
&= \int_0^{+\infty} \left[1 - \frac{1}{1-\epsilon} P^-_{\text{train}} \{\mathcal{S}_{\boldsymbol{\theta}}(X^-; x) \le s\}\right] \mathbb{1} \left[P^-_{\text{train}} \{\mathcal{S}_{\boldsymbol{\theta}}(X^-; x) \le s\} \le 1-\epsilon\right] ds \\
&= \int_0^{+\infty} \left[1 - \frac{1}{1-\epsilon} P^-_{\text{train}} \{\mathcal{S}_{\boldsymbol{\theta}}(X^-; x) \le s\}\right] \mathbb{1}(s \le s^*) ds \\
&= \frac{1}{1-\epsilon} \left[(1-\epsilon)s^* - \int_0^{s^*} P^-_{\text{train}} \{\mathcal{S}_{\boldsymbol{\theta}}(X^-; x) \le s\} \, ds\right] \\
&= \frac{1}{1-\epsilon} \left\{\left[s \, P^-_{\text{train}} \{\mathcal{S}_{\boldsymbol{\theta}}(X^-; x) \le s\}\right]\Big|_0^{s^*} - \int_0^{s^*} P^-_{\text{train}} \{\mathcal{S}_{\boldsymbol{\theta}}(X^-; x) \le s\} \, ds\right\} \\
&= \frac{1}{1-\epsilon} \int_0^{s^*} s \, dP^-_{\text{train}} \{\mathcal{S}_{\boldsymbol{\theta}}(X^-; x) \le s\}.
\end{aligned} \tag{A32}$$

For the variance term in (A28), we consider the 2nd moment:

$$\begin{aligned}
\Re_2 &\triangleq \mathbb{E}_{P^*} \left[\{\mathcal{S}_{\boldsymbol{\theta}}(X^-; x)\}^2\right] \\
&= 2 \int_0^{+\infty} s \, [1 - P^* \{\mathcal{S}_{\boldsymbol{\theta}}(X^-; x) \le s\}] \, ds \\
&= \int_0^{+\infty} 2s \cdot \left[1 - \frac{1}{1-\epsilon} P^-_{\text{train}} \{\mathcal{S}_{\boldsymbol{\theta}}(X^-; x) \le s\}\right] \mathbb{1}(s \le s^*) ds \\
&= \frac{1}{1-\epsilon} \left[(1-\epsilon)(s^*)^2 - \int_0^{s^*} 2s P^-_{\text{train}} \{\mathcal{S}_{\boldsymbol{\theta}}(X^-; x) \le s\} \, ds\right] \\
&= \frac{1}{1-\epsilon} \left\{\left[s^2 \, P^-_{\text{train}} \{\mathcal{S}_{\boldsymbol{\theta}}(X^-; x) \le s\}\right]\Big|_0^{s^*} - \int_0^{s^*} 2s P^-_{\text{train}} \{\mathcal{S}_{\boldsymbol{\theta}}(X^-; x) \le s\} \, ds\right\} \\
&= \frac{1}{1-\epsilon} \int_0^{s^*} s^2 \, dP^-_{\text{train}} \{\mathcal{S}_{\boldsymbol{\theta}}(X^-; x) \le s\}.
\end{aligned} \tag{A33}$$

Thus, we obtain the upper bound on the outlier robust risk $\mathcal{R}_{\mathrm{x}}^{-}(\boldsymbol{\theta}; P_{\mathrm{train}}^{-}, \delta, \epsilon)$ given in (A28):

$$
\begin{aligned}
\mathcal{R}_{\mathrm{x}}^{-}(\boldsymbol{\theta}; P_{\mathrm{train}}^{-}, \delta, \epsilon) &\leq \mathbb{E}_{P^*}\left\{\mathcal{S}_{\boldsymbol{\theta}}(\mathrm{X}^-; \mathrm{x})\right\} + \sqrt{\delta \mathbb{V}_{P^*}\left\{\mathcal{S}_{\boldsymbol{\theta}}(\mathrm{X}^-; \mathrm{x})\right\}} \\
&= \Re_1 + \sqrt{\delta(\Re_2 - \Re_1^2)},
\end{aligned}
$$

where $\Re_1$ and $\Re_2$ are given in (A32) and (A33), respectively.

**Proof of (ii) with $d$ set as the KL-divergence:**

If the discrepancy metric $d$ in (3) is chosen as the KL-divergence, by (4) and Lemma 4, we have that

$$
\begin{aligned}
\mathcal{R}_{\mathrm{x}}^{-}(\boldsymbol{\theta}; P_{\mathrm{train}}^{-}, \delta, \epsilon) &= \inf_{P' \in \mathcal{P}_p(\mathcal{X})}\left\{\mathcal{R}_{\mathrm{x}}^{-}(\boldsymbol{\theta}; P', \delta) : \exists \widetilde{P}' \in \mathcal{P}_p(\mathcal{X}) \; s.t. \; P_{\mathrm{train}}^{-} = (1 - \epsilon)P' + \epsilon \widetilde{P}'\right\} \\
&= \inf_{P' \in \mathcal{P}_p(\mathcal{X})}\left\{\mathbb{E}_{P'}\left\{\mathcal{S}_{\boldsymbol{\theta}}(\mathrm{X}^-; \mathrm{x})\right\} + \sqrt{2\delta \mathbb{V}_{P'}\left\{\mathcal{S}_{\boldsymbol{\theta}}(\mathrm{X}^-; \mathrm{x})\right\}} : \right. \\
&\qquad\qquad \left. \exists \widetilde{P}' \in \mathcal{P}_p(\mathcal{X}) \; s.t. \; P_{\mathrm{train}}^{-} = (1 - \epsilon)P' + \epsilon \widetilde{P}'\right\}.
\end{aligned}
$$

Similar to the proof of Theorem 4.1 (i) with the $\chi^2$-divergence, we construct the distribution $P^*$ in (A31) and obtain the following upper bound on the outlier robust risk $\mathcal{R}_{\mathrm{x}}^{-}(\boldsymbol{\theta}; P_{\mathrm{train}}^{-}, \delta, \epsilon)$:

$$
\begin{aligned}
\mathcal{R}_{\mathrm{x}}^{-}(\boldsymbol{\theta}; P_{\mathrm{train}}^{-}, \delta, \epsilon) &\leq \mathbb{E}_{P^*}\left\{\mathcal{S}_{\boldsymbol{\theta}}(\mathrm{X}^-; \mathrm{x})\right\} + \sqrt{2\delta \mathbb{V}_{P^*}\left\{\mathcal{S}_{\boldsymbol{\theta}}(\mathrm{X}^-; \mathrm{x})\right\}} \\
&= \Re_1 + \sqrt{2\delta(\Re_2 - \Re_1^2)},
\end{aligned}
$$

where $\Re_1$ and $\Re_2$ are given in (A32) and (A33), respectively.

**Proof of (iii) with $d$ set as the $p$-Wasserstein distance:**

If the discrepancy metric $d$ in (3) is chosen as the $p$-Wasserstein distance, by (4) and Lemma 4, we have that

$$
\begin{aligned}
\mathcal{R}_{\mathrm{x}}^{-}(\boldsymbol{\theta}; P_{\mathrm{train}}^{-}, \delta, \epsilon) &= \inf_{P' \in \mathcal{P}_p(\mathcal{X})}\left\{\mathcal{R}_{\mathrm{x}}^{-}(\boldsymbol{\theta}; P', \delta) : \exists \widetilde{P}' \in \mathcal{P}_p(\mathcal{X}) \; s.t. \; P_{\mathrm{train}}^{-} = (1 - \epsilon)P' + \epsilon \widetilde{P}'\right\}. \\
&\leq \inf_{P' \in \mathcal{P}_p(\mathcal{X})}\left\{\mathbb{E}_{P'}\left\{\mathcal{S}_{\boldsymbol{\theta}}(\mathrm{X}^-; \mathrm{x})\right\} + \delta\left\{\mathbb{E}_{P'}\|\nabla\mathcal{S}_{\boldsymbol{\theta}}(\mathrm{X}^-; \mathrm{x})\|_*^q\right\}^{1/q} + \mathcal{O}(\delta^{2 \wedge p}) : \right. \\
&\qquad\qquad \left. \exists \widetilde{P}' \in \mathcal{P}_p(\mathcal{X}) \; s.t. \; P_{\mathrm{train}}^{-} = (1 - \epsilon)P' + \epsilon \widetilde{P}'\right\}.
\end{aligned}
$$

Similar to the proof of Theorem 4.1 (i) with the $\chi^2$-divergence, we construct the distribution $P^*$ in (A31) and obtain the following upper bound on the outlier robust risk $\mathcal{R}_{\mathrm{x}}^{-}(\boldsymbol{\theta}; P_{\mathrm{train}}^{-}, \delta, \epsilon)$:

$$
\begin{aligned}
\mathcal{R}_{\mathrm{x}}^{-}(\boldsymbol{\theta}; P_{\mathrm{train}}^{-}, \delta, \epsilon) &\leq \mathbb{E}_{P^*}\left\{\mathcal{S}_{\boldsymbol{\theta}}(\mathrm{X}^-; \mathrm{x})\right\} + \delta\left\{\mathbb{E}_{P^*}\|\nabla\mathcal{S}_{\boldsymbol{\theta}}(\mathrm{X}^-; \mathrm{x})\|_*^q\right\}^{1/q} \\
&= \Re_1 + \delta\left\{\mathbb{E}_{P^*}\|\nabla\mathcal{S}_{\boldsymbol{\theta}}(\mathrm{X}^-; \mathrm{x})\|_*^q\right\}^{1/q},
\end{aligned}
$$

where $\Re_1$ is given in (A32).

## A.4 PROOF OF THEOREM 4.2

We complete the proof following the deviations for Theorem 3.2 of Guo et al. (2024). By Lemma 1, when the $p$-Wasserstein distance with $0 - 1$ cost is used to construct the uncertainty set, the robust

risk $\mathcal{R}_x^+(p; x^+, \delta)$ in (5) for positive example $x^+$, can be equivalently written as:

$$\mathcal{R}_x^+(p; x^+, \delta) = \sup_{q^+ \in \Gamma_\delta(p^+)} \langle q^+, -p \rangle$$

$$= \sup \left[ \mathbb{E}_{\mathbf{Y} \sim q^+} \left\{ \sum_{k=1}^K -p_k \mathbb{1}(\mathbf{Y} = k) \right\} : W_p(p^+, q^+) \leq \delta \right]$$

$$= \inf_{\gamma \geq 0} \left( \gamma \delta^p + \mathbb{E}_{\mathbf{Y} \sim p^+} \left\{ \sup_{y' \in [K]} \left[ \left\{ \sum_{k=1}^K -p_l \mathbb{1}(y' = k) \right\} - \gamma \left\{ \mathbb{1}(y' = \mathbf{Y}) \right\}^p \right] \right\} \right)$$

$$= \inf_{\gamma \geq 0} \left[ \gamma \delta^p + \sum_{j=1}^K p_j^+ \max \left\{ -p_1 - \gamma, \ldots, -p_{j-1} - \gamma, -p_j, -p_{j-1} - \gamma, \ldots, -p_K - \gamma \right\} \right]$$

$$= \inf_{\gamma \geq 0} \left[ \gamma \delta^p + \sum_{j=1}^K p_j^+ \max \left\{ 1 - p_1 - \gamma, \ldots, 1 - p_{j-1} - \gamma, 1 - p_j, 1 - p_{j-1} - \gamma, \ldots, \right. \right.$$

$$\left. \left. 1 - p_K - \gamma \right\} \right] - 1$$

$$\triangleq \inf_{\gamma \geq 0} \left\{ \mathsf{h}(\gamma; p) \right\} - 1,$$

where $\mathsf{h}(\gamma; p) \triangleq \gamma \delta^p + \sum_{j=1}^K p_j^+ \max \left\{ 1 - p_1 - \gamma, \ldots, 1 - p_{j-1} - \gamma, 1 - p_j, 1 - p_{j-1} - \gamma, \ldots, 1 - p_K - \gamma \right\}$. Consequently, the minimax problem (5) can be equivalently expressed as:

$$\inf_{p \in \Delta^{K-1}} \inf_{\gamma \geq 0} \left\{ \mathsf{h}(\gamma; p) \right\} - 1,$$

which is a special case of the optimization problem in Theorem 3.2 of Guo et al. (2024), where the constant term $-1$ has no effect on the optimal solution. Thus, Theorem 4.2 follows directly from Theorem 3.2 of Guo et al. (2024).

## B  EXPERIMENTAL DETAILS

**Source Models.**    For the source models, we use those provided by Liang et al. (2020) and Yang et al. (2021a) for the Office-Home and VisDA2017 datasets. Since no open-source models are available for Office-31 and DomainNet-126, we train the source models ourselves using the training methodologies from SHOT (Liang et al., 2020) and C-SFDA (Karim et al., 2023), respectively.

**Target Adaptation Training.**    We train both the model backbone and classifier during the adaptation process, primarily following the SHOT (Liang et al., 2020) and AaD (Yang et al., 2022) setups. For optimization, we use SGD with a momentum of 0.9 and a weight decay of $1e^{-3}$. We also use the Nesterov update method. The initial learning rate for the bottleneck and classification layers is set to 0.001 across all datasets. For the backbone models, the initial learning rates are set as follows: $5e^{-4}$ for Office-Home, $1e^{-4}$ for DomainNet-126 and Office-31, and $5e^{-5}$ for VisDA2017. We use the same learning rate scheduler as Liang et al. (2020) for the Office-Home and DomainNet-126 datasets. The batch size is 64 for all datasets. We train for 30 epochs on VisDA2017 and 45 epochs on Office-Home, Office-31, and DomainNet-126. All experiments are run on a single 32GB V100 or 40GB A100 GPU.

**Hyperparameters Selection.** In SFDA, hyperparameter selection presents a significant challenge due to the lack of labeled target data and the distribution shift between domains. In our experiments, we follow the common pipeline for hyperparameter tuning in the literature (e.g., Yang et al. (2022); Hwang et al. (2024)), and employ the SND (Soft Neighborhood Density) score (Saito et al., 2021) and sensitivity analysis to guide the hyperparameter selection. Notably, most hyperparameters in our method do not require intensive tuning, and their choices can be guided by our theoretical analysis outlined below.

Our UCon-SFDA method consists of three main components: the basic contrastive loss $\mathcal{L}_{\mathrm{CL}}$, the dispersion control term $\mathcal{L}_{\mathrm{DC}}^-$, and the partial label term $\mathcal{L}_{\mathrm{PL}}^+$. Given the complexity of the parameter

space, we simplify the hyperparameter selection process by avoiding exhaustive consideration of all parameter combinations. Instead, we adopt a **sequential**, **incremental** approach to tune the parameters for the three loss terms, one at a time.

First, for the hyperparameters in the $\mathcal{L}_{\mathrm{CL}}$ terms (first three columns in Table B1), including the number of positive samples $\kappa$, the decay exponent $\beta$ for the negative term, and the negative sample loss coefficient $\lambda_{\mathrm{CL}}^{-}$, we largely follow the configurations used in Yang et al. (2022) and Hwang et al. (2024). As in previous works, we directly set $\lambda_{\mathrm{CL}}^{-}$ to 1. For datasets with more classification categories, such as Office-Home, Office, and DomainNet-126, where noise in negative samples is less pronounced, we use a smaller decay exponent to enhance the impact of true-negative samples during adaptation. In contrast, for VisDA, which contains only 12 classes with a batch size of 64, we apply a faster decay rate to mitigate the influence of false-negative samples.

Next, we consider the hyperparameter associated with the dispersion term, $\lambda_{\mathrm{DC}}$. In our initial experimental trials, we set this value to either 0.5 or 1, based on a balance between the loss terms, $\mathcal{L}_{\mathrm{CL}}^{+}$ and $\mathcal{L}_{\mathrm{DC}}^{-}$, and the sensitivity analysis of hyperparameters.

Finally, for the hyperparameters $\lambda_{\mathrm{PL}}$, $K_{\mathrm{PL}}$, and $\tau$ in the partial label loss, we also perform the basic sequential tuning under the guidance of theoretical insights. According to the proposed algorithm, we use $\tau$ to select uncertain data points and merge the top-$K_{\mathrm{PL}}$ predicted classes into the partial label set for each selected data point. Theoretically, a smaller $\tau$ (yet naturally larger than 1) represents a more uncertain set. As we want to apply the partial label loss only to uncertain data points and avoid the introduction of additional label uncertainty for more confident data points, we consider a value in $\{1.1, 1.3, 1.5\}$ for $\tau$. We find that $\tau = 1.1$ is sufficient for achieving promising performance, except for simpler tasks with a high initial prediction accuracy, such as Office-31. Next, the value of the partial label number $K_{\mathrm{PL}}$ should be determined based on the algorithm and the number of categories in the dataset. Generally, a small $K_{\mathrm{PL}}$ is preferred, as the partial label set is gradually enlarged with each epoch. A large $K_{\mathrm{PL}}$ could result in an overly large partial label set, potentially introducing more uncertainty. Empirically, we evaluate $K_{\mathrm{PL}} \in \{1, 2, 3\}$, and find that $K_{\mathrm{PL}} = 2$ performs well for most datasets, except for VisDA2017, whose total number of classes is only 12 and $K_{\mathrm{PL}} = 1$ is sufficient. Finally, we tune $\lambda_{\mathrm{PL}}$ by considering $\lambda_{\mathrm{PL}} \in \{0.001, 0.01, 0.05, 0.1\}$ and select the best-performing value based on the guidance of the hyperparameter sensitivity analyses.

The final selected parameter values used in our experiments are summarized in Table B1, which are obtained by a relatively straightforward tuning process conducted on a subspace of hyperparameters. We note that more refined tuning over the full combinatorial hyperparameter space can further enhance the performance of our algorithm; additional analysis on the sensitivity of these hyperparameters is provided in Appendix C.5.

Table B1: Hypermaraters on different datasets

| Dataset | $\kappa$ | $\lambda_{\mathrm{CL}}^{-}$ | $\beta$ | $\lambda_{\mathrm{DC}}$ | $\lambda_{\mathrm{PL}}$ | $K_{\mathrm{PL}}$ | $\tau$ |
|---|---|---|---|---|---|---|---|
| Office-31 | 3 | 1 | 1 | 1 | 0.05 | 2 | 1.3 |
| Office-Home | 3 | 1 | 0 | 0.5 | 0.001 | 2 | 1.1 |
| Office-Home (partial set) | 5 | 1 | 0.75 | 1 | 0.1 | 2 | 1.1 |
| VisDA2017 | 5 | 1 | 5 | 1 | 0.01 | 1 | 1.1 |
| VisDA-RUST | 3 | 1 | 5 | 0.5 | 0.1 | 2 | 1.1 |
| DomainNet-126 | 2 | 1 | 0.75 | 0.5 | 0.1 | 2 | 1.1 |

**Algorithm.** The overall description of adaptation process with our UCon-SFDA method is shown in Algorithm 1

---

**Algorithm 1: UCon-SFDA** - Uncertainty-Controlled Source-Free Domain Adaptation

---

**Input:** Pre-Trained Source Model: $f_{\mathrm{S}}(\mathrm{x}; \boldsymbol{\theta}_{\mathbf{S}})$

         Target Data: $\mathcal{D}_{\mathrm{T}} \triangleq \{\mathrm{x}_i^{\mathrm{T}}\}_{i=1}^{N_{\mathrm{T}}}$

         Training Epochs: T

1 // Initialization Process
2 Initialize a target model $f_{\mathrm{T}}(\mathrm{x}; \boldsymbol{\theta}_0) = f_{\mathrm{S}}(\mathrm{x}; \boldsymbol{\theta}_{\mathbf{S}})$
3 Construct feature bank $\mathscr{Z}$ and predicted score bank $\mathscr{F}$ as described in Yang et al. (2022)
4 Initialize the partial label bank $\mathscr{Y}_{\mathrm{PL}}$ and uncertainty sample bank $\mathscr{U}$ as proposed in Section 4.4
5 // Training/Adaptation Process
6 **for** *epoch=1* **to** T **do**
7      **for** *iterations t = 1,2,3,...* **do**
8          Forward Propagation: obtain feature $\mathbf{z}_i$, predicted probabilities $f_{\mathrm{T}}(\mathrm{x}_i; \boldsymbol{\theta}_t)$ and
9                      $f_{\mathrm{T}}(\mathtt{AUG}(\mathrm{x}_i); \boldsymbol{\theta}_t)$ for each sample $\mathrm{x}_i$ in mini-batch $\mathcal{B}$
10          Bank Refresh: update $\mathscr{Z}$ and $\mathscr{F}$ using $\mathbf{z}_{\mathcal{B}}$ and $f_{\mathrm{T}}(\mathrm{x}_{\mathcal{B}}; \boldsymbol{\theta}_t)$ as described in
11                  Yang et al. (2022); update $\mathscr{Y}_{\mathrm{PL}}$ and $\mathscr{U}$ as proposed in Section 4.4
12          Compute Negative Uncertainty Control Loss $\mathcal{L}_{\mathrm{UCon}}^{-}$ in Eq. (7) using $f_{\mathrm{T}}(\mathrm{x}_{\mathcal{B}}; \boldsymbol{\theta}_t)$ and
          $f_{\mathrm{T}}(\mathtt{AUG}(\mathrm{x}_{\mathcal{B}}); \boldsymbol{\theta}_t)$
13          Compute Positive Uncertainty Control Loss $\mathcal{L}_{\mathrm{UCon}}^{+}$ in Eq. (8) using $\mathscr{Z}, \mathscr{F}, \mathscr{Y}_{\mathrm{PL}}$ and $\mathscr{U}$
14          Compute the total Uncertainty Control Source-Free Domain Adaptation Loss
          $\mathcal{L}_{\mathrm{UCon-SFDA}} = \mathcal{L}_{\mathrm{UCon}}^{+} + \mathcal{L}_{\mathrm{UCon}}^{-}$
15          Update the parameters of $f_{\mathrm{T}}(\boldsymbol{\theta}_t)$ via $\mathcal{L}_{\mathrm{UCon-SFDA}}$
16      **end for**
17 **end for**
**Output:** Target Adapted Model $f_{\mathrm{T}}(\mathrm{x}_i; \boldsymbol{\theta}_{\mathrm{T}})$

---

## C ADDITIONAL EXPERIMENTAL RESULTS

### C.1 EXPERIMENTAL RESULT ON OFFICE-HOME

The experimental results on the Office-Home dataset are reported in Table C2.

Table C2: Classification accuracy (%) on the Office-Home dataset (ResNet-50)

| Method | Ar→Cl | Ar→Pr | Ar→Rw | Cl→Ar | Cl→Pr | Cl→Rw | Pr→Ar | Pr→Cl | Pr→Rw | Rw→Ar | Rw→Cl | Rw→Pr | Avg. |
|---|---|---|---|---|---|---|---|---|---|---|---|---|---|
| SHOT (Liang et al., 2020) | 57.1 | 78.1 | 81.5 | 68.0 | 78.2 | 78.1 | 67.4 | 54.9 | 82.2 | 73.3 | 58.8 | 84.3 | 71.8 |
| A²Net (Xia et al., 2021) | 58.4 | 79.0 | 82.4 | 67.5 | 79.3 | 78.9 | **68.0** | 56.2 | 82.9 | 74.1 | 60.5 | 85.0 | 72.8 |
| G-SFDA (Yang et al., 2021b) | 57.9 | 78.6 | 81.0 | 66.7 | 77.2 | 77.2 | 65.6 | 56.0 | 82.2 | 72.0 | 57.8 | 83.4 | 71.3 |
| NRC (Yang et al., 2021a) | 57.7 | 80.3 | 82.0 | 68.1 | 79.8 | 78.6 | 65.3 | 56.4 | 83.0 | 71.0 | 58.6 | 85.6 | 72.2 |
| CPGA (Qiu et al., 2021) | 59.3 | 78.1 | 79.8 | 65.4 | 75.5 | 76.4 | 65.7 | 58.0 | 81.0 | 72.0 | **64.4** | 83.3 | 71.6 |
| CoWA-JMDS (Lee et al., 2022) | 56.9 | 78.4 | 81.0 | 69.1 | 80.0 | **79.9** | 67.7 | 57.2 | 82.4 | 72.8 | 60.5 | 84.5 | 72.5 |
| DaC (Zhang et al., 2022) | 59.1 | 79.5 | 81.2 | 69.3 | 78.9 | 79.2 | 67.4 | 56.4 | 82.4 | 74.0 | 61.4 | 84.4 | 72.8 |
| C-SFDA (Karim et al., 2023) | 60.3 | 80.2 | **82.9** | 69.3 | 80.1 | 78.8 | 67.3 | 58.1 | **83.4** | 73.6 | 61.3 | 86.3 | 73.5 |
| AaD (Yang et al., 2022) | 59.3 | 79.3 | 82.1 | 68.9 | 79.8 | 79.5 | 67.2 | 57.4 | 83.1 | 72.1 | 58.5 | 85.4 | 72.7 |
| I-SFDA (Mitsuzumi et al., 2024) | 60.7 | 78.9 | 82.0 | **69.9** | 79.5 | 79.7 | 67.1 | 58.8 | 82.3 | **74.2** | 61.3 | **86.4** | 73.4 |
| **UCon-SFDA (Ours)** | **61.5** | **80.5** | 82.1 | 69.3 | **80.8** | 78.7 | 67.0 | **62.2** | 82.0 | 72.2 | 61.9 | 85.5 | **73.6** |

### C.2 PARTIAL LABEL SET EVALUATION

We conduct the self-prediction, partial label set, and neighbor label set evaluations across all 12 tasks on the office-home dataset. The results of self-prediction are shown in Figure C1 to Figure C4, and the results of partial label set and neighbor set comparison are shown in Figure C5 to Figure C8.

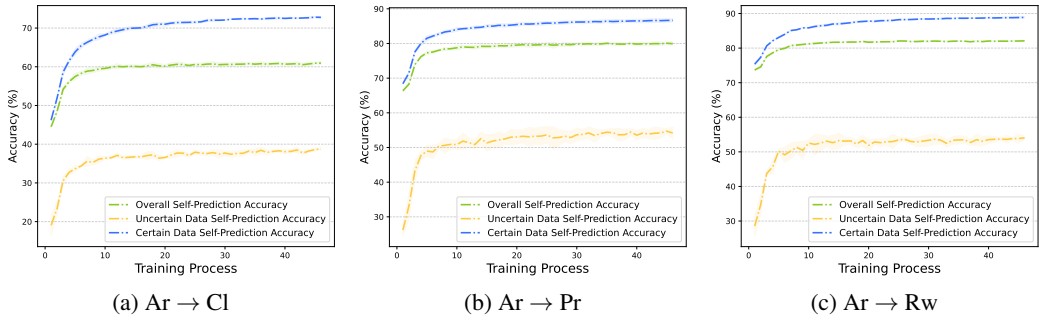

(a) Ar → Cl  (b) Ar → Pr  (c) Ar → Rw

Figure C1: *Self-prediction accuracy among different data certainty levels on Office-Home dataset with source domain Ar*

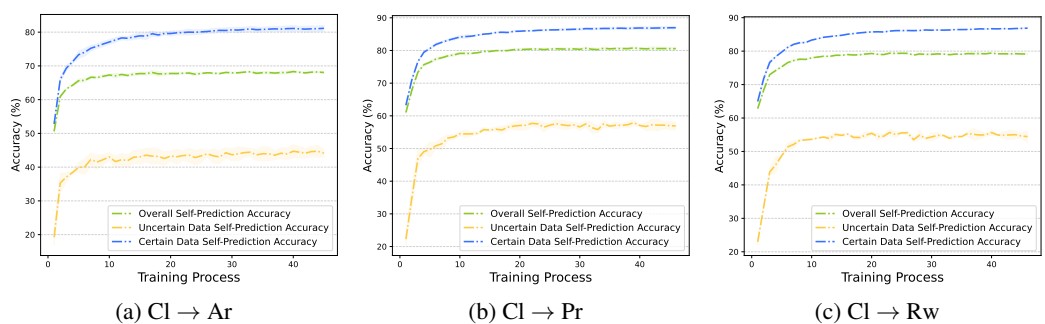

(a) Cl → Ar  (b) Cl → Pr  (c) Cl → Rw

Figure C2: *Self-prediction accuracy among different data certainty levels on Office-Home dataset with source domain Cl*

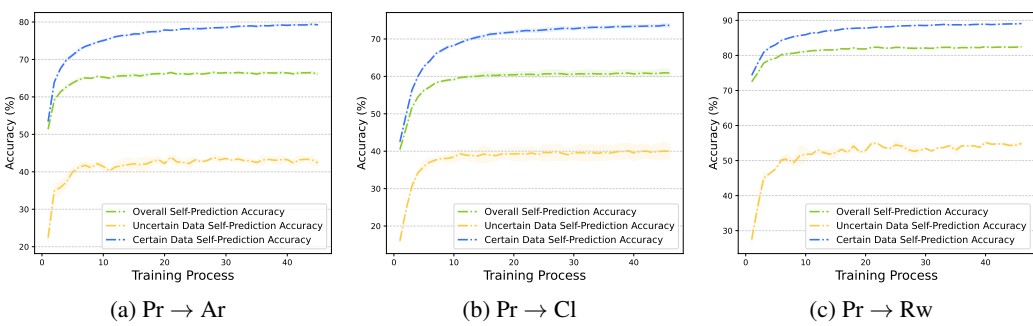

(a) Pr → Ar  (b) Pr → Cl  (c) Pr → Rw

Figure C3: *Self-prediction accuracy among different data certainty levels on Office-Home dataset with source domain Pr*

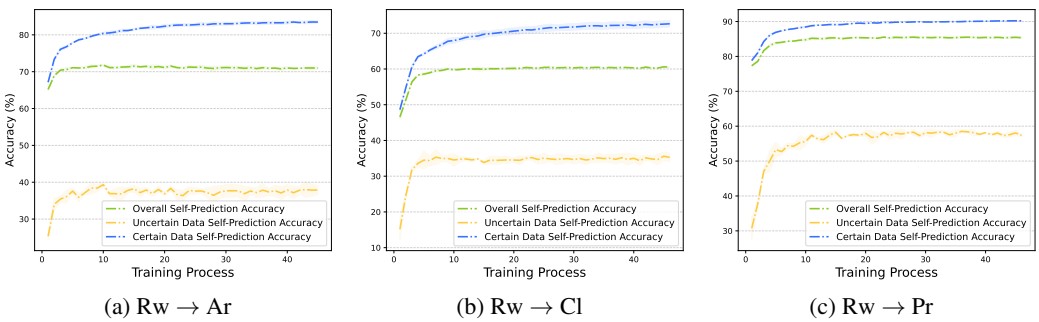

Figure C4: *Self-prediction accuracy among different data certainty levels on Office-Home dataset with source domain Rw*

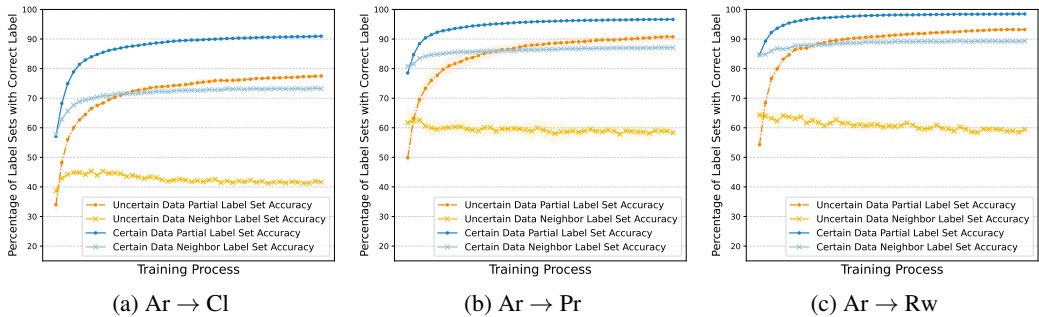

Figure C5: *Label set correctness among different data certainty levels on Office-Home dataset with source domain Ar*

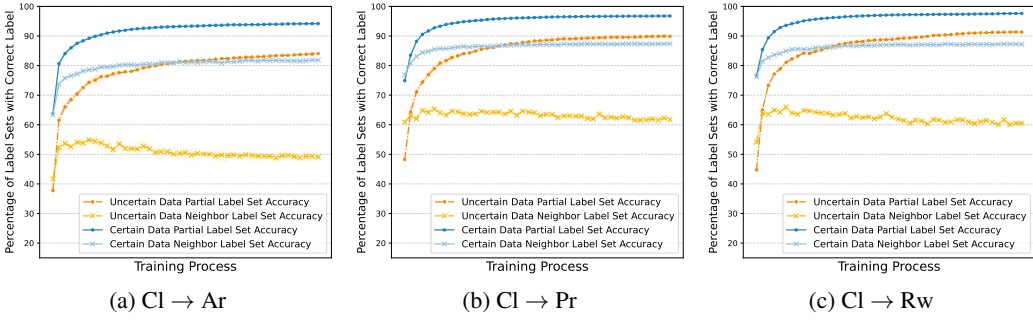

Figure C6: *Label set correctness among different data certainty levels on Office-Home dataset with source domain Cl*

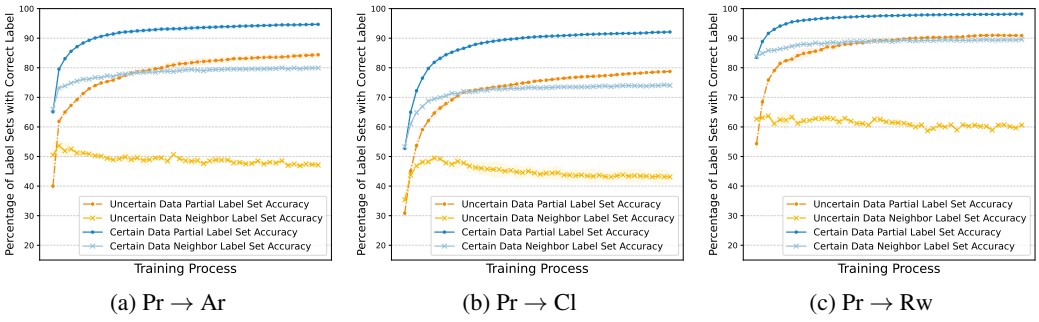

Figure C7: *Label set correctness among different data certainty levels on Office-Home dataset with source domain Pr*

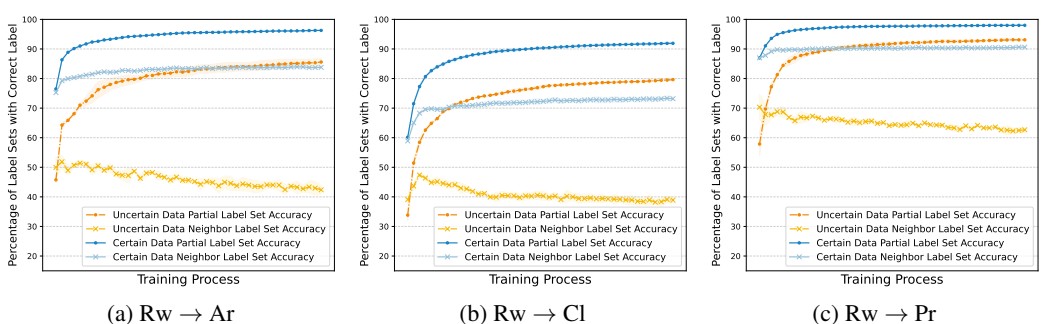

Figure C8: *Label set correctness among different data certainty levels on Office-Home dataset with source domain Rw*

## C.3    DATA AUGMENTATION IN SFDA

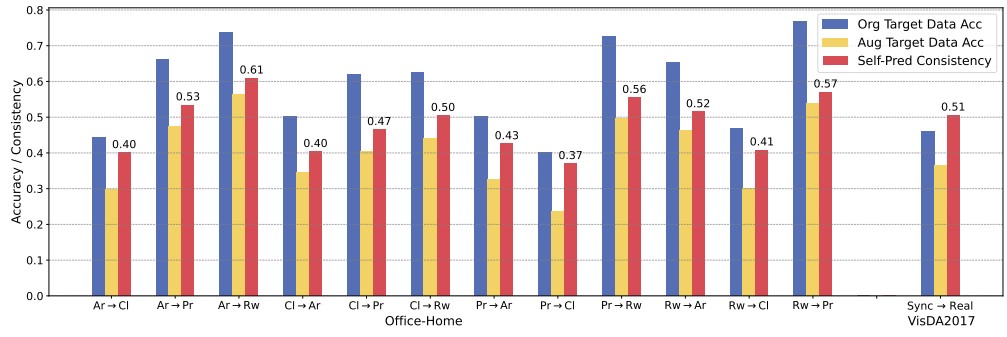

Figure C9: *Inconsistency between the prediction results between the anchor image and its augmented view by source model*

The data augmentation technique has been used in SFDA to improve prediction consistency, enhance the target model's generalizability, and control feature space variance (Karim et al., 2023; Mitsuzumi et al., 2024; Xu et al., 2025). However, these methods intuitively treat the augmented views as positive samples of the original image, without considering the model's initial representational or predictive capacity on these data. Moreover, they often overlook the fact that such data are more likely to be negative samples in terms of the self-predicted pseudo-label (Pu et al., 2021).

Here, we evaluate the prediction accuracy and consistency of the original target data and their augmented version by applying the source model to Office-Home and VisDA-2017. The consistency

is defined as:

$$\text{CONSISTENCY} \triangleq \sum_{i=1}^{N_{\mathrm{T}}} \mathbb{1}\left\{ f_{\mathrm{S}}(\mathrm{x}_i; \boldsymbol{\theta}_{\mathbf{S}}) = f_{\mathrm{S}}(\text{AUG}(\mathrm{x}_i); \boldsymbol{\theta}_{\mathbf{S}}) \right\}.$$

As shown in Figure C9, the source model exhibits a low accuracy in predicting the augmented data and demonstrates a high inconsistency between the predictions for the anchor data and its augmented versions. This experimental result is counterintuitive. It empirically explains why directly using the augmented predictions as additional labels or supervisory signals sometimes fails to effectively improve SFDA performance and may even have a negative impact.

## C.4 VARIANCE CONTROL EFFECT

We evaluate the dispersion control effect achieved by our augmentation-based $\mathcal{L}_{\mathrm{DC}}^{-}$ across all 12 tasks on the office-home dataset. The results are shown in Figure C10 to Figure C13. The consistent dispersion reduction achieved validates the effectiveness of our proposed method.

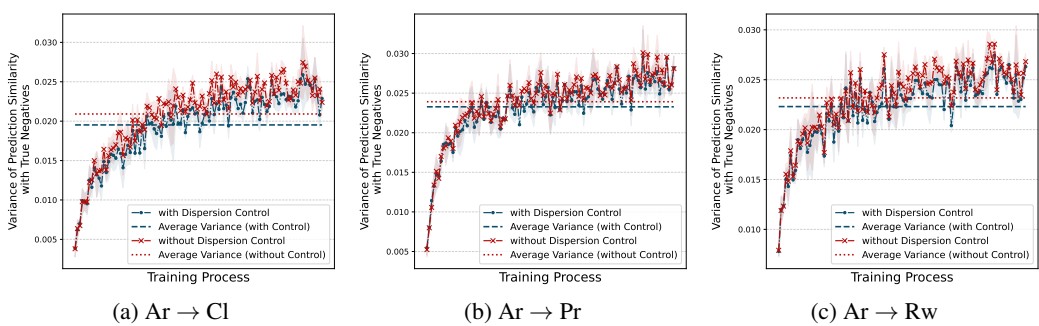

(a) Ar → Cl      (b) Ar → Pr      (c) Ar → Rw

Figure C10: *Dispersion control loss effect on Office-Home dataset with source domain Ar*

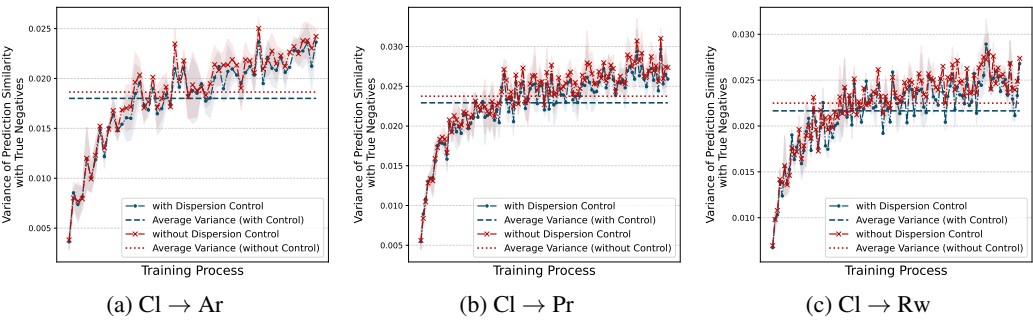

(a) Cl → Ar      (b) Cl → Pr      (c) Cl → Rw

Figure C11: *Dispersion control loss effect on Office-Home dataset with source domain Cl*

## C.5 SENSITIVITY ANALYSES OF HYPERPARAMETERS

To further understand the performance of the proposed method, we conduct comprehensive experiments to study the sensitivity of our method to different choices of hyperparameters involved in our algorithm. While we primarily use the hyperparameter configurations from previous works (Yang et al., 2022; Hwang et al., 2024) for $\lambda_{\mathrm{CL}}^{-}$, $\kappa$ and $\beta$, we also investigate the sensitivity of our method relative to different choices of $\beta$, $K_{\mathrm{PL}}$, $\tau$, $\lambda_{\mathrm{PL}}$ and $\lambda_{\mathrm{DC}}$. The experimental results are summarized in Figure C14(a), (b), (c), Figure C15 and Figure C16, respectively.

Specifically, in Figure C14(a)-(c), the solid lines represent the accuracy of different methods with respect to different values of $\beta$, $K_{\mathrm{PL}}$, and $\tau$. In Figure C14(b)-(c), we add the dashed horizontal lines to indicate the performance on different datasets without the partial label loss for a clear comparison.

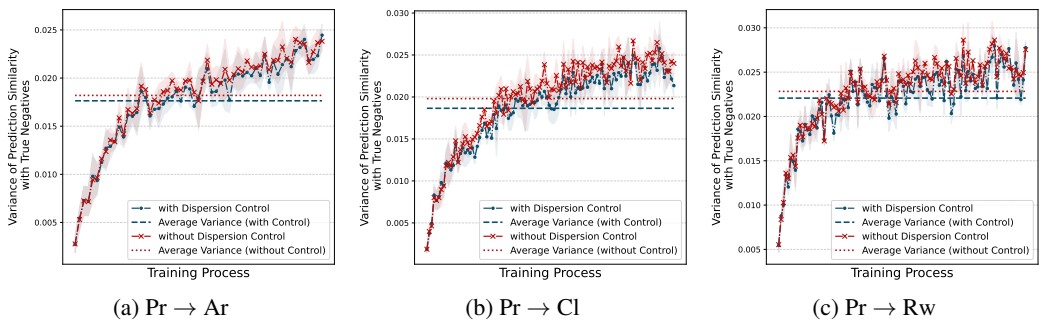

Figure C12: *Dispersion control loss effect on Office-Home dataset with source domain Pr*

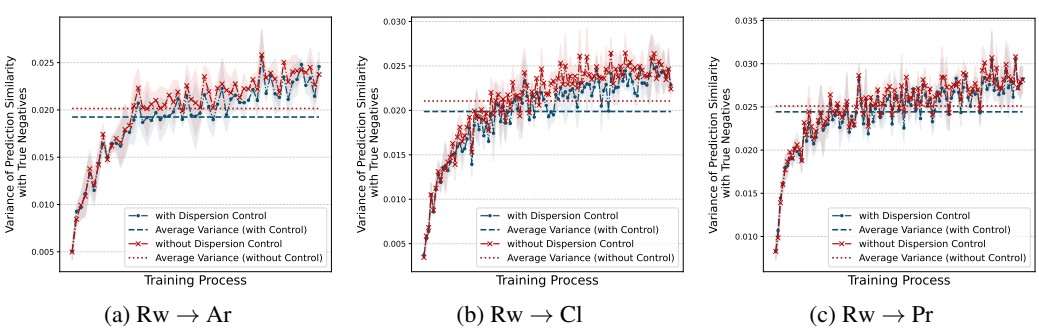

Figure C13: *Dispersion control loss effect on Office-Home dataset with source domain Rw*

In Figures C15- C16, the blue, red, and yellow lines represent the accuracy on the target dataset, the accuracy on the small evaluation set, and the SND score, respectively. The shaded regions correspond to the results reported in the main text and the associated parameter values. For Figures C14- C16, except for the parameter values that vary along the x-axis, all other parameters are set according to Table B1.

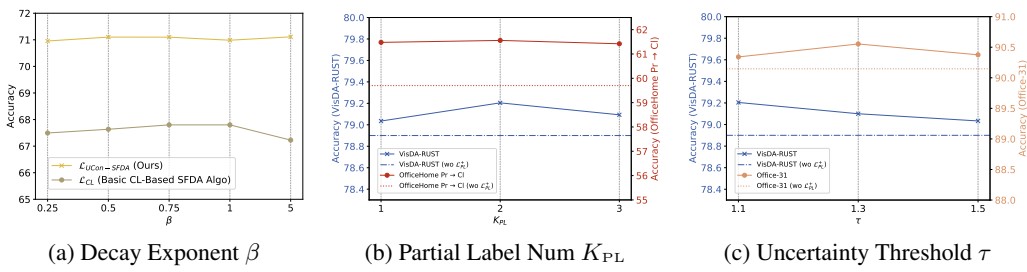

Figure C14: *Sensitivity analysis of the proposed method relative to different values of hyperparameters $\beta$, $K_{\mathrm{PL}}$, and $\tau$. In the legend, "wo" is the abbreviation for "without".*

**Decay Exponent $\beta$.** Figure C14(a) reveals that the dispersion control term can help mitigate the sensitivity of $\beta$ in contrastive learning based SFDA algorithms. Specifically, we compare the performance of an SFDA task (R to P on DomainNet-126 dataset) using our proposed method (UCon-SFDA) against the basic contrastive learning approach introduced in Yang et al. (2022). Beyond providing stable performance improvements, our method demonstrates reduced sensitivity to the hyperparameter $\beta$, benefiting from the uncertainty-controlling regularizations.

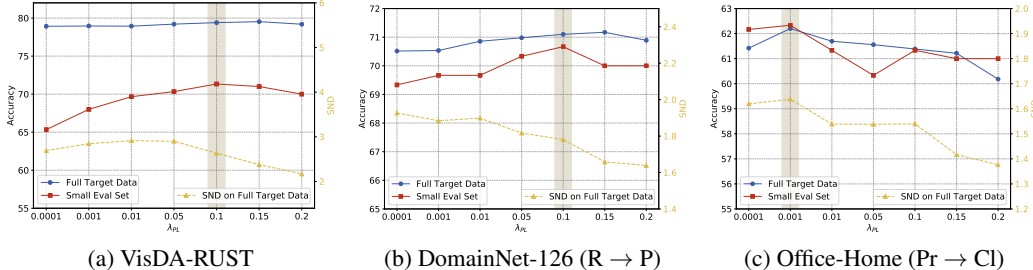

| (a) VisDA-RUST | (b) DomainNet-126 (R → P) | (c) Office-Home (Pr → Cl) |

Figure C15: *Sensitivity analysis of dispersion control loss coefficient $\lambda_{\mathrm{PL}}$. Different colors represent various criteria for hyperparameter selection, while the shaded area indicates the parameter values chosen corresponding to the results reported in the main text.*

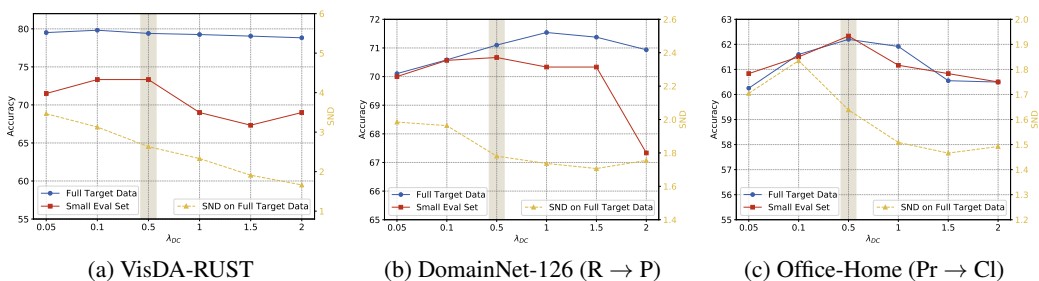

| (a) VisDA-RUST | (b) DomainNet-126 (R → P) | (c) Office-Home (Pr → Cl) |

Figure C16: *Sensitivity analysis of dispersion control loss coefficient $\lambda_{\mathrm{DC}}$. Different colors represent various criteria for hyperparameter selection, while the shaded area indicates the parameter values chosen corresponding to the results reported in the main text.*

**Partial Label Number $K_{\mathrm{PL}}$ and Uncertainty Threshold $\tau$.** Figure C14(b) and (c) illustrate the sensitivity of our method to the partial label number $K_{\mathrm{PL}}$ and uncertainty threshold $\tau$. By comparing the performance variations on VisDA-RUST, Office-31, and Office-Home (Pr to Cl task) under different $K_{\mathrm{PL}}$ and $\tau$, we observe that the accuracy of our method is not significantly affected by varying values of $K_{\mathrm{PL}}$ and $\tau$. Moreover, the performance improvements by the partial label loss are both evident and stable, as shown by the comparison between the solid and dashed lines.

**Partial Labeling Term Coefficient $\lambda_{\mathrm{CL}}$ and Dispersion Control Term Coefficient $\lambda_{\mathrm{DC}}$.** As shown in Figures C15- C16, we conduct an ablation study with finer-grained variations of $\lambda_{\mathrm{CL}}$ and $\lambda_{\mathrm{DC}}$ on three datasets to access sensitivity of the experimental results. Relative to the blue lines, the adaptation performance remains stable and robust across different values of these two hyperparameters, with the regions of optimal performance being well-concentrated.

**Additional Insights for Advanced and Practical Hyperparameter Selection Strategies.** Hyperparameter tuning in SFDA poses significant challenges due to the lack of target labels and substantial distribution shifts across domains. In our experiments, we find that SND scores often fail to correlate consistently with performance on the full target dataset. Moreover, sensitivity analysis based on the full target data incurs high computational costs, making it less feasible for real-world applications. To overcome these limitations, we explore a novel small evaluation set-based method. Specifically, we randomly select a subset (300 data points) from the full unlabeled target data (typically containing 5k-50k data points), manually label it, and create a pseudo-validation set. Hyperparameters are subsequently selected based on their performance on this small evaluation set. While this approach requires some manual annotation, the amount of labeled data needed is minimal, making it both practical and effective for real-world scenarios, while improving the accuracy of hyperparameter selection.

Figures C15 and C16 demonstrate that the performance of our method on the small human-labeled evaluation set (red lines) aligns more closely with the desired model performance (blue lines). In

contrast, the SND score (yellow lines), which is based on feature space similarity and self-prediction entropy, sometimes fails to identify the optimal hyperparameters.

**Better Performance with Finer-Grained Hyperparameter Ranges.** Refining the parameter selection range (as shown Figure C15(a)-(b)) or adopting a different tuning order (e.g., tuning the partial label term first, followed by the dispersion control term, as shown in Figure C16(a)-(b)) can achieve even better results, as indicated by the highest points on the blue lines. For instance, while we initially report the UCon-SFDA performance of 79.4 on VisDA-RUST (with $L_{\text{PL}} = 0.1$ and $L_{\text{DC}} = 0.5$), we find that using a slightly smaller $L_{\text{DC}} = 0.1$ improves its performance to 79.82. These findings demonstrate that satisfactory performance of our approach does not depend on excessive hyperparameter tuning, and further highlights the robustness and effectiveness of our algorithm.

## C.6 Different Losses for Dispersion Control term

We evaluate the performance of the dispersion control term under different similarity metrics between an anchor data point and its augmented version, $\mathbb{d}_{\boldsymbol{\theta}} \left( \text{AUG} \left( \mathrm{x}_i \right), \mathrm{x}_i \right)$, in Eq. (7).

Specifically, for Eq. (7) in the main text, we define:

$$\mathbb{d}_{\boldsymbol{\theta}} \left( \text{AUG} \left( \mathrm{x}_i \right), \mathrm{x}_i \right) \triangleq \langle f(\mathrm{x}_i; \boldsymbol{\theta}), \log f \left( \text{AUG} \left( \mathrm{x}_i \right); \boldsymbol{\theta} \right) \rangle.$$

To further validate the role of data augmentation from the perspective of negative sampling uncertainty, we experiment with different similarity metrics, including the direct dot product and the $L^2$ norm, respectively given by

$$\mathbb{d}_{\boldsymbol{\theta},\text{dot}} \left( \text{AUG} \left( \mathrm{x}_i \right), \mathrm{x}_i \right) \triangleq \langle f(\mathrm{x}_i; \boldsymbol{\theta}), f \left( \text{AUG} \left( \mathrm{x}_i \right); \boldsymbol{\theta} \right) \rangle,$$

and

$$\mathbb{d}_{\boldsymbol{\theta},\text{L}^2} \left( \text{AUG} \left( \mathrm{x}_i \right), \mathrm{x}_i \right) \triangleq \| f(\mathrm{x}_i; \boldsymbol{\theta}) - f \left( \text{AUG} \left( \mathrm{x}_i \right); \boldsymbol{\theta} \right) \|^2.$$

Additional experimental results, reported in Table C3, demonstrate the importance of treating data augmentations as negative samples as well as the effectiveness of the proposed dispersion control term. Furthermore, while the proposed $\mathbb{d}_{\boldsymbol{\theta}}$ achieves the best performance across most datasets, other loss formulations also present comparable results. These experimental observations provide guidance on effectively leveraging data augmentations in SFDA and verify the generalizability of our algorithm.

Table C3: Classification accuracy (%) under different distance measurements in dispersion control term. **Bold** text indicates the best results, and underlined text represents results that outperform the baseline.

| Methods | Office-Home (Pr → Cl) | VisDA-RUST | DomainNet126 (R → P) |
|---|---|---|---|
| $\mathcal{L}_{\text{CL}}$ | 57.90 | 75.50 | 67.80 |
| $\mathcal{L}_{\text{CL}} + \mathcal{L}_{\text{DC}}^-$ with $\mathbb{d}_{\boldsymbol{\theta}}$ | 59.70 | **78.90** | **70.30** |
| $\mathcal{L}_{\text{CL}} + \mathcal{L}_{\text{DC}}^-$ with $\mathbb{d}_{\boldsymbol{\theta},\text{dot}}$ | **60.21** | 78.02 | 70.08 |
| $\mathcal{L}_{\text{CL}} + \mathcal{L}_{\text{DC}}^-$ with $\mathbb{d}_{\boldsymbol{\theta},\text{L}^2}$ | 59.14 | 77.77 | 69.34 |

## C.7 Training time and Resource Usage Analysis

To further validate the practical value of our proposed methodology, we conduct the training time and resource usage analysis here.

Compared to the baseline model, AaD (Yang et al., 2022), a widely utilized contrastive learning and memory bank-based SFDA method, our UCon-SFDA introduces explicit data augmentation and an additional partial label bank component. These additions increase both resource usage and computational complexity. However, such costs are consistent with recent trends in the field (Hwang

et al., 2024; Karim et al., 2023; Mitsuzumi et al., 2024), where enhanced resource utilization is commonly accepted to achieve significant performance improvements.

The computational complexity of our approach remains comparable to other modern techniques that leverage data augmentation or consistency regularization. For instance, compared to Karim et al. (2023) and Mitsuzumi et al. (2024), which also incorporate explicit data augmentation during training, our UCon-SFDA avoids relying on additional network structures. Moreover, the partial label bank only incurs a small additional memory overhead that scales linearly with the size of the target domain data, making it practical for real-world SFDA applications. Importantly, our method demonstrates superior performance, as evidenced by the experimental results presented in main text.

Nevertheless, we acknowledge that the explicit data augmentation employed in UCon-SFDA inevitably increases the GPU memory usage, which could present challenges in resource-constrained settings. Although our approach ensures that the additional overhead remains manageable, further algorithmic and implementation-level optimizations could help mitigate this issue. For instance, future work could explore more memory-efficient augmentation techniques, optimize the computational graph during training, or incorporate mixed-precision training. These efforts hold promise for enhancing scalability while maintaining performance.

## D  THEORY-MOTIVATED HYPERPARAMETER DETERMINATION AND AUTOUCON-SFDA

In SFDA problems where neither target domain labels nor a validation set are available, minimizing the numbers of hyperparameters is crucial to ensuring the algorithm's practicality for new tasks. When designing the UCon-SFDA algorithm (as presented in the main paper), we prioritize engineering flexibility and ease of implementation, which lead us to introduce four hyperparameters: $\lambda_{\mathrm{DC}}$, $\lambda_{\mathrm{PL}}$, $K_{\mathrm{PL}}$ and $\tau$. However, three of these hyperparameters have explicit expressions derived from our theoretical results or can be determined based on dataset and source model properties.

Here, we provide a detailed explanation of how theoretical insights can guide the direct selection or derivation of hyperparameters, thereby eliminating the need for manual tuning. Building on these theoretical principles, we propose two enhanced variants (autoUCon-SFDA). Additional experimental results demonstrate that directly using theoretically derived parameters not only simplifies the tuning process but also achieves promising-and in some cases, superior-performance across all benchmarks.

### D.1  THEORETICAL GUIDANCE FOR HYPERPARAMETER DETERMINATION

Based on our theoretical findings, the hyperparameters $\lambda_{\mathrm{DC}}$ in the dispersion control term and $K_{\mathrm{PL}}$, $\tau$ in the partial label term can be directly determined. Specifically,

**Inconsistency Rate $\lambda_{\mathrm{DC}}$.**  As suggested by Theorem 4.1 and Remark 4.2, the dispersion control effect can be achieved by minimizing the negative similarity between the anchor point and its augmented prediction. If the inconsistency rate between anchor points and their associated augmented predictions is high, it indicates greater uncertainty in negative sampling, thus requiring stronger dispersion control. Based on this observation, we propose directly using the model prediction inconsistency rate as the coefficient for the dispersion control term.

**Parameter $K_{\mathrm{PL}}$ ($k_0$ in Theorem 4.2).**  By Theorem 4.2, when the uncertainty set in Eq. ( 5) is defined using the 1-Wasserstein distance, the length of the partial label set, denoted by $K_{\mathrm{PL}}$, can be explicitly determined as $K_{\mathrm{PL}} = k_0$, where $k_0$ is defined as follows:

- If $\frac{1}{K} \geq \frac{1}{k} \sum_{j=1}^{k} p_{(j)}^+ - \frac{1}{k}\delta$ for all $k \in [K-1]$, then we take $k_0 = K$.

- Otherwise, we take the $k_0 \in [K-1]$ that satisfies $\frac{1}{k_0} \sum_{j=1}^{k_0} p_{(j)}^+ - \frac{1}{k_0}\delta \geq \frac{1}{k} \sum_{j=1}^{k} p_{(j)}^+ - \frac{1}{k}\delta$ for all $k \in [K-1]$.

In the formulas above, $K$ represents the number of classes, and $p_{(j)}^+$ denotes the $j$-th largest predicted probability for the considered anchor point. Hence, the length of the partial label set, which can

be directly calculated, is determined by the model's predictions for the anchor point as well as the specific classification task at hand.

**Uncertainty Threshold $\tau$.**   We propose two approaches to distinguish between certain and uncertain label information and determine the uncertainty threshold $\tau$.

- **Statistical Insights Approach**: This approach leverages the properties of the source model and the target data, combined with statistical insights. Specifically, we first use the source model to compute the predicted probabilities for each target data point. Next, we calculate the ratio of the two highest predicted probabilities for all target data points and select the 10th percentile of these ratios as the value of $\tau$. This selection capitalizes on the information about the data distribution and identifies the 10% most uncertain data. The 10th percentile is chosen because it is a widely used measure in statistical research to highlight low-end values. This uncertainty threshold determination method leads to the development of the **autoUCon-SFDA (Stat.)** algorithm.

- **Theoretical Criterion Approach**: Alternatively, we can bypass the ratio of the two highest predicted probabilities and directly apply the criterion outlined in Remark 4.3 to distinguish between certain and uncertain label information. As discussed in Remark 4.3, in the special case where $p_{(1)}^{+} \geq \max\{\frac{1}{K} + \delta, p_{(2)}^{+} + \delta\}$, we refer to it as *certain label information*. Conversely, if this condition is not satisfied, the label information is deemed *uncertain*, and the corresponding data point is added to the uncertain data bank. This uncertainty threshold determination method leads to the development of the **autoUCon-SFDA (Theory)** algorithm.

Building upon the preceding illustrations and different approaches to determining the uncertainty threshold $\tau$, we propose two automated versions of UCon-SFDA: autoUCon-SFDA (Statistics) and autoUCon-SFDA (Theory).

## D.2   Experimental Results of autoUCon-SFDA

Compared with the original UCon-SFDA, autoUCon-SFDA (Statistics) and autoUCon-SFDA (Theory) incorporate the following modifications in the implementation:

1. The original manually tuned hyperparameter $\lambda_{\mathrm{DC}}$ (Orig. $\lambda_{\mathrm{DC}}$) has been replaced by new $\lambda_{\mathrm{DC}}$, which represents the inconsistency ratio between anchor points and their associated augmented predictions, derived by the source model.

2. The original fixed $K_{\mathrm{PL}}$ (Orig. $K_{\mathrm{PL}}$) has been replaced by the calculated $k_0$, which is instance- and task-dependent (class category), self-adaptive during the training process, and computationally efficient.

3. We propose two alternatives for the fixed parameter $\tau$ (Orig. $\tau$):

   - In the statistical insights approach, autoUCon-SFDA (Stat.), $\tau_s$ is computed using the source model and fixed at the beginning of the adaptation process.

   - In the theoretical criterion approach, autoUCon-SFDA (Theory), $\tau_t$ is dynamically calculated based on the uncertain data selected in each epoch.

Table D4: Performance comparisons across different hyper-parameter selection (calculation) methods. **Bold** text indicates the best results.

| Dataset | UCon-SFDA | autoUCon-SFDA (Theory) | autoUCon-SFDA (Stat.) | SOTA Method Performance | SOTA Method |
|---|---|---|---|---|---|
| **Office31** | **90.6** | **90.6** | 90.2 | 90.5 | C-SFDA |
| **OfficeHome** | 73.6 | 73.6 | **73.8** | 73.5 | C-SFDA |
| **OfficeHome (partial set)** | 80.3 | **80.8** | 80.7 | 79.7 | AaD |
| **VisDA2017** | **89.6** | 89.3 | 89.2 | 88.4 | I-SFDA |
| **VisDA-RUST** | 79.4 | 79.2 | **79.5** | 77.3 | SF(DA)$^2$ |
| **DomainNet126** | 71.5 | 71.5 | **71.6** | 69.6 | GPUE |

Table D5: Hyperparameter values across different datasets. "Orig. $\lambda_{\mathrm{DC}}$", "Orig. $K_{\mathrm{PL}}$", and "Orig. $\tau$" refer to the original values used in our paper, which are selected following the general hyper-parameter tuning pipeline in the literature. Other hyperparameters are directly calculated with theory-motivated hyperparameter determination approaches, where "Init." and "Final" indicate the first and the last training epochs, respectively.

| Metric | Office31 | OfficeHome | OfficeHome (partial set) | VisDA2017 | VisDA-RUST | DomainNet126 |
|---|---|---|---|---|---|---|
| **Orig. $\lambda_{\mathrm{DC}}$** | 1.000 | 0.500 | 1.000 | 1.000 | 0.500 | 0.500 |
| **New $\lambda_{\mathrm{DC}}$** | 0.390 | 0.520 | 0.476 | 0.494 | 0.461 | 0.553 |
| **Orig. $K_{\mathrm{PL}}$** | 2.000 | 2.000 | 2.000 | 1.000 | 2.000 | 2.000 |
| **Init. $k_0$ (Averaged)** | 1.320 | 1.535 | 1.513 | 1.341 | 1.348 | 1.644 |
| **Final $k_0$ (Averaged)** | 1.003 | 1.028 | 1.003 | 1.008 | 1.020 | 1.079 |
| **Orig. $\tau$** | 1.300 | 1.100 | 1.100 | 1.100 | 1.100 | 1.100 |
| **Init. $\tau_t$** | 1.308 | 1.265 | 1.238 | 1.790 | 1.674 | 1.232 |
| **Final $\tau_t$** | 1.056 | 1.090 | 1.042 | 1.260 | 1.368 | 1.092 |
| $\tau_s$ **(10th percentile)** | 2.037 | 1.230 | 1.268 | 1.164 | 1.163 | 1.264 |

Table D6: Per source-target task configuration on DomainNet126. The metric notations are the same as in **Table** D5.

| Task | Acc. of Ucon-SFDA | Acc. of autoUCon-SFDA (Theory) | Acc. of autoUCon-SFDA (Stat.) | Orig. $\lambda_{\mathrm{DC}}$ | New $\lambda_{\mathrm{DC}}$ | Orig. $K_{\mathrm{PL}}$ | Init. $k_0$ (Averaged) | Final $k_0$ (Averaged) | Orig. $\tau$ | Init. $\tau_t$ | Final $\tau_t$ | $\tau_s$ |
|---|---|---|---|---|---|---|---|---|---|---|---|---|
| **C→S** | 66.5 | 64.5 | 66.0 | 0.50 | 0.52 | 2 | 1.70 | 1.08 | 1.1 | 1.20 | 1.08 | 1.23 |
| **P→C** | 69.3 | 70.3 | 70.0 | 0.50 | 0.59 | 2 | 2.33 | 1.11 | 1.1 | 1.30 | 1.11 | 1.17 |
| **P→R** | 81.0 | 81.4 | 81.4 | 0.50 | 0.45 | 2 | 1.64 | 1.04 | 1.1 | 1.28 | 1.08 | 1.36 |
| **R→C** | 75.2 | 77.0 | 77.3 | 0.50 | 0.59 | 2 | 1.45 | 1.08 | 1.1 | 1.19 | 1.09 | 1.27 |
| **R→P** | 71.1 | 71.3 | 71.0 | 0.50 | 0.58 | 2 | 1.39 | 1.09 | 1.1 | 1.17 | 1.11 | 1.32 |
| **R→S** | 64.3 | 68.1 | 67.7 | 0.50 | 0.61 | 2 | 1.52 | 1.07 | 1.1 | 1.20 | 1.09 | 1.23 |
| **S→P** | 68.1 | 67.9 | 67.6 | 0.50 | 0.55 | 2 | 1.49 | 1.08 | 1.1 | 1.30 | 1.08 | 1.27 |
| **Avg.** | 71.5 | 71.5 | 71.6 | 0.50 | 0.55 | 2 | 1.64 | 1.08 | 1.1 | 1.23 | 1.09 | 1.26 |

We first conduct a comprehensive performance comparison of the original UCon-SFDA, its automated variants, and state-of-the-art (SOTA) methods across all six benchmarks, as shown in Table D4. Notably, our findings validate that directly using theoretically derived parameters can achieve promising—and in some cases, superior—performance across all benchmarks. (For the remaining three hyperparameters $\kappa$, $\beta$ and $\lambda_{\mathrm{PL}}$, we keep them the same as those used in UCon-SFDA.)

A detailed parameter comparison is further provided in Table D5. For $k_0$ and $\tau_t$, we report their values at the first and the last training epochs to illustrate their changing trend, denoted as "Init." and "Final" in the tables, respectively. It can be observed that the theoretically determined parameters are largely aligned with the hyperparameters used in UCon-SFDA. However, they offer greater flexibility in certain scenarios. For instance, based on the averaged values of $k_0$ at the initial and final training epochs, the instance-dependant $k_0$ automatically adapts throughout the adaptation, unlike the fixed $K_{\mathrm{PL}}$, thereby better capturing uncertainty. A similar self-adaptive behavior is observed for $\tau_t$.

Additionally, we present the per-source-target task configuration on DomainNet126 (Table D6) to clearly illustrate parameter variations and their impact. For instance, as shown in the sixth coloumn of Table D6, the new $\lambda_{\mathrm{DC}}$ is task-dependent, offering greater flexibility without requiring manual selection.

In summary, the automatic versions of UCon-SFDA demonstrate promising performance while significantly reducing the number of hyperparameters in the algorithm, retaining only three hyperparameters in autoUCon-SFDA, of which only one is directly related to our proposed methods. Furthermore, additional experimental results also highlight the effectiveness of the uncertainty-guided parameter determination process.

