# OpenReview forum: "Revisiting Source-Free Domain Adaptation: a New Perspective via Uncertainty Control"
_ICLR.cc/2025/Conference — ICLR 2025 Poster_

### Official Review · Reviewer_1NWJ · 2024-11-01

**Soundness:** 4
**Presentation:** 3
**Contribution:** 3
**Rating:** 8
**Confidence:** 2

**Summary:**

The current SFDA methods that use source models for self supervised adaptation neglects the uncertainties involved in the transfer process. The authors conduct a comprehensive analysis of the two types of uncertainties arising from both positive and negative samples, providing theoretical insights for both. Based on these theoretical insights, the authors propose a dispersion control method to mitigate the uncertainty introduced by noisy negative samples. For the uncertainty arising from positive samples, the authors suggest leveraging partial labels to fully utilize the predictive uncertainty. By managing these two types of uncertainties, the authors significantly improve the model's performance.

**Strengths:**

Overall, this could be an important theoretical and algorithmic contribution. The paper comprehensively analyzes the two types of uncertainties caused by positive and negative samples during the adaptation process, proves their impact on model performance, and provides effective solutions.

**Weaknesses:**

The method of the paper requires handling both positive and negative sample uncertainties, which increases the computational complexity and training difficulty of the model.The theoretical analysis section may be too complex and difficult for general readers to understand.

**Questions:**

How are the weights λ-CL, λDC, and λPL for the different loss terms (L-CL, L-DC, L-PL) determined? What principles or guidelines were used to set these hyperparameter values?
How are the threshold and the number of retained labels for the partial label method determined?

---

> ### Author Response · Authors · 2024-11-21
> **Reply from the Authors**
>
> **Part 1/5**
>
> We greatly appreciate your constructive and detailed feedback! Below are our responses to your questions and concerns in the weaknesses part. We hope this will help you understand our paper better. We are happy to provide additional clarification if needed.
>
> ### **1. About the computational complexity.**
>
> Thank you for your thoughtful comments. We have now revised the manuscript by carefully   addressing your comments. Details are explained below.
>
> * **Compared to traditional SFDA approaches (e.g., [A]).**
>
>     Compared to prior approaches (e.g., [A]), our method introduces additional computational demands, including increased resource usage and complexity, primarily due to the explicit incorporation of data augmentations and a partial label bank during training. However, this trade-off is consistent with recent trends in the field [B, D, E], where such increases in complexity are typical for achieving significant performance improvements.
>
> * **Compared to current SOTA methods (e.g., [B, D, E]).**
>
>     The computational complexity of our approach is comparable to other modern techniques that rely on data augmentation or consistency regularization [B, D, E]. However, our method demonstrates superior performance, as evidenced by the experimental results presented in the paper.
>
>     * Specifically, unlike [D] and [E], which also involve explicit data augmentation during training, our UCon-SFDA does not require additional network structures.
>
>     * Furthermore, the partial label bank incurs only a small additional memory overhead that scales linearly with the size of the target domain data, making it manageable in real-world SFDA application scenarios.
>
>     * We also compared the training time and GPU memory usage of UCon-SFDA with AaD [A] and a recent SOTA method, SF(DA)$^2$, which leverages implicit data augmentation in SFDA problems [B]. As shown in **Table R6**, the evaluation results on VisDA2017 further demonstrate that, with tolerable computational and storage overhead, our method achieves superior performance.
>
>         **Table R6:** Comparison of Training Time, Memory Usage, and Accuracy on VisDA2017.
>
>         | Method                | Training Time (Normalized w.r.t. AaD) | Memory Usage (Normalized w.r.t. AaD) | Accuracy (%) |
>         |-----------------------|---------------------------------------|--------------------------------------|--------------|
>         | **AaD**              | 1.000                                 | 1.000                                | 87.3         |
>         | **SF(DA)$^2$**        | 1.036                                 | 1.052                                | 88.1         |
>         | **UCon-SFDA (Ours)**  | 1.058                                 | 1.112                                | 89.6         |
>
> Moreover, the computational demands of our method can be further mitigated through parallel computation strategies, making it more efficient in practice. We have incorporated these discussions and additional experimental results in the revised manuscript (**Appendix C.7**).

---

> ### Author Response · Authors · 2024-11-21
> **Reply from the Authors**
>
> **Part 2/5**
> ### **2. Elaboration on theoretical results.**
>
> Thank you for the comment. In  this revision we have carefully addressed this aspect for greater clarity.
>
> * To improve the presentation of the initial version and make it more succinct while maintaining accessibility and thoroughness, we have carefully refined the content in this revision to present our development in a clear and concise manner.  Specifically, in the revised manuscript, we have
>
>     * streamlined the presentation to emphasize key insights while preserving  the necessary rigor (**Remarks 4.1, 4.2, 4.3**).
>
>     * added remarks to link the theoretical results with their practical implications for the algorithm, providing intuitive explanations in the context of SFDA (**Remarks 4.2, 4.4**).
>
> * To provide a clearer link between theory and its practical implementation, we emphasize the two main theoretical results and their roles below:
>
>     * **(i) Theorem 4.1: Dispersion Control via Data Augmentation Alignment.**
>
>         Theorem 4.1 demonstrates that controlling robust risk in the presence of potential false negatives requires addressing both the negative sample loss ($\mathcal{L}\^{-}\_{CL}$ in Eq. (7)) and a dispersion term ($\mathcal{V}_{\mathscr{d}}$). Below, we elaborate on how the dispersion control term $\mathcal{V}\_{\mathscr{d}}$ is utilized to derive the loss term $\mathcal{L}\^{-}\_{DC}$ in Eq. (7), and further highlight its theoretical significance as well as its potential to guide future research directions.
>
>         * **From theory to algorithm.**
>
>             In applications, domain shift makes it challenging to distinguish false negatives from true negatives. To address this, we propose to achieve the dispersion control effect by minimizing the negative similarity between an instance $\mathbf{x}$ and its manually constructed pseudo-false negatives, as illustrated in **Figures 2 (b)-(c)**. Specifically, as shown in **Figure 1(b)**, we observe that the source model's prediction on the augmented version of $\mathbf{x}$, denoted as $\mathtt{AUG}(\mathbf{x})$, may not align with the prediction for $\mathbf{x}$. When this occurs, $\mathtt{AUG}(\mathbf{x})$ is automatically treated as a false negative for $\mathbf{x}$. This dispersion control effect is then implemented through the loss term  $\mathcal{L}^{-}_{DC}$ in Eq. (7), which minimizes the negative similarity of $\mathbf{x}$ and $\mathtt{AUG}(\mathbf{x})$.
>
>         * **Theoretical significance and promising future research directions.**
>
>             These insights establish a theoretical foundation for the use of consistency loss in SFDA, thus bridging the gap between theory and applications, explaining the observed performance improvements. In addition, beyond the specific implementation in Eq. (7), the general result of Theorem 4.1 allows for flexible extensions, such as constructing false negatives using mixup or large-scale pre-trained models, showcasing the adaptability of our approach.
>
>
>     * **(ii) Theorem 4.2: Supervision Relaxation via Partial Label Training.**
>
>         Theorem 4.2 introduces a method to leverage both certain and uncertain label information to capture prediction uncertainty effectively, in contrast to prior work that neglects or naively uses such uncertainty.
>
>         * **Leverage certain label information: positive supervision loss $\mathcal{L}^{+}_{CL}$.**
>
>             If an instance $\mathbf{x}$ receives a certain supervision signal,  the optimal prediction for $\mathbf{x}$ corresponds to the label with the highest predicted probability. This certain label information is incorporated through the **positive supervision loss** term, $\mathcal{L}^{+}_{CL}$, as defined in Eqs. (8)-(9).
>
>         * **Leverage uncertain label information: partial label loss $\mathcal{L}^{+}_{PL}$.**
>
>             If the prediction information is uncertain,  the optimal prediction for the  instance can be represented as a set of (instance-dependent) partial labels.  **Rather than directly relying on the estimated pseudo labels, we construct a partial label set, $\mathscr{Y}_{PL,i}$**, for each instance. This approach offers a more robust supervisory signal by accounting for multiple potential labels and reducing reliance on noisy single-label predictions. This uncertain label information is captured through the **partial label loss** term, $\mathcal{L}^{+}_{PL}$, as outlined in Eqs. (8)-(9).
>
> * As the integration of SFDA, contrastive learning, and uncertainty modeling  introduces complex notation, which may pose challenges for readers. To help alleviate this, we have included a **notation table** in the revised manuscript (**Appendix A.1**) to serve as a quick reference for terms and symbols used throughout the paper. In addition, we have revised **Figure 2** to provide a clearer visualization of the dispersion control effect.
>
> We have revised the manuscript to make the theoretical section more digestible while preserving the depth and clarity.

---

> ### Author Response · Authors · 2024-11-21
> **Reply from the Authors**
>
> **Part 3/5**
>
> ### **3. Hyper-parameter tuning.**
>
> In this revision we have carefully addressed your comments on hyperparameter selection, as explained below.
>
> * **Coefficient for different loss terms ($\lambda\^{-}\_{CL}$, $\lambda\_{DC}$ and  $\lambda_{PL}$).**
>
>     For $\lambda\^{-}\_{CL}$, **we followed the previous works** [A, B] and set its value to 1 for all datasets. For $\lambda_{DC}$ and  $\lambda_{PL}$, **we tuned them with narrow ranges** ({$0.5, 1$} for $\lambda_{DC}$ and {$0.001, 0.01, 0.05, 0.1$} for $\lambda_{PL}$) and selected the best-performing values. Additional sensitivity experiments (**Appendix C.5**) further confirmed the robustness of these hyperparameters across different values.
>
>
> * **Threshold ($\tau$) and the Number of Retained Labels ($K_{PL}$) in Partial Label Loss.**
>
>     For the hyperparameters ($K_{PL}$ and $\tau$) in partial label loss term, **inspired by our theoretical findings, we determined that a small $K_{PL}$ (e.g., 1 or 2) and a small $\tau$ (slightly larger than 1) are most effective.** These settings help focus on the most uncertain data with minimal additional label noise, which is consistent with our theoretical results (Theorem 4.2). Dataset properties, such as dataset scale, task difficulty, and the number of class categories, also influenced their selection. For example, on VisDA2017, which has more target data but fewer class categories, a smaller value of $K_{PL}$, such as 1, performs well. As for the selection of $\tau$, on simpler datasets like office-31 with higher source model prediction accuracy, a larger $\tau$ is beneficial for relaxing the identification of the most uncertain data on the target domain.
>
>
> * **Hyperparameter Selection Guidelines.**
>
>     Rather than exhaustively searching the entire combinatorial space, **we adopted a sequential and stage-wise tuning strategy**. Specifically, we began by adopting $\kappa$, $\lambda\^{-}\_{CL}$ and $\beta$ from previous works [A,B]. We then incrementally introduced the dispersion control term and determined the best $\lambda_{DC}$. Finally, we added the partial label term, selected the most suitable $K_{PL}$ and $\tau$ based on the dataset properties, and tuned $\lambda_{PL}$. For datasets with multiple source-target pairs (e.g., DomainNet126, Office-31 and Office-Home), we tuned hyperparameters on one or two sub-tasks and applied the same hyperparameters across the entire dataset.
>
>     In our experiments, we followed the common pipeline for hyperparameter tuning in the literature (e.g., [A,B]) and employed the hyperparameter sensitivity analyses combined with some unsupervised learning metrics (such as SND score [C]) to determine the best-performing values for each hyperparameter.
>
> We have revised the hyperparameter selection process in **Appendix B** and included the above discussions in the revised manuscript. Additionally, to comprehensively study the impact of different hyperparameters in UCon-SFDA, we conducted additional sensitivity analyses in **Appendix C.5**. The experimental results are summarized in **Tables R1-R2** (for $K_{PL}$ and $\tau$), **Tables R3-R4** (for $\lambda_{PL}$ and $\lambda_{DC}$), and **Table R5** (for $\beta$). These results are further discussed and visualized in **Figures 17-19** in the revised manuscript (**Appendix C.5**). The findings suggest that:
>
> * (i) our method is robust across different hyperparameter values and achieves stable performance improvements within a suitable range;
>
> * (ii) the proposed dispersion control term stabilizes the performance of negative samples and makes the model more robust to varying decay exponents;
>
> * (iii) with finer-grained hyperparameter searches, UCon-SFDA can achieve even better performance.

---

> ### Author Response · Authors · 2024-11-21
> **Reply from the Authors**
>
> **Part 4/5**
>
> ### **4. Summarization.**
>
> For ease of your review, here we include  key changes in response to your comments; other revisions can be found in Remarks 4.1-4.4, Figure 2, Appendix A.1, Appendix B, Appendix C.5, and Appendix C.7:
>
> * We have refined the theoretical analysis, added remarks connecting theory to applications (**Remarks 4.1-4.4**), revised **Figure 2** to provide a clearer visualization of the dispersion control effect, and included a notation table for clarity in the revised manuscript (**Appendix A.1**).
>
> * We have clarified the hyperparameters directly related to our method and elaborated on our straightforward hyperparameter selection strategy, resulting in effective selections (**Appendix B**). We have also provided new experimental results on hyperparameter sensitivity analysis (**Appendix C.5**).
>
> *  Regarding the inherently increased computational complexity, we have provided additional discussions on its comparability to existing state-of-the-art methods and highlighted the trade-off it offers for achieving superior performance (**Appendix C.7**).
>
>
> **References:**
>
> [A] Yang, S., Jui, S., \& van de Weijer, J. "Attracting and dispersing: A simple approach for source-free domain adaptation." NeurIPS, 2022.
>
> [B] Hwang, U., Lee, J., Shin, J., \& Yoon, S. "SF(DA)$^2$: Source-free Domain Adaptation Through the Lens of Data Augmentation." ICLR, 2024.
>
> [C] Saito, K., Kim, D., Teterwak, P., Sclaroff, S., Darrell, T., \& Saenko, K. "Tune it the right way: Unsupervised validation of domain adaptation via soft neighborhood density." ICCV, 2021.
>
> [D] Karim, N., Mithun, N. C., Rajvanshi, A., Chiu, H. P., Samarasekera, S., \& Rahnavard, N. "C-sfda: A curriculum learning aided self-training framework for efficient source free domain adaptation." CVPR, 2023.
>
> [E] Mitsuzumi, Y., Kimura, A., \& Kashima, H. "Understanding and Improving Source-free Domain Adaptation from a Theoretical Perspective." CVPR, 2024.

---

> ### Author Response · Authors · 2024-11-21
> **Reply from the Authors**
>
> **Part 5/5**
>
> ### **Tables**
>
> **Table R1.** Performance comparisons under different values of $K_{PL}$.
>
>
> |     Dataset       | **$K_{PL}$** = 1     | **$K_{PL}$** = 2     | **$K_{PL}$** = 3     |
> |:---------------------|:-------:|:-------:|:-------:|
> | **VisDA-RUST**     | 79.04 | **79.20** | 79.09 |
> | **Office-Home (Pr → Cl)** | 61.48 | **61.56** | 61.42 |
>
> **Table R2.** Performance comparisons under different values of $\tau$.
>
> |      **Dataset**       | **$\tau$ = 1.1**   | **$\tau$ = 1.3**   | **$\tau$ = 1.5**   |
> |:---------------------|:-------:|:-------:|:-------:|
> | **Office-31**      | 90.34 | **90.55** | 90.38 |
> | **VisDA-RUST**     | **79.20** | 79.10 | 79.03 |
>
>
>
>
> **Table R3.** Performance comparison across different values of $L_{PL}$ on three datasets. **Bold** text indicates the results obtained with the best-performing hyperparameter set reported in the main paper, while $\underline{\text{underlined}}$ text highlights improved results found using finer hyperparameter tuning during the rebuttal phase.
> | Dataset | $\lambda_{PL}$=0.0001 | $\lambda_{PL}$=0.001 | $\lambda_{PL}$=0.01 | $\lambda_{PL}$=0.05 | $\lambda_{PL}$=0.1 | $\lambda_{PL}$=0.15 | $\lambda_{PL}$=0.2 |
> |:---|:---:|:---:|:---:|:---:|:---:|:---:|:---:|
> | VisDA-RUST | 78.92  | 78.96  | 78.94  | **79.20** | $\underline{\text{79.40}}$ | $\underline{\text{79.53}}$ | 79.19  |
> | DomainNet126 (R $\to$ P) | 70.52  | 70.53  | 70.86  | 70.98  | **71.10** | $\underline{\text{71.17}}$ | 70.89  |
> | Office-Home (Pr $\to$ Cl) | 61.42  | **62.20** | 61.70  | 61.56  | 61.39  | 61.21  | 60.18  |
>
> **Table R4.** Performance comparison across different values of $L_{DC}$ on three datasets. **Bold** text indicates the results obtained with the best-performing hyperparameter set reported in the main paper, while $\underline{\text{underlined}}$ text highlights improved results found using finer hyperparameter tuning during the rebuttal phase.
>
> | Dataset | $\lambda_{DC}$=0.05 | $\lambda_{DC}$=0.1 | $\lambda_{DC}$=0.5 | $\lambda_{DC}$=1 | $\lambda_{DC}$=1.5 | $\lambda_{DC}$=2 |
> |:---|---|---|---|---|---|---|
> | VisDA-RUST | $\underline{\text{79.51}}$ | $\underline{\text{79.82}}$ | **79.40** | 79.25  | 79.04  | 78.82  |
> | DomainNet126 (R $\to$ P) | 70.10  | 70.58  | **71.10** | $\underline{\text{71.54}}$ | $\underline{\text{71.38}}$ | 70.94  |
> | Office-Home (Pr $\to$ Cl) | 60.25  | 61.60  | **62.20** | 61.92  | 60.55  | 60.50  |
>
>
>
> **Table R5:** Performance comparison under different $\beta$ values on DomainNet126 (R → P).
> | $\beta$           | 0.25  | 0.5   | 0.75  | 1     | 5     |
> |---------------------|-------|-------|-------|-------|-------|
> | **UCon-SFDA (Ours)** | 70.96 | 71.10 | 71.10 | 70.99 | 71.11 |
> | **Basic CL method (AaD)** | 67.49 | 67.63 | 67.80 | 67.80 | 67.23 |
>
>
> **Table R6:** Comparison of Training Time, Memory Usage, and Accuracy on VisDA2017.
>
> | Method                | Training Time (Normalized w.r.t. AaD) | Memory Usage (Normalized w.r.t. AaD) | Accuracy (%) |
> |-----------------------|---------------------------------------|--------------------------------------|--------------|
> | **AaD**              | 1.000                                 | 1.000                                | 87.3         |
> | **SF(DA)$^2$**        | 1.036                                 | 1.052                                | 88.1         |
> | **UCon-SFDA (Ours)**  | 1.058                                 | 1.112                                | 89.6         |

---

> ### Author Response · Authors · 2024-11-27
> **Looking forward to hearing from you**
>
> Dear Reviewer 1NWJ,
>
> We hope this message finds you well.
>
> We deeply appreciate your time and effort in reviewing our submission and providing valuable feedback. Your insights are crucial to our work.
>
> In our previous response, we carefully addressed the concerns you raised, including providing a comprehensive clarification of the hyperparameter setups, offering detailed elaboration on our theoretical findings, and thoroughly analyzing the computational complexity of the proposed method.
>
> If there are any points that remain unclear, we would be glad to provide further clarification or engage in further discussion. We look forward to hearing from you.
>
> Thanks,
>
> The Authors of submission 8466

---

> ### Author Response · Authors · 2024-12-01
> **Follow-Up Response on Hyperparameter Determination (1/3)**
>
> Thank you once again for your valuable comments on the hyperparameter selection process. Inspired by the discussion with another reviewer, we have extended our previous algorithm into **two automatic versions with fewer hyperparameters requiring manual tuning**. These extensions aim to thoroughly optimize the incorporation of parameters in our method, enhancing its efficiency and adaptability.
>
> ---
> ### **Theory-Motivated Hyper-Parameter Determination and New Notations.**
>
> In designing the UCon-SFDA algorithm, we prioritized engineering flexibility and ease of implementation, which led us to introduce four hyperparameters. **However, three of these parameters have explicit expressions derived from our theoretical results or can be determined based on dataset and source model properties, thereby eliminating the need for manual hyperparameter tuning.** Specifically,
>
> * $\lambda_{DC}$: Inconsistency Ratio (Motivated by Theorem 4.1 and Remark 4.2).
>
>     As suggested by Theorem 4.1 and Remark 4.2, the dispersion control effect can be achieved by minimizing the negative similarity between the anchor point and its augmented prediction. If the inconsistency rate between anchor points and their associated augmented predictions is high, it indicates greater uncertainty in negative sampling,  thus requiring stronger dispersion control. **Based on this observation, we propose directly using the model prediction inconsistency ratio (denoted as "$\color{blue}\text{New}\ \lambda_{DC}$") as the coefficient for the dispersion control term.**
>
> * $K_{PL}$:$\color{blue}k_0$ (Theorem 4.2)
>
>   By Theorem 4.2, when the uncertainty set in Eq. (5) of our paper is defined using the 1-Wasserstein distance, the length of the partial label set, denoted by $K_{PL}$, can be explicitly determined as $K_{PL}=$$\color{blue}k_0$, where $\color{blue}k_0$ is defined as follows:
>
>   * (i) If $\frac{1}{K}\ge \frac{1}{k}\sum_{j=1}^{k}\mathcal{p}^{+}_{(j)}-\frac{1}{k}\delta$ for all $k\in[K-1]$, then we take $\color{blue}k_0$$=K$.
>   * (ii) Otherwise, we take the $\color{blue}k_0$$\in[K-1]$ that satisfies $\frac{1}{\color{blue}{k\_0}}\sum\_{j=1}\^{\color{blue}{k\_0}}\mathcal{p}\^{+}\_{(j)}-\frac{1}{\color{blue}{k\_0}}\delta\ge\frac{1}{k}\sum\_{j=1}\^{k}\mathcal{p}\^{+}\_{(j)}-\frac{1}{k}\delta$ for all $k\in[K-1]$.
>
>   In the formulas above, $K$ represents the number of classes, $\mathcal{p}^{+}_{(j)}$ denotes the $j$-th largest predicted probability for the considered anchor point, and $\delta$ could be taken as $\frac{1}{K}$ as suggested by the proof of Theorem 4.2. **Hence, the length of the partial label set, which can be directly calculated, is determined by the model's predictions for the anchor point as well as the specific classification task at hand.**
>
>
> * $\tau$: We propose two approaches to distinguish between certain and uncertain label information.
>
>   * (i) **Statistical Insights Approach - $\color{blue}{\tau_{s}}$**
>
>     This approach leverages the properties of the source model and the target data, combined with statistical insights. Specifically, we first use the source model to compute the predicted probabilities for each target data point. Next, we calculate the ratio of the two highest predicted probabilities for all data points and **select the 10th percentile of these ratios as the value of $\tau$, denoted as $\color{blue}{\tau_{s}}$** in the updated tables. This value selection allows us to summarize the data distribution and identify the 10% most uncertain data. The 10th percentile is chosen because it is a widely used measure in statistical research to analyze data distributions and highlight low-end values.
>
>   * (ii) **Theoretical Criterion Approach - $\color{blue}{\tau_{t}}$**
>
>     Alternatively, we can bypass the ratio of the two highest predicted probabilities and directly apply the criterion outlined in Remark 4.3 to distinguish between certain and uncertain label information. As discussed in Remark 4.3, in the special case where $\mathcal{p}^{+}_{(1)}\ge\max\{\frac{1}{K}+\delta,\mathcal{p}^{+}_{(2)}+\delta\}$, we refer to it as **certain label information**. Conversely, if this condition is NOT satisfied, the label information is deemed **uncertain**, and the corresponding data is added to the uncertain data bank. **Based on the selected uncertain data, we calculate a corresponding ratio $\color{blue}{\tau_{t}}$**, as reported in the updated tables, for post-comparison purposes.
>
> ---
> Building upon different uncertain data selection strategies, we proposed two automatic UCon-SFDA methods: $\color{blue}\text{autoUCon-SFDA (Theory)}$ and $\color{blue}\text{autoUCon-SFDA (Stat.)}$. For $\color{blue}k_0$ and $\color{blue}\tau_{t}$, we present their values in the first and the last training epochs to illustrate their changing trend, indicated by "$\color{blue}\text{Init.}$" and "$\color{blue}\text{Final}$" in the tables, respectively.

---

> > ### Author Response · Authors · 2024-12-01
> > **Follow-Up Response on Hyperparameter Determination (2/3)**
> >
> > ---
> > ### **Experimental Results.**
> >
> > We present the experimental results in **Tables R2-1, R2-2, and R2-3**. Specifically:
> >
> > * Performance comparisons between the original UCon-SFDA, the newly extended methods, and SOTA methods across all six benchmarks are shown in **Table R2-1**. **Notably, our findings validate that directly using theoretically derived parameters can achieve promising—and in some cases, superior—performance across all benchmarks.** (For the remaining three hyperparameters $\kappa$, $\beta$ and $\lambda_{PL}$, we kept them the same as those used in UCon-SFDA.)
> >
> > * A comprehensive parameter comparison is provided in **Table R2-2**. **It can be observed that the theoretically determined parameters are largely aligned with the hyperparameters used in UCon-SFDA. However, they offer greater flexibility in certain scenarios.**
> >
> >     * For instance, we present the **averaged values of $\color{blue}k_0$ at the initial and final training epochs**. Unlike the fixed $K_{PL}$, the instance-dependant $\color{blue}k_0$ automatically adapts throughout the adaptation process to better capture uncertainty.
> >
> >     * A similar self-adaptive behavior is observed for $\color{blue}\tau_{t}$.
> >
> > * Additionally, we present the per source-target task configuration on DomainNet126 to clearly illustrate parameter changes and their impact. For instance, as shown in the 6th coloumn of **Table R2-3**, the $\color{blue}\text{New}\ \lambda_{DC}$ is task-dependent, offering greater flexibility without requiring a manual selection process.
> >
> > ---
> > ### **Summary and Future Work Insights.**
> >
> > In summary, **(1)** the automatic versions of UCon-SFDA have demonstrated **promising performance while significantly reducing the number of hyperparameters** in the algorithm (retaining only three hyperparameters in autoUCon-SFDA, with just one directly related to our proposed methods). **(2)** The additional experimental results also illustrate the effectiveness of the uncertainty-guided parameter determination process. We believe that our theoretical framework **offers valuable insights into addressing the challenge of hyperparameter selection and tuning in UDA**.
> >
> > We sincerely thank you once again for your valuable time and effort in reviewing our paper. We hope that this follow-up response regarding hyperparameter determination, along with the additional experimental results, adequately addresses your concerns about the parameters used in our experiments.

---

> > > ### Author Response · Authors · 2024-12-01
> > > **Follow-Up Response on Hyperparameter Determination (3/3)**
> > >
> > > ### **Tables**
> > >
> > > **Table R2-1.** Performance comparisons across different hyper-parameter selection (calculation) methods. **Bold** text indicates the best results.
> > >
> > > |  Dataset  |  UCon-SFDA  |  autoUCon-SFDA (Theory)  |  autoUCon-SFDA (Stat.)  |  SOTA Method Performance  |  SOTA Method |
> > > |:---------------------|:-------:|:-------:|:-------:| :-------:|:-------:|
> > > |  **Office31**  |  **90.6**  |  **90.6**  |  90.2  |  90.5  |  C-SFDA  |
> > > |  **OfficeHome**  |  73.6  |  73.6   |  **73.8**  |  73.5  |  C-SFDA  |
> > > |  **OfficeHome (partial set)**  |  80.3  |  **80.8**  |  80.7   |  79.7  |  AaD  |
> > > |  **VisDA2017**  |  **89.6**  |  89.3   |  89.2   |  88.4  |  I-SFDA  |
> > > |  **VisDA-RUST**  |  79.4  |  79.2  |  **79.5**  |  77.3  |  SF(DA)$^2$  |
> > > |  **DomainNet126**  |  71.5  |  71.5  |  **71.6**  |  69.6  |  GPUE  |
> > >
> > >
> > > **Table R2-2.** Hyper-parameter values across different datasets. "Orig. $\lambda_{DC}$", "Orig. $K_{PL}$", and "Orig. $\tau$" refer to the original values used in our paper, which are selected following the general hyper-parameter tuning pipeline in the literature. $\color{blue}\text{The hyper-parameters highlighted in blue}$ are directly calculated with theory-motivated hyper-parameter determination approaches, where "$\color{blue}\text{Init.}$" and "$\color{blue}\text{Final}$" indicate the first and the last training epochs, respectively. $\color{green}\text{The text in green}$ specifies the associated selection/calculation methods.
> > >
> > > |  Metric  |  Office31  |  OfficeHome  |  OfficeHome (partial set)  |  VisDA2017  |  VisDA-RUST  |  DomainNet126  |
> > > |:---------------------|:-------:|:-------:|:-------:| :-------:|:-------:|:-------:|
> > > |  Orig. $\lambda_{DC}\ \color{green}\text{(Original value used in our paper)}$  |  1.000  |  0.500  |  1.000  |  1.000  |  0.500  |  0.500  |
> > > |  $\color{blue}\text{New}\ \lambda_{DC}\ \color{green}\text{(Inconsistency Ratio: Guided by Theorem 4.1)}$  |  0.390  |  0.520  |  0.476  |  0.494  |  0.461  |  0.553  |
> > > |  Orig. $K_{PL}\ \color{green}\text{(Original value used in our paper)}$  |  2.000  |  2.000  |  2.000  |  1.000  |  2.000  |  2.000  |
> > > |  $\color{blue}\text{Init.}\ k_0\ \text{(Averaged)}\ \color{green}(k_0\ \text{in Theorem 4.2})$  |  1.320  |  1.535  |  1.513  |  1.341  |  1.348  |  1.644  |
> > > |  $\color{blue}\text{Final.}\ k_0\ \text{(Averaged)}\ \color{green}(k_0\ \text{in Theorem 4.2})$  |  1.003	 |  1.028	|  1.003  |	1.008  |	1.020  |	1.079
> > > |  Orig. $\tau\ \color{green}\text{(Original value used in our paper)}$  |  1.300	  |  1.100  |	1.100  |	1.100  |	1.100  |	1.100  |
> > > |  $\color{blue}\text{Init.}\ \tau_{t}\ \color{green}\text{(Calculated Using Theoretical Criterion: Remark 4.3)}$  |  1.308  |	1.265  |	1.238  |	1.790  |	1.674  |	1.232  |
> > > |  $\color{blue}\text{Final}\ \tau_{t}\ \color{green}\text{(Calculated Using Theoretical Criterion: Remark 4.3)}$  |  1.056  |	1.090  |	1.042  |	1.260  |	1.368  |	1.092  |
> > > |  $\color{blue}\tau_{s}\ \color{green}\text{(Derived from Statistical Insights: 10th percentile)}$  |  2.037  | 	1.230  |	1.268  |	1.164  |	1.163  |	1.264  |
> > >
> > > **Table R2-3.** Per source-target task configuration on DomainNet126. The metric notations are the same as in **Tables R2-2**.
> > >
> > > | Task | Acc. of Ucon-SFDA | $\color{blue}\text{Acc. of}$  $\color{blue}\text{autoUCon-SFDA}$ $\color{blue}\text{(Theory)}$ |$\color{blue}\text{Acc. of}$  $\color{blue}\text{autoUCon-SFDA}$ $\color{blue}\text{(Stat.)}$ | Orig. $\lambda_{DC}$  | $\color{blue}\text{New}\ \lambda_{DC}$ | Orig. $K_{PL}$ | $\color{blue}\text{Init.}\ k_0$ $\color{blue}\text{(Averaged)}$  | $\color{blue}\text{Final}\ k_0$ $\color{blue}\text{(Averaged)}$   | Orig. $\tau$ | $\color{blue}\text{Init.}\ \tau_{t}$        | $\color{blue}\text{Final}\ \tau_{t}$       | $\color{blue}\tau_{s}$  |
> > > |:---:|:---:|:---:|:---:|:---:|:---:|:---:|:---:|:---:|:---:|:---:|:---:|:---:|
> > > | **C$\to$S** | 66.5  | 64.5  | 66.0  | 0.50  | 0.52  | 2 | 1.70  | 1.08  | 1.1 | 1.20  | 1.08  | 1.23  |
> > > | **P$\to$C** | 69.3  | 70.3  | 70.0  | 0.50  | 0.59  | 2 | 2.33  | 1.11  | 1.1 | 1.30  | 1.11  | 1.17  |
> > > | **P$\to$R** | 81.0  | 81.4  | 81.4  | 0.50  | 0.45  | 2 | 1.64  | 1.04  | 1.1 | 1.28  | 1.08  | 1.36  |
> > > | **R$\to$C** | 75.2  | 77.0  | 77.3  | 0.50  | 0.59  | 2 | 1.45  | 1.08  | 1.1 | 1.19  | 1.09  | 1.27  |
> > > | **R$\to$P** | 71.1  | 71.3  | 71.0  | 0.50  | 0.58  | 2 | 1.39  | 1.09  | 1.1 | 1.17  | 1.11  | 1.32  |
> > > | **R$\to$S** | 64.3  | 68.1  | 67.7  | 0.50  | 0.61  | 2 | 1.52  | 1.07  | 1.1 | 1.20  | 1.09  | 1.23  |
> > > | **S$\to$P** | 68.1  | 67.9  | 67.6  | 0.50  | 0.55  | 2 | 1.49  | 1.08  | 1.1 | 1.30  | 1.08  | 1.27  |
> > > | **Avg.** | 71.5  | 71.5  | 71.6  | 0.50  | 0.55  | 2 | 1.64  | 1.08  | 1.1 | 1.23  | 1.09  | 1.26  |

---

### Official Review · Reviewer_GdvL · 2024-11-03

**Soundness:** 3
**Presentation:** 3
**Contribution:** 3
**Rating:** 6
**Confidence:** 3

**Summary:**

This paper studies source-fee domain adaptation, with a specific focus on the uncertainty in the contrastive learning-based source-free domain adaptation (SFDA) solution. The authors comprehensively analyze two types of uncertainty including both negative and positive uncertainty through the lens of Distributionally Robust Optimization. Based on the theoretical framework the authors propose an uncertainty-control SFDA method (UCon-SFDA). Extensive experiments demonstrate the advantages of UCon-SFDA over existing SFDA approaches.

**Strengths:**

(i) Significance. This source-free domain adaptation problem studied in the submission is significant for real-world applications of deep learning models. The practical assumption of no source access is meaningful for privacy-sensitive scenarios such as medical data.

(ii) Novelty. This paper provides an in-depth theoretical analysis of the overlooked uncertainty problem in previous SFDA methods within a unified DRO framework.

(iii) Quality. Extensive experiments and comparisons demonstrate that the proposed approach UCon-SFDA outperforms existing SFDA methods.

(iv) Clarity. The presentation is clear with detailed theoretical analysis and illustrative figures.

**Weaknesses:**

Although the approach is supported by in-depth theoretical analysis and impressive experimental results, I remain concerned about the hyperparameters setting.

Since source-free domain adaptation only has unlabeled target-domain data, it is quite difficult to accurately tune hyperparameters for the SFDA approach. Usually, one solution is to avoid involving many hyperparameters in the SFDA approach. However, the proposed UCon-SFDA necessitates the tuning of many hyperparameters including different $\lambda$ as shown in Table 6 in the Appendix. Moreover, Table 6 demonstrates that the authors adopt different hyperparameter values for different datasets. The challenge is that in practical applications, we would be given an unseen unlabeled target domain where existing hyperparameters can be worse.

My specific questions are:

- How to tune various hyperparameters to ensure UCon-SFDA can work well in both this paper and other new SFDA tasks?

- Hyperparameter sensitivity analysis is critical for understanding the method but missing in the paper.

**Questions:**

Please see the question in Weaknesses

---

> ### Author Response · Authors · 2024-11-21
> **Reply from the Authors**
>
> **Part 1/3**
>
> We greatly appreciate your constructive and detailed feedback! Below are our responses to your questions and concerns in the weaknesses part. We hope this will help you understand our paper better. We are happy to provide additional clarification if needed.
>
> ### **1. Hyper-parameter Tuning.**
>
> To address your concerns about the hyperparameter tuning procedure in SFDA, we have carefully revised the manuscript for greater clarity. Below, we highlight our responses.
>
>    * (a) We explain why only a limited number of hyperparameters in **Table 6** require lightweight tuning, guided by our theoretical insights and the design of our algorithm.
>
>    * (b) We outline the entire hyperparameter tuning procedure, which is designed to be straightforward and facilitate effective application in new SFDA tasks.
>
>    * (c) While we followed the common setting for hyperparameter tuning in our paper, in this rebuttal, we also propose a more practical criterion that offers a novel perspective on addressing the challenges of hyperparameter selection in real-world applications for the SFDA problem.
>
>
> * **Lightweight Hyperparameter Tuning with Small Search Space.**
>
>     While we list seven hyperparameters in **Table 6** in the **Appendix B**, for completeness and ease of reference, it is important to note that **only four hyperparameters are directly tied to our proposed method**, with their selection guided by our theoretical insights, as explained below.
>
>     * For the first three hyperparameters ($\kappa$, $\lambda^{-}_{CL}$ and $\beta$), we primarily **followed the setups from previous works** [A, C].
>
>     * The selections of hyperparameters ($K_{PL}$ and $\tau$) in partial label loss term were **guided by our theoretical results**. Specifically, the theoretical results in Theorem 4.2 suggest that the partial label loss should be applied to the most uncertain data while minimizing additional label noise. Therefore, we choose a relatively small value for $K_{PL}$ (e.g., 1 or 2) and $\tau$ (slightly larger than 1), which largely reduces the search space. This choice is also confirmed by our empirical observations, as shown in **Tables R1-R2** and **Figure 17** in the revised manuscript (**Appendix C.5**).
>
>     * For $\lambda_{PL}$ and $\lambda_{DC}$, we **tuned them within reduced range** ({$0.5, 1$} for $\lambda_{DC}$ and {$0.001, 0.01, 0.05, 0.1$} for $\lambda_{PL}$) and selected the best-performing values, following the approach used in previous studies [A, C]. Sensitivity experiments confirmed that these hyperparameters are robust across different values.
>
> * **Stage-Wise Tuning Procedure.**
>
>     Rather than exhaustively searching the entire combinatorial space, **we adopted a sequential and incremental tuning strategy**. Specifically, we began by adopting $\kappa$, $\lambda\^{-}\_{CL}$ and $\beta$ from previous works [A,C]. We then incrementally introduced the dispersion control term and determined the best $\lambda_{DC}$. Finally, we added the partial label term, selected the most suitable $K_{PL}$ and $\tau$ based on the dataset properties, and tuned $\lambda_{PL}$. For datasets with multiple source-target pairs (e.g., DomainNet126, Office-31 and Office-Home), we tuned hyperparameters on one or two sub-tasks and applied the same hyperparameters across the entire dataset.
>
>
> * **Different Hyperparameter Selection Criteria.**
>
>     Hyperparameter tuning in SFDA can be particularly challenging due to the absence of target labels and the presence of significant distribution shifts across domains. In our experiments, we followed the general pipeline for hyperparameter tuning in the literature (e.g., [A, C]), and employed the SND (Soft Neighborhood Density) score [B] and sensitivity analysis to guide the hyperparameter selection. However, we found that SND scores do not always correlate well with performance on the full target data, and sensitivity analysis often incurs significant computational overhead. To address this, in this rebuttal, we also investigate a new small evaluation set-based method. Specifically, we randomly select a subset (300 data points) from the full unlabeled target data (containing 5k-50k data points), manually label it, and create a pseudo-validation set. Hyperparameters are then selected based on their performance on the small evaluation set. While this approach does involve manual annotation, the required amount of labeled data is minimal, making it practical in real-world scenarios and providing more accurate hyperparameter selection. As shown in **Tables R3-R4** and visualized in **Figure 18** and **Figure 19** (revised **Appendix C.5**), the small evaluation set method outperforms SND scores in identifying the best-performing hyperparameters.

---

> ### Author Response · Authors · 2024-11-21
> **Reply from the Authors**
>
> **Part 2/3**
>
> ### **2. Hyper-parameter sensitivity analysis.**
>
> Regarding your concerns of hyperparameter tuning in SFDA, we have conducted additional experiments to comprehensively analyze the sensitivity of different hyperparameters on several datasets. The results are summarized in **Table R1** (for $K_{PL}$), **Table R2** (for $\tau$), **Table R3** (for $\lambda_{PL}$), **Table R4** (for $\lambda_{DC}$), and **Table R5** (for $\beta$). These results are further discussed and visualized in **Figures 17-19** in the revised manuscript (**Appendix C.5**). The findings show that:
>
> * (a) Our method is robust across a wide range of hyperparameter values and achieves stable performance,  even when exact tuning is not feasible.
> * (b) The proposed dispersion control term plays a key role in stabilizing performance, particularly for negative samples, across varying decay exponents.
> * (c) Better performance has been achieved with finer-grained searches.
>
> ### **3. Summarization**
>
> Thank you for your thoughtful comments, which led us to improve the presentation of our work.
> For ease of your review, here we include  key changes in response to your comments; other revisions can be found in **Appendix B** and **Appendix C.5**:
>
> * We have clarified the hyperparameters directly related to our method and elaborated on our stage-wise tuning strategy, resulting in effective selections (**Appendix B**).
> * We have provided further experimental results on hyperparameter sensitivity analysis (**Appendix C.5**).
>
> **References:**
>
> [A] Yang, S., Jui, S., \& Van de Weijer, J. "Attracting and dispersing: A simple approach for source-free domain adaptation." NeurIPS, 2022.
>
> [B] Saito, K., Kim, D., Teterwak, P., Sclaroff, S., Darrell, T., \& Saenko, K. "Tune it the right way: Unsupervised validation of domain adaptation via soft neighborhood density." ICCV, 2021.
>
> [C] Hwang, U., Lee, J., Shin, J., \& Yoon, S. "SF(DA)$^2$: Source-free Domain Adaptation Through the Lens of Data Augmentation." ICLR, 2024.

---

> ### Author Response · Authors · 2024-11-21
> **Reply from the Authors**
>
> **Part 3/3**
>
> **Tables**
>
> **Table R1.** Performance comparisons under different values of $K_{PL}$.
>
> |     Dataset       | **$K_{PL}$** = 1     | **$K_{PL}$** = 2     | **$K_{PL}$** = 3     |
> |:---------------------|:-------:|:-------:|:-------:|
> | **VisDA-RUST**     | 79.04 | **79.20** | 79.09 |
> | **Office-Home (Pr → Cl)** | 61.48 | **61.56** | 61.42 |
>
> **Table R2.** Performance comparisons under different values of $\tau$.
>
> |      **Dataset**       | **$\tau$ = 1.1**   | **$\tau$ = 1.3**   | **$\tau$ = 1.5**   |
> |:---------------------|:-------:|:-------:|:-------:|
> | **Office-31**      | 90.34 | **90.55** | 90.38 |
> | **VisDA-RUST**     | **79.20** | 79.10 | 79.03 |
>
> **Table R3.** Performance comparison across different values of $L_{PL}$ on three datasets. **Bold** text indicates the results obtained with the best-performing hyperparameter set reported in the main paper, while $\underline{\text{underlined}}$ text highlights improved results found using finer hyperparameter tuning during the rebuttal phase. Acc. represents the accuracy on the full target dataset, while Acc. on Small Eval Set refers to the accuracy on a small, human-annotated evaluation subset.
>
> | Dataset | Strategy | $\lambda_{PL}$=0.0001 | $\lambda_{PL}$=0.001 | $\lambda_{PL}$=0.01 | $\lambda_{PL}$=0.05 | $\lambda_{PL}$=0.1 | $\lambda_{PL}$=0.15 | $\lambda_{PL}$=0.2 |
> |:---|:---|:---:|:---:|:---:|:---:|:---:|:---:|:---:|
> | VisDA-RUST | Acc. | 78.92  | 78.96  | 78.94  | **79.20** | $\underline{\text{79.40}}$ | $\underline{\text{79.53}}$ | 79.19  |
> | VisDA-RUST | Acc. On Small Eval Set | 65.33  | 68.00  | 69.67  | **70.33** | $\underline{\text{71.33}}$ | $\underline{\text{71.00}}$ | 70.00  |
> | VisDA-RUST | SND | 2.69  | 2.85  | 2.91  | **2.90** | $\underline{\text{2.63}}$ | $\underline{\text{2.38}}$ | 2.17  |
> | DomainNet126 (R $\to$ P) | Acc. | 70.52  | 70.53  | 70.86  | 70.98  | **71.10** | $\underline{\text{71.17}}$ | 70.89  |
> | DomainNet126 (R $\to$ P) | Acc. On Small Eval Set | 69.33  | 69.67  | 69.67  | 70.33  | **70.67** | $\underline{\text{70.00}}$ | 70.00  |
> | DomainNet126 (R $\to$ P) | SND | 1.93  | 1.88  | 1.90  | 1.82  | **1.78** | $\underline{\text{1.66}}$ | 1.64  |
> | Office-Home (Pr $\to$ Cl) | Acc. | 61.42  | **62.20** | 61.70  | 61.56  | 61.39  | 61.21  | 60.18  |
> | Office-Home (Pr $\to$ Cl) | Acc. On Small Eval Set | 62.17  | **62.33** | 61.33  | 60.33  | 61.33  | 61.00  | 61.00  |
> |Office-Home (Pr $\to$ Cl) | SND | 1.62  | **1.64** | 1.54  | 1.54  | 1.54  | 1.42  | 1.38  |
>
> **Table R4.** Performance comparison across different values of $L_{DC}$ on three datasets. **Bold** text indicates the results obtained with the best-performing hyperparameter set reported in the main paper, while $\underline{\text{underlined}}$ text highlights improved results found using finer hyperparameter tuning during the rebuttal phase. Acc. represents the accuracy on the full target dataset, while Acc. on Small Eval Set refers to the accuracy on a small, human-annotated evaluation subset.
>
> | Dataset | Strategy | $\lambda_{DC}$=0.05 | $\lambda_{DC}$=0.1 | $\lambda_{DC}$=0.5 | $\lambda_{DC}$=1 | $\lambda_{DC}$=1.5 | $\lambda_{DC}$=2 |
> |:---|:---|---|---|---|---|---|---|
> | VisDA-RUST | Acc. | $\underline{\text{79.51}}$ | $\underline{\text{79.82}}$ | **79.40** | 79.25  | 79.04  | 78.82  |
> | VisDA-RUST | Acc. On Small Eval Set | $\underline{\text{71.50}}$ | $\underline{\text{73.33}}$ | **73.33** | 69.00  | 67.33  | 69.00  |
> | VisDA-RUST | SND | $\underline{\text{3.47}}$ | $\underline{\text{3.13}}$ | **2.63** | 2.33  | 1.91  | 1.66  |
> | DomainNet126 (R $\to$ P) | Acc. | 70.10  | 70.58  | **71.10** | $\underline{\text{71.54}}$ | $\underline{\text{71.38}}$ | 70.94  |
> | DomainNet126 (R $\to$ P) | Acc. On Small Eval Set | 70.00  | 70.67  | **70.67** | $\underline{\text{70.33}}$ | $\underline{\text{70.33}}$ | 67.33  |
> | DomainNet126 (R $\to$ P) | SND | 1.98  | 1.96  | **1.78** | $\underline{\text{1.74}}$ | $\underline{\text{1.71}}$ | 1.76  |
> | Office-Home (Pr $\to$ Cl) | Acc. | 60.25  | 61.60  | **62.20** | 61.92  | 60.55  | 60.50  |
> | Office-Home (Pr $\to$ Cl) | Acc. On Small Eval Set | 60.83  | 61.50  | **62.33** | 61.17  | 60.83  | 60.50  |
> | Office-Home (Pr $\to$ Cl) | SND | 1.70  | 1.83  | **1.64** | 1.51  | 1.47  | 1.49  |
>
>
> **Table R5.** Performance comparison under different $\beta$ values on DomainNet126 (R → P).
> | $\beta$           | 0.25  | 0.5   | 0.75  | 1     | 5     |
> |:---------------------|-------|-------|-------|-------|-------|
> | **UCon-SFDA (Ours)** | 70.96 | 71.10 | 71.10 | 70.99 | 71.11 |
> | **Basic CL method** | 67.49 | 67.63 | 67.80 | 67.80 | 67.23 |

---

> ### Author Response · Authors · 2024-11-27
> **Looking forward to hearing from you**
>
> Dear Reviewer GdvL,
>
> We hope this message finds you well.
>
> We deeply appreciate your time and effort in reviewing our submission and providing valuable feedback. Your insights are crucial to our work.
>
> In our previous response, we carefully addressed the concerns you raised, including clarifying the hyperparameter setups and providing detailed hyperparameter sensitivity analyses.
>
> If there are any points that remain unclear, we would be glad to provide further clarification or engage in further discussion. We look forward to hearing from you.
>
> Thanks,
>
> The Authors of submission 8466

---

> > ### Comment · Reviewer_GdvL · 2024-11-27
> >
> > I appreciate the extensive results added to the author's response. These results can help mitigate but not resolve the concern on the hyperparameters, since these hyperparameters are embedded with the proposed method design. However, I understand that the hyperparameter tuning problem for unsupervised learning tasks such as source-free domain adaptation has been a long-standing challenge. Therefore, I decided to maintain my initial score.

---

> ### Author Response · Authors · 2024-11-28
> **Follow-up response: Theory-motivated hyper-parameter determination**
>
> Thank you for your thoughtful feedback and for recognizing the additional experimental results we provided. We agree that reducing the number of hyperparameters is crucial for enhancing the practicality of SFDA, particularly when adapting the algorithm to new tasks.
>
> To further the discussion, we provide a more detailed explanation below regarding how theoretical insights can guide the **direct selection** (**or derivation**) of hyperparameters. **We wish to assure you that our intention is not to add to your workload but to encourage meaningful discussion.** We hope this could further address your concerns about our algorithm and offer ideas for alleviating the long-standing challenges of hyperparameter tuning in this field.
>
> * (1) In designing the UCon-SFDA algorithm, we prioritized engineering flexibility and ease of implementation, which led us to introduce four hyperparameters. **However, three of these parameters have explicit expressions derived from our theoretical results or can be determined based on dataset and source model properties, thereby eliminating the need for manual hyperparameter tuning.** Specifically,
>
>   * $\lambda_{DC}$: Inconsistency rate (Motivated by Theorem 4.1 and Remark 4.2).
>
>     As suggested by Theorem 4.1 and Remark 4.2, the dispersion control effect can be achieved by minimizing the negative similarity between the anchor point and its augmented prediction. If the inconsistency rate between anchor points and their associated augmented predictions is high, it indicates greater uncertainty in negative sampling,  thus requiring stronger dispersion control. **Based on this observation, we propose directly using the model prediction inconsistency rate as the coefficient for the dispersion control term.**
>
>   * $K_{PL}$: $k_0$ (Theorem 4.2)
>
>     By Theorem 4.2, when the uncertainty set in Eq. (5) of our paper is defined using the 1-Wasserstein distance, the length of the partial label set, denoted by $K_{PL}$, can be explicitly determined as $K_{PL}=k_0$, where $k_0$ is defined as follows:
>
>     * (i) If $\frac{1}{K}\ge \frac{1}{k}\sum_{j=1}^{k}\mathcal{p}^{+}_{(j)}-\frac{1}{k}\delta$ for all $k\in[K-1]$, then we take $k_0=K$.
>     * (ii) Otherwise, we take the $k_0\in[K-1]$ that satisfies $\frac{1}{k_0}\sum\_{j=1}\^{k_0}\mathcal{p}\^{+}\_{(j)}-\frac{1}{k_0}\delta\ge\frac{1}{k}\sum\_{j=1}\^{k}\mathcal{p}\^{+}\_{(j)}-\frac{1}{k}\delta$ for all $k\in[K-1]$.
>
>     In the formulas above, $K$ represents the number of classes, $\mathcal{p}^{+}_{(j)}$ denotes the $j$-th largest predicted probability for the considered anchor point, and $\delta$ could be taken as $\frac{1}{K}$ as suggested by the proof of Theorem 4.2. **Hence, the length of the partial label set, which can be directly calculated, is determined by the model's predictions for the anchor point as well as the specific classification task at hand.**
>
>
>   * $\tau$: We propose two approaches to distinguish between certain and uncertain label information.
>
>     * (i) **Statistical Insights Approach**
>
>       This approach leverages the properties of the source model and the target data, combined with statistical insights. Specifically, we first use the source model to compute the predicted probabilities for each target data point. Next, we calculate the ratio of the two highest predicted probabilities for all data points and select the 10th percentile of these ratios as the value of $\tau$. This value selection allows us to summarize the data distribution and identify the 10% most uncertain data. The 10th percentile is chosen because it is a widely used measure in statistical research to analyze data distributions and highlight low-end values.
>
>     * (ii) **Theoretical Criterion Approach**
>
>       Alternatively, we can bypass the ratio of the two highest predicted probabilities and directly apply the criterion outlined in Remark 4.3 to distinguish between certain and uncertain label information. As discussed in Remark 4.3, in the special case where $\mathcal{p}\^{+}\_{(1)}\ge\max${$\frac{1}{K}+\delta,\mathcal{p}\^{+}\_{(2)}+\delta$}, we refer to it as **certain label information**. Conversely, if this condition is NOT satisfied, the label information is deemed **uncertain**, and the corresponding data is added to the uncertain data bank.
>
>
> * (2) We appreciate your insightful comments on addressing the hyperparameter tuning problem in SFDA, which help us extend our algorithm and more effectively apply our theoretical results to practical scenarios. **We will supplement our experiments following the aforementioned approach with direct hyperparameter determination and post the updated results in the comments as soon as they are available, hopefully within 2-3 days.**
>
> Thank you again for your valuable insights and for highlighting this important aspect of our work.

---

> > ### Author Response · Authors · 2024-12-01
> > **Follow-Up Experimental Results (1/2)**
> >
> > Thanks for your insightful question, which has guided us in exploring extensions of our previous algorithm toward an automatic version with fewer hyperparameters requiring manual tuning. These improvements, inspired by your valuable feedback, will further enhance the algorithmic contribution and the quality of our work.
> >
> > ---
> > ### **Notations and Explanation.**
> >
> > Specifically, building upon our previous response, we experimented with two extended methods, named $\color{blue}\text{autoUCon-SFDA (Theory)}$ and $\color{blue}\text{autoUCon-SFDA (Stat.)}$. These methods incorporate the following modifications:
> >
> > * $\lambda_{DC}$: The orginal hyperparameter $\lambda_{DC}$ (Orig. $\lambda_{DC}$) has been replaced by $\color{blue}\text{New}\ \lambda_{DC}$ which represents the inconsistency ratio between anchor points and their associated augmented predictions, derived by the source model.
> >
> > * $K_{PL}$: The original fixed $K_{PL}$ (Orig. $K_{PL}$) has been replaced by the calculated $\color{blue}k_0$, which is instance- and task-dependent (class category), self-adaptive during the training process, and computationally efficient, as described in our previous response.
> >
> > * $\tau$: We poroposed two alternatives for the fixed parameter $\tau$ (Orig. $\tau$):
> >
> >   * In the statistical insights approach ($\color{blue}\text{autoUCon-SFDA (Stat.)}$), $\color{blue}\tau_{s}$ is computed using the source model and fixed at the beginning of the adaptation process.
> >
> >   * In the theoretical criterion approach ($\color{blue}\text{autoUCon-SFDA (Theory)}$), $\color{blue}\tau_{t}$ is dynamically calculated based on the uncertain data selected in each epoch.
> >
> > For $\color{blue}k_0$ and $\color{blue}\tau_{t}$, we present their values in the first and the last training epochs to illustrate their changing trend, indicated by "$\color{blue}\text{Init.}$" and "$\color{blue}\text{Final}$" in the tables, respectively.
> >
> > ---
> > ### **Experimental Results.**
> >
> > We present the experimental results in **Tables R2-1, R2-2, and R2-3**. Specifically:
> >
> > * Performance comparisons between the original UCon-SFDA, the newly extended methods, and SOTA methods across all six benchmarks are shown in **Table R2-1**. **Notably, our findings validate that directly using theoretically derived parameters can achieve promising—and in some cases, superior—performance across all benchmarks.** (For the remaining three hyperparameters $\kappa$, $\beta$ and $\lambda_{PL}$, we kept them the same as those used in UCon-SFDA.)
> >
> > * A comprehensive parameter comparison is provided in **Table R2-2**. **It can be observed that the theoretically determined parameters are largely aligned with the hyperparameters used in UCon-SFDA. However, they offer greater flexibility in certain scenarios.**
> >
> >     * For instance, we present the **averaged values of $\color{blue}k_0$ at the initial and final training epochs**. Unlike the fixed $K_{PL}$, the instance-dependant $\color{blue}k_0$ automatically adapts throughout the adaptation process to better capture uncertainty.
> >
> >     * A similar self-adaptive behavior is observed for $\color{blue}\tau_{t}$.
> >
> > * Additionally, we present the per source-target task configuration on DomainNet126 to clearly illustrate parameter changes and their impact. For instance, as shown in the 6th coloumn of **Table R2-3**, the $\color{blue}\text{New}\ \lambda_{DC}$ is task-dependent, offering greater flexibility without requiring a manual selection process.
> >
> > ---
> > ### **Summary and Future Work Insights.**
> >
> > In summary, **(1)** the automatic versions of UCon-SFDA have demonstrated **promising performance while significantly reducing the number of hyperparameters** in the algorithm (retaining only three hyperparameters in autoUCon-SFDA, with just one directly related to our proposed methods). **(2)** The additional experimental results also illustrate the effectiveness of the uncertainty-guided parameter determination process. We believe that our theoretical framework **offers valuable insights into addressing the challenge of hyperparameter selection and tuning in UDA**.
> >
> > We would like to sincerely thank you once again for your valuable time spent reviewing our paper and for your insightful comments, which have guided us in further extending our method and enhancing its practicality for real-world SFDA tasks.

---

> ### Author Response · Authors · 2024-12-01
> **Follow-Up Experimental Results (2/2)**
>
> ### **Tables**
>
> **Table R2-1.** Performance comparisons across different hyper-parameter selection (calculation) methods. **Bold** text indicates the best results.
>
> |  Dataset  |  UCon-SFDA  |  autoUCon-SFDA (Theory)  |  autoUCon-SFDA (Stat.)  |  SOTA Method Performance  |  SOTA Method |
> |:---------------------|:-------:|:-------:|:-------:| :-------:|:-------:|
> |  **Office31**  |  **90.6**  |  **90.6**  |  90.2  |  90.5  |  C-SFDA  |
> |  **OfficeHome**  |  73.6  |  73.6   |  **73.8**  |  73.5  |  C-SFDA  |
> |  **OfficeHome (partial set)**  |  80.3  |  **80.8**  |  80.7   |  79.7  |  AaD  |
> |  **VisDA2017**  |  **89.6**  |  89.3   |  89.2   |  88.4  |  I-SFDA  |
> |  **VisDA-RUST**  |  79.4  |  79.2  |  **79.5**  |  77.3  |  SF(DA)$^2$  |
> |  **DomainNet126**  |  71.5  |  71.5  |  **71.6**  |  69.6  |  GPUE  |
>
>
> **Table R2-2.** Hyper-parameter values across different datasets. "Orig. $\lambda_{DC}$", "Orig. $K_{PL}$", and "Orig. $\tau$" refer to the original values used in our paper, which are selected following the general hyper-parameter tuning pipeline in the literature. $\color{blue}\text{The hyper-parameters highlighted in blue}$ are directly calculated with theory-motivated hyper-parameter determination approaches, where "$\color{blue}\text{Init.}$" and "$\color{blue}\text{Final}$" indicate the first and the last training epochs, respectively. $\color{green}\text{The text in green}$ specifies the associated selection/calculation methods.
>
> |  Metric  |  Office31  |  OfficeHome  |  OfficeHome (partial set)  |  VisDA2017  |  VisDA-RUST  |  DomainNet126  |
> |:---------------------|:-------:|:-------:|:-------:| :-------:|:-------:|:-------:|
> |  Orig. $\lambda_{DC}\ \color{green}\text{(Original value used in our paper)}$  |  1.000  |  0.500  |  1.000  |  1.000  |  0.500  |  0.500  |
> |  $\color{blue}\text{New}\ \lambda_{DC}\ \color{green}\text{(Inconsistency Ratio: Guided by Theorem 4.1)}$  |  0.390  |  0.520  |  0.476  |  0.494  |  0.461  |  0.553  |
> |  Orig. $K_{PL}\ \color{green}\text{(Original value used in our paper)}$  |  2.000  |  2.000  |  2.000  |  1.000  |  2.000  |  2.000  |
> |  $\color{blue}\text{Init.}\ k_0\ \text{(Averaged)}\ \color{green}(k_0\ \text{in Theorem 4.2})$  |  1.320  |  1.535  |  1.513  |  1.341  |  1.348  |  1.644  |
> |  $\color{blue}\text{Final.}\ k_0\ \text{(Averaged)}\ \color{green}(k_0\ \text{in Theorem 4.2})$  |  1.003	 |  1.028	|  1.003  |	1.008  |	1.020  |	1.079
> |  Orig. $\tau\ \color{green}\text{(Original value used in our paper)}$  |  1.300	  |  1.100  |	1.100  |	1.100  |	1.100  |	1.100  |
> |  $\color{blue}\text{Init.}\ \tau_{t}\ \color{green}\text{(Calculated Using Theoretical Criterion: Remark 4.3)}$  |  1.308  |	1.265  |	1.238  |	1.790  |	1.674  |	1.232  |
> |  $\color{blue}\text{Final}\ \tau_{t}\ \color{green}\text{(Calculated Using Theoretical Criterion: Remark 4.3)}$  |  1.056  |	1.090  |	1.042  |	1.260  |	1.368  |	1.092  |
> |  $\color{blue}\tau_{s}\ \color{green}\text{(Derived from Statistical Insights: 10th percentile)}$  |  2.037  | 	1.230  |	1.268  |	1.164  |	1.163  |	1.264  |
>
> **Table R2-3.** Per source-target task configuration on DomainNet126. The metric notations are the same as in **Tables R2-2**.
>
> | Task | Acc. of Ucon-SFDA | $\color{blue}\text{Acc. of}$  $\color{blue}\text{autoUCon-SFDA}$ $\color{blue}\text{(Theory)}$ |$\color{blue}\text{Acc. of}$  $\color{blue}\text{autoUCon-SFDA}$ $\color{blue}\text{(Stat.)}$ | Orig. $\lambda_{DC}$  | $\color{blue}\text{New}\ \lambda_{DC}$ | Orig. $K_{PL}$ | $\color{blue}\text{Init.}\ k_0$ $\color{blue}\text{(Averaged)}$  | $\color{blue}\text{Final}\ k_0$ $\color{blue}\text{(Averaged)}$   | Orig. $\tau$ | $\color{blue}\text{Init.}\ \tau_{t}$        | $\color{blue}\text{Final}\ \tau_{t}$       | $\color{blue}\tau_{s}$  |
> |:---:|:---:|:---:|:---:|:---:|:---:|:---:|:---:|:---:|:---:|:---:|:---:|:---:|
> | **C$\to$S** | 66.5  | 64.5  | 66.0  | 0.50  | 0.52  | 2 | 1.70  | 1.08  | 1.1 | 1.20  | 1.08  | 1.23  |
> | **P$\to$C** | 69.3  | 70.3  | 70.0  | 0.50  | 0.59  | 2 | 2.33  | 1.11  | 1.1 | 1.30  | 1.11  | 1.17  |
> | **P$\to$R** | 81.0  | 81.4  | 81.4  | 0.50  | 0.45  | 2 | 1.64  | 1.04  | 1.1 | 1.28  | 1.08  | 1.36  |
> | **R$\to$C** | 75.2  | 77.0  | 77.3  | 0.50  | 0.59  | 2 | 1.45  | 1.08  | 1.1 | 1.19  | 1.09  | 1.27  |
> | **R$\to$P** | 71.1  | 71.3  | 71.0  | 0.50  | 0.58  | 2 | 1.39  | 1.09  | 1.1 | 1.17  | 1.11  | 1.32  |
> | **R$\to$S** | 64.3  | 68.1  | 67.7  | 0.50  | 0.61  | 2 | 1.52  | 1.07  | 1.1 | 1.20  | 1.09  | 1.23  |
> | **S$\to$P** | 68.1  | 67.9  | 67.6  | 0.50  | 0.55  | 2 | 1.49  | 1.08  | 1.1 | 1.30  | 1.08  | 1.27  |
> | **Avg.** | 71.5  | 71.5  | 71.6  | 0.50  | 0.55  | 2 | 1.64  | 1.08  | 1.1 | 1.23  | 1.09  | 1.26  |

---

### Official Review · Reviewer_Bkmw · 2024-11-06

**Soundness:** 3
**Presentation:** 2
**Contribution:** 3
**Rating:** 5
**Confidence:** 2

**Summary:**

This paper presents a novel approach for source-free domain adaptation (SFDA), grounded in an uncertainty-guided theoretical analysis of contrastive learning-based SFDA methods. We introduce a distributionally robust optimization framework to elucidate the role of uncertainty. Additionally, the method incorporates augmentation-driven dispersion control and an optimal solution for partial label sets within a contrastive learning-based SFDA approach. The proposed method is thoroughly evaluated across four benchmark datasets.

**Strengths:**

- This paper induces a new aspect to solve the source-free domain adaptation problem.
- It induces an efficient method based on the guide of the proposed theoretical analysis.
- The experimental results prove the advantage of the proposed method

**Weaknesses:**

- Intuitively, applying cross-entropy loss on uncertain target samples with estimated pseudo labels can be problematic and may lead to negative transfer due to noisy pseudo label information, as certain samples are more likely to have accurate pseudo labels than uncertain ones..
- There are seven hyper-parameters, as shown in Table 6, which suggests that the proposed method requires specific parameter tuning to achieve promising results.
- The data augmentation alignment in Eq. 7 is, in fact, a consistency regularization between the weak and strong augmentations of the same data, which is not a novel technique as it has been well studied.

**Questions:**

See weaknesses

---

> ### Author Response · Authors · 2024-11-21
> **Reply from the Authors**
>
> **Part 1/6**
>
> We appreciate your detailed feedback. Below, we address the concerns and questions raised in the weaknesses section. Please feel free to reach out if further clarification is required.
>
> ### **1. About the loss on certain/uncertain data ($\mathcal{L}\^{+}\_{CL}$ and $\mathcal{L}\^{+}\_{PL}$)**
>
> We thank you  for this insightful comment regarding the potential issues of directly applying cross-entropy loss to uncertain pseudo-labels. We agree that they can indeed lead to negative transfer due to noisy pseudo-label information. **In fact, rather than being a weakness, your concern aligns precisely with the motivation and contribution of our work, as it has directly motivated the development of our approach, particularly the incorporation of the positive uncertainty control loss term ($\mathcal{L}\^{+}\_{UCon} = \mathcal{L}\^{+}\_{CL} + \lambda_{PL}\mathcal{L}\^{+}\_{PL}$) introduced in Section 4.3 as detailed in Eqs. (8)-(9).** Below, we highlight the key features of our method, focusing on its role in mitigating the negative impact of noisy pseudo-label information, which notably sets our developed algorithm apart from existing ones.
>
> * **Motivation of Our Work.**
>
>     In previous SFDA research, methods relying on neighboring information often risk introducing harmful supervision signals by ignoring prediction uncertainty and including incorrect neighbors, thereby amplifying the challenges associated with noisy pseudo-labels. To offset the negative impact of noisy pseudo-label information in SFDA, our method takes a fundamentally different approach by **carefully analyzing and explicitly utilizing prediction uncertainty**, rather than neglecting it or relying on it blindly as  has been the case in prior research.
>
> * **Theoretical and Methodological Innovations.**
>
>     As indicated by Theorem 4.2 and Eq. (8) in our paper, **our framework distinguishes between certain and uncertain label information and leverages them in distinct ways to ensure robust supervision:**
>
>     * **(i) Leverage certain label information: positive supervision loss $\mathcal{L}^{+}_{CL}$.**
>
>         When an instance $\mathbf{x}$ receives a clear and confident supervision signal, the optimal prediction for $\mathbf{x}$ corresponds to the label with the highest predicted probability. This certain label information is incorporated via the **positive supervision loss** term, $\mathcal{L}^{+}_{CL}$, as defined in Eqs. (8)-(9).
>
>     * **(ii) Leverage uncertain label information: partial label loss $\mathcal{L}^{+}_{PL}$.**
>
>         For instances with uncertain predictions, simply relying on single estimated pseudo-labels would risk amplifying noise. Instead, we construct an instance-dependent **partial label set**, $\mathscr{Y}\_{PL,i}$ to represent multiple plausible labels. This approach provides a more robust supervisory signal by accounting for multiple potential labels and reducing reliance on noisy single-label predictions. The uncertain label information is captured through the **partial label loss** term, $\mathcal{L}\^{+}\_{PL}$, as detailed in Eqs. (8)-(9).
>
> * **Smoothed cross-entropy loss applied to the partial label set.**
>
>     Rather than directly applying the cross-entropy loss to a single estimated pseudo-label, we employ a **smoothed cross-entropy loss applied to the partial label set**. This approach enhances the discriminability of the source model and facilitates better alignment with the target data [A], and the partial label set could benefit the uncertain data training process. As noted in partial label learning literature (e.g., [A], [B]), this loss format is particularly effective in mitigating the impact of noisy labels, and we demonstrate its utility for the SFDA problem both theoretically and empirically in our paper, making it a significant and non-trivial contribution.
>
> * **Experimental observation of more accurate supervision signals provided from the partial label set.**
>
>     **Figure 1(c)** in our paper highlights that leveraging partial labels offers more accurate supervisory signals for uncertain data compared to those solely neighborhood-based methods. This improvement results in better performance and underscores the effectiveness of our approach. Our extensive experiments demonstrate our method’s robustness and highlight its ability to mitigate the adverse effects of noisy pseudo-labels, achieving superior alignment with the target data.

---

> ### Author Response · Authors · 2024-11-21
> **Reply from the Authors**
>
> **Part 2/6**
>
> ### **2. About the hyper-parameters.**
>
> * >"There are seven hyper-parameters, as shown in Table 6"
>
>     While we list seven hyperparameters in **Table 6** for completeness and ease of reference, it is important to note that **only four hyperparameters are directly tied to our proposed method**. These parameters fall into three categories:
>
>     * **(i) $\kappa$, $\lambda^{-}_{CL}$, and $\beta$: Inherited from Previous Work.**
>
>         The three hyperparameters ($\kappa$, $\lambda^{-}_{CL}$, and $\beta$) in the basic contrastive learning loss **were adopted directly from previous works [C, D]** to ensure consistency and comparability.
>
>     * **(ii) $K_{PL}$ and $\tau$: Selected under Theoretical Guidance.**
>
>         Although we introduce additional hyperparameters ($K_{PL}$ and $\tau$) in the partial label loss term, selecting them is not difficult as it is guided by our theoretical results. Specifically, the theoretical results in Theorem 4.2 suggest that the partial label loss should be applied to most uncertain data while minimizing additional label noise. Therefore, a relatively small value for $K_{PL}$ (e.g., 1 or 2) and $\tau$ (slightly larger than 1) can be chosen, which are also confirmed by our empirical studies, as shown in **Tables R1-R2** and **Figure 17** in the revised manuscript (**Appendix C.5**).
>
>         **Table R1.** Performance comparisons under different values of $K_{PL}$.
>
>         |     Dataset       | **$K_{PL}$** = 1     | **$K_{PL}$** = 2     | **$K_{PL}$** = 3     |
>         |:---------------------|:-------:|:-------:|:-------:|
>         | **VisDA-RUST**     | 79.04 | **79.20** | 79.09 |
>         | **Office-Home (Pr → Cl)** | 61.48 | **61.56** | 61.42 |
>
>         **Table R2.** Performance comparisons under different values of $\tau$.
>
>         |      **Dataset**       | **$\tau$ = 1.1**   | **$\tau$ = 1.3**   | **$\tau$ = 1.5**   |
>         |:---------------------|:-------:|:-------:|:-------:|
>         | **Office-31**      | 90.34 | **90.55** | 90.38 |
>         | **VisDA-RUST**     | **79.20** | 79.10 | 79.03 |
>
>     * **(iii) $\lambda_{PL}$ and $\lambda_{DC}$: Empirically Selected yet NOT Finely Tuned.**
>
>         We followed the hyperparameter selection principle employed in prior studies [C,D], and only searched for two loss coefficients, $\lambda_{PL}$ and $\lambda_{DC}$, within reduced ranges: {$0.5, 1$} for $\lambda_{DC}$ and {$0.001, 0.01, 0.05, 0.1$} for $\lambda_{PL}$. To ensure computational feasibility, we did not perform exhaustive combinatorial tuning by considering all possible parameter combinations, although better performance might be achievable with finer-grained searches. The tuning strategy is further elaborated  in the response to the following bullet point, where   additional sensitivity analyses on $\lambda_{PL}$ and $\lambda_{DC}$ are also provided.

---

> ### Author Response · Authors · 2024-11-21
> **Reply from the Authors**
>
> **Part 3/6**
>
>  * >"which suggests that the proposed method requires specific parameter tuning to achieve promising results."
>
>     While the proposed method requires parameter tuning, **we would like to clarify that no combinatorial over-tuning or excessive optimization was performed. Instead, we adopted a sequential and incremental tuning strategy.**
>
>     * **(i) Stage-Wise Tuning Strategy:**
>
>         Specifically, we began by adopting $\kappa$, $\lambda\^{-}\_{CL}$ and $\beta$ from previous works [A,C]. We then incrementally added the dispersion control term and determined the best $\lambda_{DC}$. Finally, we introduced the partial label term, identified the most suitable $K_{PL}$ and $\tau$ based on the dataset properties, and tuned $\lambda_{PL}$. For datasets with multiple source-target pairs (e.g., DomainNet126, Office-31 and Office-Home), we tuned hyperparameters on one or two sub-tasks (such as R $\to$ P task on DomainNet126) and applied the same set of hyperparameters across the entire dataset. Such hyperparameter searching schema is straightforward and not computationally intensive.
>
>     * **(ii) Sensitivity Analyses and Further Validation:**
>
>         We conducted additional sensitivity analyses during the rebuttal period, exploring finer-grained values for $\lambda_{DC}$ and $\lambda_{PL}$. As shown in **Tables R3-R4** and **Figures 18-19** in the revised manuscript (**Appendix C.5**), **slight adjustments led to further performance improvements**. While we initially reported the UCon-SFDA performance of 79.4 on VisDA-RUST (with $L_{PL} = 0.1$ and $L_{DC} = 0.5$), we found that using a slightly smaller $L_{DC} = 0.1$ improved its performance to 79.82. A similar trend was observed for DomainNet126, where better performance was achieved under $L_{DC} = $1 or $1.5$. These experimental results validate the effectiveness and the potential for further optimization of our proposed method, even without exhaustive tuning.
>
>         **Table R3.** Performance comparison across different values of $L_{PL}$ on three datasets. **Bold** text indicates the results obtained with the best-performing hyperparameter set reported in the main paper, while $\underline{\text{underlined}}$ text highlights improved results found using finer hyperparameter tuning during the rebuttal phase.
>         | Dataset | $\lambda_{PL}$=0.0001 | $\lambda_{PL}$=0.001 | $\lambda_{PL}$=0.01 | $\lambda_{PL}$=0.05 | $\lambda_{PL}$=0.1 | $\lambda_{PL}$=0.15 | $\lambda_{PL}$=0.2 |
>         |:---|:---:|:---:|:---:|:---:|:---:|:---:|:---:|
>         | VisDA-RUST | 78.92  | 78.96  | 78.94  | **79.20** | $\underline{\text{79.40}}$ | $\underline{\text{79.53}}$ | 79.19  |
>         | DomainNet126 (R $\to$ P) | 70.52  | 70.53  | 70.86  | 70.98  | **71.10** | $\underline{\text{71.17}}$ | 70.89  |
>         | Office-Home (Pr $\to$ Cl) | 61.42  | **62.20** | 61.70  | 61.56  | 61.39  | 61.21  | 60.18  |
>
>         **Table R4.** Performance comparison across different values of $L_{DC}$ on three datasets. **Bold** text indicates the results obtained with the best-performing hyperparameter set reported in the main paper, while $\underline{\text{underlined}}$ text highlights improved results found using finer hyperparameter tuning during the rebuttal phase.
>
>         | Dataset | $\lambda_{DC}$=0.05 | $\lambda_{DC}$=0.1 | $\lambda_{DC}$=0.5 | $\lambda_{DC}$=1 | $\lambda_{DC}$=1.5 | $\lambda_{DC}$=2 |
>         |:---|---|---|---|---|---|---|
>         | VisDA-RUST | $\underline{\text{79.51}}$ | $\underline{\text{79.82}}$ | **79.40** | 79.25  | 79.04  | 78.82  |
>         | DomainNet126 (R $\to$ P) | 70.10  | 70.58  | **71.10** | $\underline{\text{71.54}}$ | $\underline{\text{71.38}}$ | 70.94  |
>         | Office-Home (Pr $\to$ Cl) | 60.25  | 61.60  | **62.20** | 61.92  | 60.55  | 60.50  |
>
> * In summary, while experiments demonstrate that finer-grained searches are possible to enhance performance, our UCon-SFDA algorithm achieves satisfactory results with lightweight tuning on a subset of hyperparameters guided by theoretical insights, without requiring combinatorial over-tuning. In response to your comments, we have incorporated the above discussion into the revised manuscript for clarity.

---

> ### Author Response · Authors · 2024-11-21
> **Reply from the Authors**
>
> **Part 4/6**
>
> ### **3. About data augmentation alignment in $\mathcal{L}^{-}_{DC}$**
>
> We agree with your comments on the importance and widespread use of consistency loss in the SFDA setting. However, our contributions significantly extend beyond existing works. In a nutshell, our theoretical results bridge the gap between theory and applications, providing theoretical insights for the observed performance improvements.
>
> Specifically, **(i) through a rigorous theoretical analysis of uncertainty in SFDA, we naturally derive a dispersion control term that aligns with consistency loss. This derivation provides a solid theoretical explanation for the effectiveness of data augmentation-based consistency loss in improving SFDA performance. (ii) Furthermore, our proposed dispersion control term offers a unified framework that accommodates various loss formats, including the data augmentation alignment loss format, which provides deeper insights and guidance for future research on SFDA methodologies.**
>
> Further details are explained below.
>
> * **(i) Strong theoretical foundation.**
>
>     * **Introduction of the dispersion control term in SFDA.**
>
>         Our work goes beyond the conventional application of consistency loss by providing a solid theoretical foundation. Specifically, grounded in DRO and uncertainty perspectives, our theoretical findings (outlined in Theorem 4.1) reveal that controlling the robust risk in the presence of potential false negative samples requires not only addressing the widely studied negative sample loss, denoted as $\mathcal{L}\^{-}\_{CL}$ in Eq. (7) [C, D], but also incorporating an additional dispersion term $\mathcal{V}\_{\mathscr{d}}$, which plays a crucial role in achieving improved performance. Below, we elaborate on how the dispersion control term $\mathcal{V}_{\mathscr{d}}$ is utilized to derive the loss term $\mathcal{L}\^{-}\_{DC}$ in Eq. (7).
>
>     * **Explanation of using data augmentations in dispersion control term.**
>
>         In applications, domain shift makes it challenging to distinguish false negatives from true negatives. To address this, as illustrated in **Figures 2 (b)-(c)**, we propose to achieve the dispersion control effect by minimizing the negative similarity between an instance $\mathbf{x}$ and its manually constructed pseudo-false negatives, which are incorrectly identified by the model as negative samples but should belong to the same class as $\mathbf{x}$. Specifically, as shown in **Figure 1(b)**, we observe that the source model's prediction on the augmented version of $\mathbf{x}$, denoted as $\mathtt{AUG}(\mathbf{x})$, may not align with the prediction for $\mathbf{x}$. In this context, the augmented versions of $\mathbf{x}$, denoted as $\mathtt{AUG}(\mathbf{x})$, naturally serve as candidates for these pseudo-false negatives. This explains why the dispersion control effect can be captured by the loss term  $\mathcal{L}^{-}_{DC}$ in Eq. (7), and also highlights the benefit of using $\mathtt{AUG}(\mathbf{x})$ over alternative representations, such as the neighbors of $\mathbf{x}$, for constructing pseudo-false negatives.

---

> ### Author Response · Authors · 2024-11-21
> **Reply from the Authors**
>
> **Part 5/6**
>
> * **(ii) Unified framework.**
>
>     Beyond its specific implementation in Eq. (7), our theoretical analysis provides a unified framework that accommodates various loss formats designed to achieve the dispersion control effect. In this paper, **while we instantiate this framework using a data augmentation alignment loss format, our development can also support other loss formulations** by incorporating alternative distance metrics and diverse techniques for constructing pseudo-false negative examples.
>
>     * **Utilization of alternative distance measurements.**
>
>         In our main implementation, we minimize the cosine similarity between network output of $\mathbf{x}$ and the $\log$ probabilities of $\mathtt{AUG}(\mathbf{x})$ to achieve the dispersion control effect. However, this effect can also be realized using other distance metrics, such as the $L^2$ norm or the direct dot product. Experimental results in **Table R5** demonstrate that all loss formulations with different distance metrics improve SFDA performance compared to the baseline method ($\mathcal{L}_{CL}$). In some cases, loss formulations using these alternative distance metrics could achieve even better performance (such as dot product-based dispersion control loss on the Office-Home dataset).
>
>
>         **Table R5:** Classification Accuracy (%) Under different Distance Measurements in Dispersion Control term. **Bold** text indicates the best results, and $\underline{\text{underlined}}$ text represents results that outperform the baseline.
>
>         | Methods                                      | Office-Home (Pr → Cl) | VisDA-RUST | DomainNet126 (R → P) |
>         |---------------------------------------------|------------------------|------------|-----------------------|
>         | $\mathcal{L}\_{CL}$                          | 57.90                 | 75.50      | 67.80                |
>         | $\mathcal{L}\_{CL} + \mathcal{L}\^{-}\_{DC}$ with $\mathbb{d}\_{\theta}$ | $\underline{\text{59.70}}$ | **78.90** | **70.30** |
>         | $\mathcal{L}\_{CL} + \mathcal{L}\^{-}\_{DC}$ with $\mathbb{d}\_{\theta, \text{dot}}$ | **60.21** | $\underline{\text{78.02}}$ | $\underline{\text{70.08}}$ |
>         | $\mathcal{L}\_{CL} + \mathcal{L}\^{-}\_{DC}$ with $\mathbb{d}\_{\theta, L^2}$ | $\underline{\text{59.14}}$ | $\underline{\text{77.77}}$ | $\underline{\text{69.34}}$ |
>
>     * **Advanced techniques for constructing pseudo-false negatives.**
>
>         Furthermore, pseudo-false negative examples can be constructed in various ways, enabling integration with advanced techniques such as mixup or large-scale pre-trained models. This adaptability highlights the broad applicability of our approach in the SFDA literature and creates opportunities for future research.
>
> We have included the above discussion and additional experimental results in the revised manuscript (**Appendix C.6**).

---

> ### Author Response · Authors · 2024-11-21
> **Reply from the Authors**
>
> **Part 6/6**
>
> ### **4. Other changes in our revision during the rebuttal.**
>
> To better articulate the Introduction and related sections, we have revised these parts in the manuscript to more clearly highlight the novelty and significance of our approach, as well as to improve the clarity of the presentation. Thank you for your thoughtful comments, which helped enhance the presentation of our work.
>
>
> For ease of your review, here we include  key changes in response to your comments; other revisions can be found in Remarks 4.1-4.4, Figure 2, Appendix A.1, Appendix B, Appendix C.5, and Appendix C.6:
>
> * We have streamlined the theoretical analysis, added remarks connecting theory to practice (**Remarks 4.1-4.4**), revised **Figure 2** for a clearer visualization of the dispersion control effect, and included a notation table for clarity in the revised manuscript (**Appendix A.1**).
> * We have elaborated on how our theoretical findings guided the design of the consistency loss (**Remarks 4.1 and 4.2**) and included additional experiments to demonstrate its advantages (**Appendix C.6**).
> * We have clarified the hyperparameters directly related to our method and provided a more detailed explanation of our stage-wise tuning strategy, resulting in effective hyperparameter selections (**Appendix B**). We have also provided further experimental results on hyperparameter sensitivity analysis (**Appendix C.5**).
>
>
>
> **References**
>
> [A] Liang, J., Hu, D., \& Feng, J. "Do we really need to access the source data? source hypothesis transfer for unsupervised domain adaptation." ICML, 2020.
>
> [B] Wen, H., Cui, J., Hang, H., Liu, J., Wang, Y., \& Lin, Z. "Leveraged weighted loss for partial label learning." ICML, 2021.
>
> [C] Yang, S., Jui, S., \& Van de Weijer, J. "Attracting and dispersing: A simple approach for source-free domain adaptation." NeurIPS, 2022.
>
> [D] Hwang, U., Lee, J., Shin, J., \& Yoon, S. "SF(DA)$^2$: Source-free Domain Adaptation Through the Lens of Data Augmentation." ICLR, 2024.
>
> [E] Saito, K., Kim, D., Teterwak, P., Sclaroff, S., Darrell, T., \& Saenko, K. "Tune it the right way: Unsupervised validation of domain adaptation via soft neighborhood density." ICCV, 2021.

---

> ### Author Response · Authors · 2024-11-27
> **Looking forward to hearing from you**
>
> Dear Reviewer Bkmw,
>
> We hope this message finds you well.
>
> We deeply appreciate your time and effort in reviewing our submission and providing valuable feedback. Your insights are crucial to our work.
>
> In our previous response, we carefully addressed the concerns you raised, including clarifying the hyperparameter setups, providing more detailed explanations of our theoretical findings, and offering more comprehensive experimental justifications.
>
> If there are any points that remain unclear, we would be glad to provide further clarification or engage in further discussion. We look forward to hearing from you.
>
> Thanks,
>
> The Authors of submission 8466

---

> ### Author Response · Authors · 2024-12-01
> **Follow-Up Response on Hyperparameter Determination (1/3)**
>
> Thank you once again for your valuable comments on the hyperparameter selection process. Inspired by the discussion with another reviewer, we have extended our previous algorithm into **two automatic versions with fewer hyperparameters requiring manual tuning**. These extensions aim to thoroughly optimize the incorporation of parameters in our method, enhancing its efficiency and adaptability.
>
> ---
> ### **Theory-Motivated Hyper-Parameter Determination and New Notations.**
>
> In designing the UCon-SFDA algorithm, we prioritized engineering flexibility and ease of implementation, which led us to introduce four hyperparameters. **However, three of these parameters have explicit expressions derived from our theoretical results or can be determined based on dataset and source model properties, thereby eliminating the need for manual hyperparameter tuning.** Specifically,
>
> * $\lambda_{DC}$: Inconsistency Ratio (Motivated by Theorem 4.1 and Remark 4.2).
>
>     As suggested by Theorem 4.1 and Remark 4.2, the dispersion control effect can be achieved by minimizing the negative similarity between the anchor point and its augmented prediction. If the inconsistency rate between anchor points and their associated augmented predictions is high, it indicates greater uncertainty in negative sampling,  thus requiring stronger dispersion control. **Based on this observation, we propose directly using the model prediction inconsistency ratio (denoted as "$\color{blue}\text{New}\ \lambda_{DC}$") as the coefficient for the dispersion control term.**
>
> * $K_{PL}$:$\color{blue}k_0$ (Theorem 4.2)
>
>   By Theorem 4.2, when the uncertainty set in Eq. (5) of our paper is defined using the 1-Wasserstein distance, the length of the partial label set, denoted by $K_{PL}$, can be explicitly determined as $K_{PL}=$$\color{blue}k_0$, where $\color{blue}k_0$ is defined as follows:
>
>   * (i) If $\frac{1}{K}\ge \frac{1}{k}\sum_{j=1}^{k}\mathcal{p}^{+}_{(j)}-\frac{1}{k}\delta$ for all $k\in[K-1]$, then we take $\color{blue}k_0$$=K$.
>   * (ii) Otherwise, we take the $\color{blue}k_0$$\in[K-1]$ that satisfies $\frac{1}{\color{blue}{k\_0}}\sum\_{j=1}\^{\color{blue}{k\_0}}\mathcal{p}\^{+}\_{(j)}-\frac{1}{\color{blue}{k\_0}}\delta\ge\frac{1}{k}\sum\_{j=1}\^{k}\mathcal{p}\^{+}\_{(j)}-\frac{1}{k}\delta$ for all $k\in[K-1]$.
>
>   In the formulas above, $K$ represents the number of classes, $\mathcal{p}^{+}_{(j)}$ denotes the $j$-th largest predicted probability for the considered anchor point, and $\delta$ could be taken as $\frac{1}{K}$ as suggested by the proof of Theorem 4.2. **Hence, the length of the partial label set, which can be directly calculated, is determined by the model's predictions for the anchor point as well as the specific classification task at hand.**
>
>
> * $\tau$: We propose two approaches to distinguish between certain and uncertain label information.
>
>   * (i) **Statistical Insights Approach - $\color{blue}{\tau_{s}}$**
>
>     This approach leverages the properties of the source model and the target data, combined with statistical insights. Specifically, we first use the source model to compute the predicted probabilities for each target data point. Next, we calculate the ratio of the two highest predicted probabilities for all data points and **select the 10th percentile of these ratios as the value of $\tau$, denoted as $\color{blue}{\tau_{s}}$** in the updated tables. This value selection allows us to summarize the data distribution and identify the 10% most uncertain data. The 10th percentile is chosen because it is a widely used measure in statistical research to analyze data distributions and highlight low-end values.
>
>   * (ii) **Theoretical Criterion Approach - $\color{blue}{\tau_{t}}$**
>
>     Alternatively, we can bypass the ratio of the two highest predicted probabilities and directly apply the criterion outlined in Remark 4.3 to distinguish between certain and uncertain label information. As discussed in Remark 4.3, in the special case where $\mathcal{p}^{+}_{(1)}\ge\max\{\frac{1}{K}+\delta,\mathcal{p}^{+}_{(2)}+\delta\}$, we refer to it as **certain label information**. Conversely, if this condition is NOT satisfied, the label information is deemed **uncertain**, and the corresponding data is added to the uncertain data bank. **Based on the selected uncertain data, we calculate a corresponding ratio $\color{blue}{\tau_{t}}$**, as reported in the updated tables, for post-comparison purposes.
>
> ---
> Building upon different uncertain data selection strategies, we proposed two automatic UCon-SFDA methods: $\color{blue}\text{autoUCon-SFDA (Theory)}$ and $\color{blue}\text{autoUCon-SFDA (Stat.)}$. For $\color{blue}k_0$ and $\color{blue}\tau_{t}$, we present their values in the first and the last training epochs to illustrate their changing trend, indicated by "$\color{blue}\text{Init.}$" and "$\color{blue}\text{Final}$" in the tables, respectively.

---

> > ### Author Response · Authors · 2024-12-01
> > **Follow-Up Response on Hyperparameter Determination (2/3)**
> >
> > ---
> > ### **Experimental Results.**
> >
> > We present the experimental results in **Tables R2-1, R2-2, and R2-3**. Specifically:
> >
> > * Performance comparisons between the original UCon-SFDA, the newly extended methods, and SOTA methods across all six benchmarks are shown in **Table R2-1**. **Notably, our findings validate that directly using theoretically derived parameters can achieve promising—and in some cases, superior—performance across all benchmarks.** (For the remaining three hyperparameters $\kappa$, $\beta$ and $\lambda_{PL}$, we kept them the same as those used in UCon-SFDA.)
> >
> > * A comprehensive parameter comparison is provided in **Table R2-2**. **It can be observed that the theoretically determined parameters are largely aligned with the hyperparameters used in UCon-SFDA. However, they offer greater flexibility in certain scenarios.**
> >
> >     * For instance, we present the **averaged values of $\color{blue}k_0$ at the initial and final training epochs**. Unlike the fixed $K_{PL}$, the instance-dependant $\color{blue}k_0$ automatically adapts throughout the adaptation process to better capture uncertainty.
> >
> >     * A similar self-adaptive behavior is observed for $\color{blue}\tau_{t}$.
> >
> > * Additionally, we present the per source-target task configuration on DomainNet126 to clearly illustrate parameter changes and their impact. For instance, as shown in the 6th coloumn of **Table R2-3**, the $\color{blue}\text{New}\ \lambda_{DC}$ is task-dependent, offering greater flexibility without requiring a manual selection process.
> >
> > ---
> > ### **Summary and Future Work Insights.**
> >
> > In summary, **(1)** the automatic versions of UCon-SFDA have demonstrated **promising performance while significantly reducing the number of hyperparameters** in the algorithm (retaining only three hyperparameters in autoUCon-SFDA, with just one directly related to our proposed methods). **(2)** The additional experimental results also illustrate the effectiveness of the uncertainty-guided parameter determination process. We believe that our theoretical framework **offers valuable insights into addressing the challenge of hyperparameter selection and tuning in UDA**.
> >
> > We sincerely thank you once again for your valuable time and effort in reviewing our paper. We hope that this follow-up response regarding hyperparameter determination, along with the additional experimental results, adequately addresses your concerns about the parameters used in our experiments.

---

> > > ### Author Response · Authors · 2024-12-01
> > > **Follow-Up Response on Hyperparameter Determination (3/3)**
> > >
> > > ### **Tables**
> > >
> > > **Table R2-1.** Performance comparisons across different hyper-parameter selection (calculation) methods. **Bold** text indicates the best results.
> > >
> > > |  Dataset  |  UCon-SFDA  |  autoUCon-SFDA (Theory)  |  autoUCon-SFDA (Stat.)  |  SOTA Method Performance  |  SOTA Method |
> > > |:---------------------|:-------:|:-------:|:-------:| :-------:|:-------:|
> > > |  **Office31**  |  **90.6**  |  **90.6**  |  90.2  |  90.5  |  C-SFDA  |
> > > |  **OfficeHome**  |  73.6  |  73.6   |  **73.8**  |  73.5  |  C-SFDA  |
> > > |  **OfficeHome (partial set)**  |  80.3  |  **80.8**  |  80.7   |  79.7  |  AaD  |
> > > |  **VisDA2017**  |  **89.6**  |  89.3   |  89.2   |  88.4  |  I-SFDA  |
> > > |  **VisDA-RUST**  |  79.4  |  79.2  |  **79.5**  |  77.3  |  SF(DA)$^2$  |
> > > |  **DomainNet126**  |  71.5  |  71.5  |  **71.6**  |  69.6  |  GPUE  |
> > >
> > >
> > > **Table R2-2.** Hyper-parameter values across different datasets. "Orig. $\lambda_{DC}$", "Orig. $K_{PL}$", and "Orig. $\tau$" refer to the original values used in our paper, which are selected following the general hyper-parameter tuning pipeline in the literature. $\color{blue}\text{The hyper-parameters highlighted in blue}$ are directly calculated with theory-motivated hyper-parameter determination approaches, where "$\color{blue}\text{Init.}$" and "$\color{blue}\text{Final}$" indicate the first and the last training epochs, respectively. $\color{green}\text{The text in green}$ specifies the associated selection/calculation methods.
> > >
> > > |  Metric  |  Office31  |  OfficeHome  |  OfficeHome (partial set)  |  VisDA2017  |  VisDA-RUST  |  DomainNet126  |
> > > |:---------------------|:-------:|:-------:|:-------:| :-------:|:-------:|:-------:|
> > > |  Orig. $\lambda_{DC}\ \color{green}\text{(Original value used in our paper)}$  |  1.000  |  0.500  |  1.000  |  1.000  |  0.500  |  0.500  |
> > > |  $\color{blue}\text{New}\ \lambda_{DC}\ \color{green}\text{(Inconsistency Ratio: Guided by Theorem 4.1)}$  |  0.390  |  0.520  |  0.476  |  0.494  |  0.461  |  0.553  |
> > > |  Orig. $K_{PL}\ \color{green}\text{(Original value used in our paper)}$  |  2.000  |  2.000  |  2.000  |  1.000  |  2.000  |  2.000  |
> > > |  $\color{blue}\text{Init.}\ k_0\ \text{(Averaged)}\ \color{green}(k_0\ \text{in Theorem 4.2})$  |  1.320  |  1.535  |  1.513  |  1.341  |  1.348  |  1.644  |
> > > |  $\color{blue}\text{Final.}\ k_0\ \text{(Averaged)}\ \color{green}(k_0\ \text{in Theorem 4.2})$  |  1.003	 |  1.028	|  1.003  |	1.008  |	1.020  |	1.079
> > > |  Orig. $\tau\ \color{green}\text{(Original value used in our paper)}$  |  1.300	  |  1.100  |	1.100  |	1.100  |	1.100  |	1.100  |
> > > |  $\color{blue}\text{Init.}\ \tau_{t}\ \color{green}\text{(Calculated Using Theoretical Criterion: Remark 4.3)}$  |  1.308  |	1.265  |	1.238  |	1.790  |	1.674  |	1.232  |
> > > |  $\color{blue}\text{Final}\ \tau_{t}\ \color{green}\text{(Calculated Using Theoretical Criterion: Remark 4.3)}$  |  1.056  |	1.090  |	1.042  |	1.260  |	1.368  |	1.092  |
> > > |  $\color{blue}\tau_{s}\ \color{green}\text{(Derived from Statistical Insights: 10th percentile)}$  |  2.037  | 	1.230  |	1.268  |	1.164  |	1.163  |	1.264  |
> > >
> > > **Table R2-3.** Per source-target task configuration on DomainNet126. The metric notations are the same as in **Tables R2-2**.
> > >
> > > | Task | Acc. of Ucon-SFDA | $\color{blue}\text{Acc. of}$  $\color{blue}\text{autoUCon-SFDA}$ $\color{blue}\text{(Theory)}$ |$\color{blue}\text{Acc. of}$  $\color{blue}\text{autoUCon-SFDA}$ $\color{blue}\text{(Stat.)}$ | Orig. $\lambda_{DC}$  | $\color{blue}\text{New}\ \lambda_{DC}$ | Orig. $K_{PL}$ | $\color{blue}\text{Init.}\ k_0$ $\color{blue}\text{(Averaged)}$  | $\color{blue}\text{Final}\ k_0$ $\color{blue}\text{(Averaged)}$   | Orig. $\tau$ | $\color{blue}\text{Init.}\ \tau_{t}$        | $\color{blue}\text{Final}\ \tau_{t}$       | $\color{blue}\tau_{s}$  |
> > > |:---:|:---:|:---:|:---:|:---:|:---:|:---:|:---:|:---:|:---:|:---:|:---:|:---:|
> > > | **C$\to$S** | 66.5  | 64.5  | 66.0  | 0.50  | 0.52  | 2 | 1.70  | 1.08  | 1.1 | 1.20  | 1.08  | 1.23  |
> > > | **P$\to$C** | 69.3  | 70.3  | 70.0  | 0.50  | 0.59  | 2 | 2.33  | 1.11  | 1.1 | 1.30  | 1.11  | 1.17  |
> > > | **P$\to$R** | 81.0  | 81.4  | 81.4  | 0.50  | 0.45  | 2 | 1.64  | 1.04  | 1.1 | 1.28  | 1.08  | 1.36  |
> > > | **R$\to$C** | 75.2  | 77.0  | 77.3  | 0.50  | 0.59  | 2 | 1.45  | 1.08  | 1.1 | 1.19  | 1.09  | 1.27  |
> > > | **R$\to$P** | 71.1  | 71.3  | 71.0  | 0.50  | 0.58  | 2 | 1.39  | 1.09  | 1.1 | 1.17  | 1.11  | 1.32  |
> > > | **R$\to$S** | 64.3  | 68.1  | 67.7  | 0.50  | 0.61  | 2 | 1.52  | 1.07  | 1.1 | 1.20  | 1.09  | 1.23  |
> > > | **S$\to$P** | 68.1  | 67.9  | 67.6  | 0.50  | 0.55  | 2 | 1.49  | 1.08  | 1.1 | 1.30  | 1.08  | 1.27  |
> > > | **Avg.** | 71.5  | 71.5  | 71.6  | 0.50  | 0.55  | 2 | 1.64  | 1.08  | 1.1 | 1.23  | 1.09  | 1.26  |

---

### Author Response · Authors · 2024-11-21
**Global Response**

Dear reviewers,

Thank you for dedicating your valuable time and effort to reviewing our paper. We  appreciate your insightful feedback and constructive comments, which have greatly helped us enhance the presentation of our work. We are grateful for your recognition of the contributions and strengths of our paper, as outlined below:

* Our paper **addresses a significant and practical topic**, the source-free domain adaptation (SFDA) problem, which is critical for real-world applications (Reviewer GdvL).

* We provide a **comprehensive theoretical analysis** of the previously overlooked uncertainty problem in SFDA methods within a unified distributionally robust optimization (DRO) framework. The additional analysis examines both positive and negative sample uncertainties and guides the development of our uncertainty control algorithm for SFDA (Reviewers Bkmw, GdvL, 1NWJ).

* **Extensive experiments** demonstrate the effectiveness of our proposed approach, UCon-SFDA, which consistently outperforms existing SFDA methods. Our results underscore the practical advantages of the method derived from our theoretical insights (Reviewers Bkmw, GdvL, 1NWJ).

* The paper is **well-written**, with thorough theoretical analysis, illustrative figures, and a clear presentation that facilitates understanding (Reviewer GdvL).

We have carefully reviewed each of your queries, concerns, and remarks. In preparing a revised version, we have thoroughly addressed every single comment to improve the clarity and rigor of our paper. Below highlights the key changes we have done.

* We appreciate your insights on emphasizing the theoretical contributions and their practical relevance (Reviewers Bkmw, 1NWJ). In response, we have streamlined the theoretical analysis, added remarks connecting theory to practice, and included a notation table for clarity in the revised manuscript (see **Remarks 4.1-4.4** and **Appendix A.1**).

* We value your suggestions to better highlight the novelty of our approach, particularly the consistency loss (Reviewer Bkmw). We have now elaborated on how our theoretical findings guided the design of the consistency loss (see **Remarks 4.1 and 4.2**) and included additional experiments to demonstrate its advantages (see **Appendix C.6**).

* Addressing your comments on hyper-parameters (Reviewer Bkmw, GdvL, 1NWJ), we have clarified that most parameters were adopted from previous work or derived from theoretical insights, thus requiring lightweight tuning. For the few parameters needing adjustment, we employed a straightforward tuning strategy based on the common hyperparameter tuning pipeline in the prior studies, resulting in effective selections (see **Appendix B**). We have also provided new experimental results on hyper-parameter sensitivity analysis (see **Appendix C.5**).

* On computational complexity (Reviewer 1NWJ), we acknowledge the increased demands but emphasize that our approach remains comparable to existing state-of-the-art methods while achieving superior performance (see **Appendix C.7**).

We hope that our responses have sufficiently addressed all of the concerns raised. Should further details or clarifications are needed, please let us know and we would be happy to provide them. Thank you!

---

### Meta-Review · Area_Chair_rSoa · 2024-12-21

**Metareview:**

The paper received three reviews with ratings of 5, 6, and 8. The reviewers acknowledged its contribution in providing an in-depth theoretical analysis of uncertainties in previous SFDA methods and developing an effective uncertainty control algorithm for SFDA. However, all reviewers expressed a common concern regarding the large number of hyperparameters and the challenges in determining their values for practical applications, particularly in the context of SFDA. It is essential for the authors to include a parameter sensitivity analysis and clarify the relationship between Eq.(7) and the consistency regularization loss commonly used in previous work with weak and strong augmentations.

**Additional Comments On Reviewer Discussion:**

Some reviewers participated in the discussions, but were not entirely unconvinced by the authors rebuttal regarding the hyperparameter issue.

---

### Decision · Program_Chairs · 2025-01-22

Accept (Poster)